# On the Analysis of GAN-based Image-to-Image Translation using Gaussian Noise Injection

**Chaohua Shi**[1] **Kexin Huang**[2] **Lu Gan**[*,3] **Hongqing Liu**[4]

**Mingrui Zhu**[1] **Nannan Wang**[*,1] **Xinbo Gao**[1,4]

[1] Xidian University    [2] National University of Defense Technology

[3] Brunel University    [4] Chongqing University of Posts and Telecommunications

chshi2004@gmail.com, hkxin91@outlook.com, lu.gan@brunel.ac.uk

{mrzhu, nnwang}@xidian.edu.cn, {hongqingliu, xbgao}@cqupt.edu.cn

## Abstract

Image-to-image (I2I) translation is vital in computer vision tasks like style transfer and domain adaptation. While recent advances in GAN have enabled high-quality sample generation, real-world challenges such as noise and distortion remain significant obstacles. Although Gaussian noise injection during training has been utilized, its theoretical underpinnings have been unclear. This work provides a robust theoretical framework elucidating the role of Gaussian noise injection in I2I translation models. We address critical questions on the influence of noise variance on distribution divergence, resilience to unseen noise types, and optimal noise intensity selection. Our contributions include connecting $f$-divergence and score matching, unveiling insights into the impact of Gaussian noise on aligning probability distributions, and demonstrating generalized robustness implications. We also explore choosing an optimal training noise level for consistent performance in noisy environments. Extensive experiments validate our theoretical findings, showing substantial improvements over various I2I baseline models in noisy settings. Our research rigorously grounds Gaussian noise injection for I2I translation, offering a sophisticated theoretical understanding beyond heuristic applications.

## 1 Introduction

Image-to-image (I2I) translation has seen remarkable advancements in recent years and has emerged as a thriving field within computer vision. In particular, models based on Generative Adversarial Network (GAN) have gained significant attention due to their ability to generate high-quality images and fast inference speed (Goodfellow et al., 2020). However, their performance significantly suffers when handling noisy or distorted inputs, as shown in Fig. 1. The degradation of input image quality is a common occurrence in real-world scenarios, spanning from low-light conditions to data transmission through noisy channels (Anaya & Barbu, 2018; Plotz & Roth, 2017; Yue et al., 2020; Zamir et al., 2020), underscoring a critical vulnerability intrinsic to I2I models.

To address this challenge, we explore a simple and widely applicable approach for boosting the noise resilience of I2I translation models. This involves injecting isotropic Gaussian noise into source domain images during training, as shown in Fig. 1. Our research tackles three core questions:

- How does the variance of Gaussian noise introduced during training impact the divergence between real and generated distributions?

- How does the presence of Gaussian noise in training data influence the model's ability to handle unseen noise distributions and intensities during inference?

- Is it possible to identify an optimal noise intensity during training that guarantees consistent performance across diverse noise intensities during inference?

---

[*] Correspondence to lu.gan@brunel.ac.uk and nnwang@xidian.edu.cn

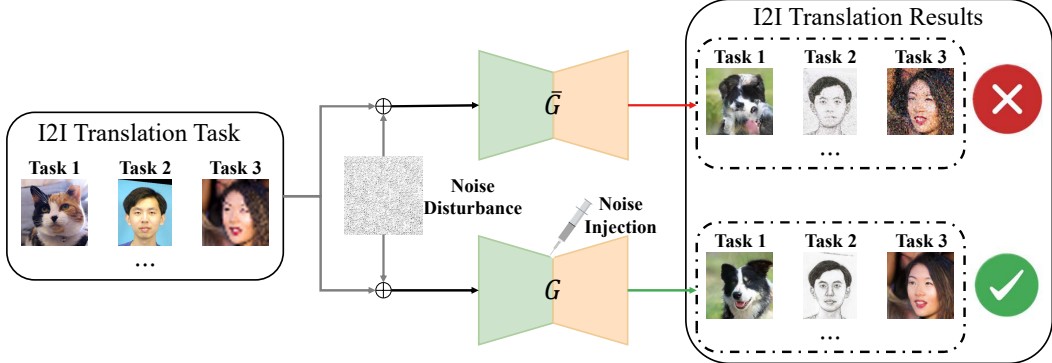

Figure 1: Overview of our framework. When dealing with noisy inputs, the original I2I translation model will get failed results (top). By applying the **Noise Injection** to the original model during training, we successfully improve the noise robustness of the original model (bottom).

Our contributions to this discourse are multifaceted: Firstly, we establish a novel connection between $f$-divergence and score matching, shedding light on the implications of Gaussian noise injection on I2I model training. This provides valuable insights into how injected noise impacts the alignment of probability distributions. Secondly, we demonstrate that for arbitrary source signal, the robustness of I2I systems to Gaussian noise implies resilience to other noise types with a matched covariance matrix, underscoring the advantages of Gaussian noise injection for enhancing general robustness. Thirdly, our study addresses the selection of the optimal noise variance, ensuring stability across diverse independent and identically distributed (i.i.d.) noise forms. Experimental results showcase the significant performance improvement achieved by the Gaussian noise injection technique, effectively reducing sensitivity to noise in various I2I translation models operating in noisy environments.

## 2 RELATED WORK

In the realm of reliable I2I translation, Chrysos et al. (2020) introduced the Robust Conditional GAN (RoCGAN), featuring a dual-pathway generator architecture with a shared decoder. Empirical evaluations in face super-resolution and inpainting tasks demonstrated RoCGAN's ability to produce consistent outputs even in the presence of noise and perturbations. However, the dual-pathway architecture's computational demands and prolonged training duration raise concerns, particularly regarding its adaptability to larger generative models where computational efficiency is crucial. Moreover, while Chrysos et al. (2020); Wang et al. (2021); Jia et al. (2021) also explored noise injection for GAN-based I2I, the research mainly focused on empirical findings, lacking theoretical analysis. Additionally, the simulation results were limited to supervised I2I with paired training samples, leaving uncertainties about its effectiveness for unsupervised I2I models.

On the theoretical front, studies on unconditional GANs (Arjovsky & Bottou, 2017; Jenni & Favaro, 2019) have demonstrated that noise injection during training can significantly improve learning consistency and mitigate issues like model overfitting. Recently, this approach has gained interest in adversarial defence, with theoretical solid justification and promising empirical results in boosting robustness and resilience (Cohen et al., 2019; Goodfellow et al., 2014; Madry et al., 2017; Pinot et al., 2019; Lee et al., 2019; Xie et al., 2023; Yang et al., 2023a; Dong & Xu, 2023). Yet it is essential to note that these successes in classification do not directly translate to the complex task of I2I translation. The distinction is profound - classification tasks culminate in discrete outputs, while I2I models generate entire images, introducing unique challenges.

Advancements in diffusion-based probabilistic models (Song et al., 2021; 2020; Dhariwal & Nichol, 2021) have also explored controlled Gaussian noise addition to input images during the diffusion process. These studies have shown promise in both supervised (Saharia et al., 2022b;a; Batzolis et al., 2021; Li et al., 2022) and unsupervised (Sasaki et al., 2021; Choi et al., 2021; Zhao et al., 2022; Kwon & Ye, 2022; Su et al., 2023) I2I tasks. However, as we will demonstrate later, these models also exhibit vulnerability to noisy inputs, highlighting the need to explore noise-robustness strategies.

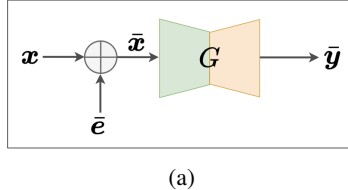 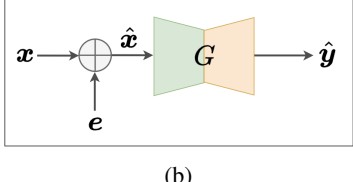

(a)                                         (b)

Figure 2: System diagram. (a) Training Phase: For each clean image $x$ in the training set, it is augmented with i.i.d. Gaussian noise, resulting in $\bar{x}$. The generator then produces the corresponding output, $\bar{y}$. (b) Inference Phase: A noise-corrupted test image, $\hat{x}$, serves as the input to the generator, synthesizing the image $\hat{y}$.

In summary, while Gaussian noise injection can enhance the noise robustness of I2I models, a comprehensive investigation of its effectiveness, implications, and limitations is required. The **main aim** of this paper is to provide a theoretical understanding of Gaussian noise injection's role in boosting the robustness of I2I translation models, rather than introducing new network architectures.

## 3 THEORETICAL ANALYSIS

Consider an image $x \in \mathbb{R}^d$ from the source domain $\mathcal{X}$. In the GNI (Gaussian Noise Injection) system of Fig. 2a, the GAN-based generative model $G$ is trained by adding isotropic Gaussian noise $\bar{e} \sim \mathcal{N}(0, \sigma_t^2 I_d)$ to each $x \in \mathcal{X}$. This produces a noisy training image $\bar{x} = x + \bar{e}$, and the corresponding output from the generator is $\bar{y} = G(\bar{x})$. During inference, a noisy input image is represented as $\hat{x} = x + e$, and the corresponding output is $\hat{y} = G(\hat{x})$, depicted in Fig. 2b. Unlike prior methods such as randomized smoothing for classification, which inject Gaussian noise during both training and inference, our framework only adds noise during training. This ensures quicker inference, facilitating easy integration with numerous I2I systems.

In what follows, we utilize $f$-divergence (Polyanskiy, 2019) to study the influence of training noise variance, adaptability to unseen noise, and the identification of optimal training noise intensity for consistent performance. For brevity, notations, definitions, and proofs can be found in the Appendix.

### 3.1 RELATION BETWEEN $f$ DIVERGENCE AND SCORE FUNCTION

Aligning the probability distributions of accurate and generated data is crucial to I2I translation, and misalignment can result in unrealistic results. In this context, we examine the influence of Gaussian noise injected into the source domain on this alignment. The theorem below describes how the $f$-divergence between these distributions varies with the noise variance.

**Theorem 1.** *Let $P_{X,Y}$ and $Q_{X,Y}$ be two joint distributions on $\mathcal{X} \times \mathcal{Y}$ representing real data and the data generated by a model, respectively. Define $\bar{X} = X + \sigma N$, where $N \sim \mathcal{N}(0, I_d)$ is standard $d$-dimensional isotropic Gaussian noise. Let $\bar{P}_{\bar{X},Y}$ and $\bar{Q}_{\bar{X},Y}$ represent the corresponding distributions after Gaussian noise injection with their respective probability densities $\bar{p}(\bar{x}, y)$ and $\bar{q}(\bar{x}, y)$. For the generator function $f$, if its second order derivative $f''$ exists and $D_f\left(P_{X,Y} \parallel Q_{X,Y}\right)$ is finite, then $D_f\left(\bar{P}_{\bar{X},Y} \parallel \bar{Q}_{\bar{X},Y}\right)$ satisfies*

$$\frac{\mathrm{d}}{\mathrm{d}\sigma^2} D_f\left(\bar{P}_{\bar{X},Y} \parallel \bar{Q}_{\bar{X},Y}\right) = -\frac{1}{2}\eta_f(\sigma^2), \tag{1}$$

*in which $\eta_f(\sigma^2)$ represents the weighted mean square error between two score functions*

$$\eta_f(\sigma^2) = \mathbb{E}_{\bar{P}_{\bar{X},Y}}\left\{\frac{\bar{p}(\bar{x}, y)}{\bar{q}(\bar{x}, y)} f''\left(\frac{\bar{p}(\bar{x}, y)}{\bar{q}(\bar{x}, y)}\right) \|\nabla_{\bar{x}} \log \bar{p}(\bar{x}, y) - \nabla_{\bar{x}} \log \bar{q}(\bar{x}, y)\|^2\right\}, \tag{2}$$

*where $\nabla_{\bar{x}} \log \bar{p}(\bar{x}, y)$ and $\nabla_{\bar{x}} \log \bar{q}(\bar{x}, y)$ are the score functions of $\bar{p}(\bar{x}, y)$ and $\bar{q}(\bar{x}, y)$, respectively.*

The above theorem unveils how the rate of change of $D_f\left(\bar{P} \parallel \bar{Q}\right)$ concerning $\sigma^2$ is portrayed through $\eta_f(\sigma_t^2)$. In the case of KL-divergence, where $f(t) = t \log t$, we can derive that

$$\eta_{KL}(\sigma^2) = \mathbb{E}_{\bar{P}_{\bar{X},Y}} \|\nabla_{\bar{x}} \log \bar{p}(\bar{x}, y) - \nabla_{\bar{x}} \log \bar{q}(\bar{x}, y)\|^2, \tag{3}$$

identifying it as the Fisher divergence between $\bar{p}(\bar{x}, y)$ and $\bar{q}(\bar{x}, y)$ (Lyu, 2012; Verdú, 2010).

For small values of $\sigma = \sigma_t$, a Taylor series expansion yields

$$D_f\left(P_{X,Y} \parallel Q_{X,Y}\right) = D_f\left(\bar{P}_{X+\sigma_t N, Y} \parallel \bar{Q}_{X+\sigma_t N, Y}\right) + \frac{\sigma_t^2}{2}\eta_f(\sigma_t^2) + o(\sigma_t^2). \quad (4)$$

Through optimization of the noise-injected term $D_f\left(\bar{P}_{X+\sigma_t N, Y} \parallel \bar{Q}_{X+\sigma_t N, Y}\right)$ for minimizing the divergence between $\bar{p}(x + \sigma_t N, y)$ and $\bar{q}(x + \sigma_t N, y)$, the term $\eta_f(\sigma_t^2)$ tends to decrease. In the ideal scenario where $\bar{p}(x + \sigma_t N, y) = \bar{q}(x + \sigma_t N, y)$, the first two terms on the right side vanish, leading to $D_f\left(P_{X,Y} \parallel Q_{X,Y}\right) = o(\sigma_t^2)$. Hence, by injecting Gaussian noise with small $\sigma_t^2$ and aligning the noise-perturbed distributions during training, the model is guided to align the original, noise-free distributions as well, which results in a coherent I2I transformation.

While previous research in information theory and machine learning has explored the relationship between KL and Fisher divergences for marginal distributions (Verdú, 2010; Lyu, 2012; Kong et al., 2023), we extend this understanding to $f$-divergence of joint distributions. This broader view deepens our insight into divergence and noise injection, thereby fortifying the theoretical base for modelling and manipulating complex dependencies within the I2I translation framework.

## 3.2 PERFORMANCE ANALYSIS FOR MISMATCHED NOISY INPUTS

This subsection explores the system's capability to handle unseen noise during inference, as depicted in Fig. 2b. Following the conventions of Theorem 1, let $X$ and $Y$ represent the clean source and target domain random variables, and $\bar{X}, \bar{Y} = G(\bar{X})$ denote the noisy counterparts during training. Considering a new noisy input $\hat{X} = X + E$ during inference, where $E$ is independent of $X$ with zero mean and covariance matrix of $\Sigma_e$, the corresponding output is denoted by $\hat{Y} = G(\hat{X})$. Marginal distributions are denoted by $\bar{P}_{\bar{X}}$, $\hat{P}_{\hat{X}}$, $\bar{Q}_{\bar{Y}}$, $\hat{Q}_{\hat{Y}}$, and joint distributions by $\bar{Q}_{\bar{X},\bar{Y}}$ and $\hat{Q}_{\hat{X},\hat{Y}}$. Using data-processing properties in $f$-divergence (Polyanskiy, 2019), we have

$$D_f\left(\hat{Q}_{\hat{X},\hat{Y}} \parallel \bar{Q}_{\bar{X},\bar{Y}}\right) = D_f\left(\hat{P}_{\hat{X}} \parallel \bar{P}_{\bar{X}}\right), \quad D_f\left(\hat{Q}_{\hat{Y}} \parallel \bar{Q}_{\bar{Y}}\right) \leq D_f\left(\hat{P}_{\hat{X}} \parallel \bar{P}_{\bar{X}}\right). \quad (5)$$

The equations reveal insights for I2I models handling unseen noise. They show that the joint input/output divergence equals the input marginal divergence. The output divergence is bounded by this value, with equality under a reversible model. Many I2I models exhibit near reversibility, approximating translation inversion. Paired I2I scenarios (Isola et al., 2017) span seasonal shifts, sketch-to-realistic image conversion, face resolution alteration, and medical image translations (e.g., MRI to CT scans). Unpaired models like CycleGAN (Zhu et al., 2017) establish near reversibility through cycle-consistency losses. Leveraging this property, the input marginal divergence $D_f\left(\hat{P}_{\hat{X}} \parallel \bar{P}_{\bar{X}}\right)$ can estimate model behavior with unseen noise. For general signal sources, obtaining a closed-form expression of the $f$-divergence is challenging. But with a Gaussian signal source, we can explicitly derive the KL-divergence, as shown in the following Lemma:

**Lemma 1.** *Let $X$ be $d$-dimensional random variable with normal distribution $\mathcal{N}(\mu_s, \Sigma_s)$. Assume that it is corrupted by noise $E$ with zero mean and covariance matrix $\Sigma_e$, independent of $X$. Denote $\rho(\sigma_t^2, \Sigma_e) \triangleq D_{KL}\left(\hat{P}_{X+E} \parallel \mathcal{N}(\mu_s, \Sigma_s + \sigma_t^2 I_d)\right)$. Then, $\rho(\sigma_t^2, \Sigma_e)$ can be expressed as*

$$\rho(\sigma_t^2, \Sigma_e) = -h(X + E) + \frac{d}{2}\log(2\pi) + \frac{1}{2}\log|\Sigma_2| + \frac{1}{2}\mathrm{Tr}\left(\Sigma_2^{-1}\Sigma_1\right), \quad (6)$$

*in which $h(X + E)$ denotes the differential entropy of $X + E$, $\Sigma_1 = \Sigma_s + \Sigma_e$ and $\Sigma_2 = \Sigma_s + \sigma_t^2 I_d$.*

Eq. (6) allows for a closed-form solution of KL-divergence for a Gaussian source corrupted by arbitrary noise. In particular, the lower bound of $\rho(\sigma_t^2, \Sigma_e)$ is achieved when $e$ follows a Gaussian distribution $\mathcal{N}(0, \Sigma_e)$, i.e., $\rho(\sigma_t^2, \Sigma_e) \geq \rho_g(\sigma_t^2, \Sigma_e)$ with

$$\rho_g(\sigma_t^2, \Sigma_e) = \frac{1}{2}\left(\mathrm{Tr}\left(\Sigma_2^{-1}\Sigma_1\right) + \log\frac{|\Sigma_2|}{|\Sigma_1|} - d\right). \quad (7)$$

This indicates that given the same $\Sigma_e$, non-Gaussian noise yields higher KL-divergence, as Gaussian distribution maximizes the entropy. The next Theorem characterizes the behavior of $\rho(\sigma_t^2, \Sigma_e)$:

**Theorem 2.** *Consider the KL-divergences denoted by $\rho(\sigma_t^2, \mathbf{\Sigma}_e)$ in (6) for general noise, and $\rho_g(\sigma_t^2, \mathbf{\Sigma}_e)$ in (7) for Gaussian noise. Under these definitions, the following properties hold:*

1. *Let $\mathbf{\Sigma}_e = \sigma_e^2 \mathbf{\Sigma}_{\widetilde{e}}$, in which $\mathbf{\Sigma}_{\widetilde{e}}$ is normalized covariance matrix with $\mathrm{Tr}(\mathbf{\Sigma}_{\widetilde{e}}) = d$. Then, $\rho(\sigma_t^2, \sigma_e^2 \mathbf{\Sigma}_{\widetilde{e}})$ is convex with respect to $\sigma_e^2$. Additionally, for small $\sigma_e^2$ with $\sigma_e^2 \ll 1$, the following approximation is valid:*

$$\rho(\sigma_t^2, \sigma_e^2 \mathbf{\Sigma}_{\widetilde{e}}) = \rho_g(\sigma_t^2, \sigma_e^2 \mathbf{\Sigma}_{\widetilde{e}}) + o(\sigma_e^2). \tag{8}$$

2. *If $\mathbf{\Sigma}_e \geq \frac{\sigma_t^2}{2} \mathbf{I}_d$, the inequality $\rho(\sigma_t^2, \mathbf{\Sigma}_e) < \rho(0, \mathbf{\Sigma}_e)$ is satisfied.*

**Part 1** of the theorem implies that the KL-divergence first decreases and then increases with respect to $\sigma_e^2$, owing to the convex nature of $\rho(\sigma_t^2, \sigma_e^2 \mathbf{\Sigma}_{\widetilde{e}})$. Specifically, for Gaussian noise with $\mathbf{\Sigma}_{\widetilde{e}} = \mathbf{I}_d$, the global optimum is at $\sigma_e^2 = \sigma_t^2$. Furthermore, Eq. (8) indicates that non-Gaussian noise with small $\sigma_e^2$ leads to a KL-divergence close to that of Gaussian noise with the same covariance matrix. Hence, an I2I system that is robust to Gaussian noise can also tolerate other types of noises with the same $\mathbf{\Sigma}_e$.

**Part 2** of the theorem establishes a comparison between a system trained with Gaussian noise injection and one trained only on clean images. Specifically, it asserts that for certain noise levels, as characterized by $\mathbf{\Sigma}_e \geq \frac{\sigma_t^2}{2} \mathbf{I}_d$, the system trained with noise injection can better handle noisy inputs compared to a system trained only on clean images. This not only underscores the practical advantage of noise injection, but also provides a sound theoretical foundation for its effectiveness.

The next Theorem discusses the case for a non-Gaussian signal.

**Theorem 3.** *Let $\mathbf{X}$ be a $d$-dimensional random vector with an arbitrary probability distribution and finite entropy $h(\mathbf{X})$. Denote $\theta(\sigma_t^2, \sigma_e^2 \mathbf{\Sigma}_{\widetilde{e}}) \triangleq D_{KL}\left(\hat{P}_{\mathbf{X}+\mathbf{E}} \| \bar{P}_{\mathbf{X}+\sigma_t \mathbf{N}}\right)$, where the definitions of $\mathbf{E}$, $\mathbf{\Sigma}_{\widetilde{e}}$, $\mathbf{N}$, $\sigma_t$ and $\sigma_e^2$ are the same as those in Theorem 2. Let $\theta_g(\sigma_t^2, \sigma_e^2 \mathbf{\Sigma}_{\widetilde{e}})$ denotes the special case of $\theta(\sigma_t^2, \sigma_e^2 \mathbf{\Sigma}_{\widetilde{e}})$ when $\mathbf{E}$ is Gaussian noise. Then,*

1. *For small $\sigma_e^2$ with $\sigma_e^2 \ll 1$,*

$$\theta(\sigma_t^2, \sigma_e^2 \mathbf{\Sigma}_{\widetilde{e}}) = \theta_g(\sigma_t^2, \sigma_e^2 \mathbf{\Sigma}_{\widetilde{e}}) + o(\sigma_e^2); \tag{9}$$

2. *When $\mathbf{E}$ is also iid Gaussian, $\theta_g(\sigma_t^2, \sigma_e^2 \mathbf{I}_d) \triangleq D_{KL}\left(\hat{P}_{\mathbf{X}+\sigma_e \mathbf{N}} \| \bar{P}_{\mathbf{X}+\sigma_t \mathbf{N}}\right)$ satisfies*

$$\frac{\mathrm{d}}{\mathrm{d}\sigma_e^2} \theta_g(\sigma_t^2, \sigma_e^2 \mathbf{I}_d) = \mathbb{E}_{\hat{p}(\hat{\mathbf{x}})} \left\{ -\frac{1}{2} \|\nabla_{\hat{\mathbf{x}}} \log \hat{p}(\hat{\mathbf{x}})\|^2 + \frac{1}{2} \nabla_{\hat{\mathbf{x}}} \log \hat{p}(\hat{\mathbf{x}}) \cdot \nabla_{\bar{\mathbf{x}}} \log \bar{p}(\bar{\mathbf{x}}) \right\} \tag{10}$$

Eq. (9) generalizes the result of Eq. (8) to non-Gaussian sources, demonstrating that an I2I system's resilience to Gaussian noise ensures robustness against noises with the same covariance matrices, regardless of whether the input signals are Gaussian or non-Gaussian. In the special case when $\mathbf{E}$ is also iid Gaussian, one can get $\theta_g(\sigma_t^2, \sigma_e^2 \mathbf{I}_d)$ by integrating the right-hand side of Eq. (10). In particular, $\frac{\mathrm{d}}{\mathrm{d}\sigma_e^2} \theta_g(\sigma_t^2, \sigma_e^2 \mathbf{I}_d) = 0$ when $\sigma_e^2 = \sigma_t^2$. Hence, when $\sigma_e^2 - \sigma_t^2$ is small, using Taylor series expansion at $\sigma_t^2$, we have $\theta_g(\sigma_t^2, \sigma_e^2 \mathbf{I}) = o(\sigma_t^2 - \sigma_e^2)$.

### 3.3 SELECTION OF TRAINING NOISE INTENSITY

For i.i.d. inference noise with bounded $\sigma_e^2$, the following corollary provides insights into choosing the optimal training noise level by either minimizing worst-case KL-divergence or the average KL-divergence given uniform noise variance distribution:

**Corollary 1.** *Given an i.i.d. input noise $\mathbf{e}$ with $\mathbf{\Sigma}_e = \sigma_e^2 \mathbf{I}_d$ and a bounded variance $0 \leq \sigma_e^2 \leq \lambda_{\max}$, define $\sigma_{t,o}^2$ as the optimal noise level that minimizes the worst-case KL distance $\rho(\sigma_t^2, \sigma_e^2 \mathbf{I}_d)$:*

$$\sigma_{t,o}^2 = \arg\min_{\sigma_t^2} \left\{ \max_{0 \leq \sigma_e^2 \leq M} \rho(\sigma_t^2, \sigma_e^2 \mathbf{I}_d) \right\}. \tag{11}$$

*For this optimal level, it satisfies $\rho(\sigma_{t,o}^2, \mathbf{0}_d) = \rho(\sigma_{t,o}^2, \lambda_{\max} \mathbf{I}_d)$. Besides, if $\sigma_e^2$ is uniformly distributed between 0 and $\lambda_{\max}$, i.e., $\sigma_e^2 \sim \mathcal{U}(0, \lambda_{\max})$, the optimal training noise intensity $\bar{\sigma}_{t,o}^2$ that minimizes*

the average KL-divergence is $\frac{1}{2}\lambda_{\max}$, i.e.,

$$\bar{\sigma}_{t,o}^2 = \arg\min_{\sigma_t^2} \mathbb{E}_{\sigma_e^2 \sim \mathcal{U}(0,\lambda_{\max})} \left\{ \rho(\sigma_t^2, \sigma_e^2 \boldsymbol{I}_d) \right\} = \frac{1}{2}\lambda_{\max}. \tag{12}$$

This corollary offers a theoretically sound method for determining the optimal training noise variance for an arbitrary type of i.i.d. inference noise. It implies that to determine $\sigma_{t,o}^2$ in Eq. (11), one can initiate the search at $\bar{\sigma}_{t,o} = \lambda_{\max}/2$ and proceed numerically until the condition $\rho(\sigma_{t,o}^2, \boldsymbol{0}_d) = \rho(\sigma_{t,o}^2, \lambda_{\max}\boldsymbol{I}_d)$ holds true.

## 4 EXPERIMENT

### 4.1 EXPERIMENTAL SETTINGS

**Baselines & Datasets:** We have employed the Gaussian noise-injected training methodology in various I2I translation models and contrasted them with their original baselines. Specifically, 1. HiFaceGAN (Yang et al., 2020), a GAN-based I2I model primarily utilized for Face Super-Resolution task on real-life facial photographs; 2. GP-UNIT (Yang et al., 2023b), a generative prior-based image translation model for converting images between unpaired data domains, such as **Cat→Dog**; 3. Sketch Transformer (Zhu et al., 2021), a Transformer-based photo-sketch paired transfer model. During training, we follow the default settings of each baseline model and use the corresponding datasets: FFHQ (Karras et al., 2019) (HiFaceGAN), AAHQ (Choi et al., 2020) (GP-UNIT), and CUFS (Wang et al., 2018) (Sketch Transformer). For more details, please refer to Appendix C.

**Evaluation Metrics:** In line with prior research (Karras et al., 2019; Yang et al., 2023b; Zhao et al., 2022), our evaluation primarily relies on the widely adopted Fréchet Inception Distance (FID) (Heusel et al., 2017) and Kernel Inception Distance (KID) (Bińkowski et al., 2018) to assess the disparity between generated images and target datasets regarding their respective distributions. Additionally, for I2I translation models trained with paired datasets, we incorporate Learned Perceptual Image Patch Similarity (LPIPS) (Zhang et al., 2018) and Peak Signal-to-Noise Ratio (PSNR). This metric enables more accurate image similarity measurement, aligning with human perception.

**Degradation Settings:** To create noisy images, we first normalize pixel values to the range $[0, 1]$. Next, we add noise and clip the pixel values to stay within $[0, 1]$. The pixel values are then rescaled to $[0, 255)$ for image file creation. During training, zero-mean, isotropic Gaussian noise with $\sigma_t^2 = 0.04$ (unless specified otherwise) is introduced. In the inference stage, we evaluate three models under five signal-independent noise types (Gaussian, Uniform, Color, Laplacian, and Salt & Pepper), each with six intensity levels applied to all input images. Furthermore, we explore the performance under signal-dependent noises, blur, JPEG compression, and other corruptions, as modelled in Imagenet-C (Hendrycks & Dietterich, 2018). Additional details are available in Appendix C, Table 4.

### 4.2 EXPERIMENTAL RESULTS

**Qualitative and Quantitative Evaluations:** Tables 1 and 2 show the quantitative results of Cat→Dog and Photo→Sketch translations, respectively, while Figs. 3 and 4 provide the corresponding qualitative results. Our empirical findings, obtained under the default setting of $\sigma_t^2 = 0.04$, closely mirror the insights from our theoretical analysis. **First**, as suggested by Theorem 1, the Gaussian noise-injected model demonstrates the ability to effectively align noise-influenced distributions, ensuring a coherent transformation between source and target domains. It outperforms GP-UNIT on Cat→Dog translation across all settings, including clean inputs (Table 1). For Photo→Sketch, only marginal objective degradation is observed on clean data, which is nearly visually imperceptible. **Second**, while the baseline method excels on clean images, it experiences a significant decline under noisy conditions. In contrast, GNI method demonstrates remarkable resilience across diverse noise types and intensities. Though Theorem 2 considers Gaussian signals, experiments indicate Gaussian injection provides noise robustness even for non-Gaussian images. In summary, simulation results substantiate the theory of aligning noisy/clean distributions and showcase generalized noise robustness.

Fig. 3 also compares the noise injection approach to DiffuseIT (Kwon & Ye, 2022) on Cat→Dog translation. Despite relying on isotrophic Gaussian denoising, DiffuseIT's performance drops substantially under colored Gaussian noise. In contrast, GP-UNIT with GNI exhibits superior robustness across all noise types.

Table 1: Quantitative comparison on the **Cat→Dog** image translation task (reference-guided). In the reference-guided approach, we randomly select 10 target domain images from the test set as additional guides and synthesize a total of 5000 images. Furthermore, we employ the FID and $1000 \times$ KID metrics to evaluate.

| Noise Type | Metric | Method | Noise Intensity | | | | | | |
| --- | --- | --- | --- | --- | --- | --- | --- | --- | --- |
| | | | Clean | S1 | S2 | S3 | S4 | S5 | S6 |
| Gaussian Noise | FID↓ | Baseline | 17.82 | 18.28 | 18.95 | 20.82 | 22.13 | 26.95 | 36.84 |
| | | $+\mathcal{N}(0, 0.04)$ | **16.47** | **16.21** | **16.24** | **16.23** | **16.19** | **16.26** | **16.31** |
| | KID↓ | Baseline | 7.08 | 7.91 | 8.79 | 10.53 | 11.35 | 15.48 | 23.03 |
| | | $+\mathcal{N}(0, 0.04)$ | **5.35** | **5.07** | **5.15** | **5.17** | **5.21** | **5.23** | **5.41** |
| Uniform Noise | FID↓ | Baseline | 17.82 | 18.31 | 19.17 | 21.19 | 22.31 | 28.42 | 43.31 |
| | | $+\mathcal{N}(0, 0.04)$ | **16.47** | **16.13** | **16.16** | **16.21** | **16.32** | **16.27** | **16.15** |
| | KID↓ | Baseline | 7.08 | 7.98 | 8.81 | 10.76 | 11.45 | 16.42 | 28.83 |
| | | $+\mathcal{N}(0, 0.04)$ | **5.35** | **5.06** | **5.09** | **5.23** | **5.26** | **5.27** | **5.43** |
| Color Noise | FID↓ | Baseline | 17.82 | 18.48 | 19.94 | 22.91 | 25.11 | 31.85 | 47.26 |
| | | $+\mathcal{N}(0, 0.04)$ | **16.47** | **16.17** | **16.31** | **16.15** | **16.18** | **16.21** | **16.19** |
| | KID↓ | Baseline | 7.08 | 8.31 | 9.68 | 12.09 | 14.02 | 19.33 | 31.78 |
| | | $+\mathcal{N}(0, 0.04)$ | **5.35** | **5.09** | **5.26** | **5.13** | **5.16** | **5.36** | **5.22** |
| Laplacian Noise | FID↓ | Baseline | 17.82 | 18.39 | 19.06 | 20.38 | 21.13 | 24.77 | 30.95 |
| | | $+\mathcal{N}(0, 0.04)$ | **16.47** | **16.16** | **16.13** | **16.12** | **16.15** | **16.14** | **16.19** |
| | KID↓ | Baseline | 7.08 | 8.01 | 8.72 | 9.99 | 10.74 | 13.49 | 18.67 |
| | | $+\mathcal{N}(0, 0.04)$ | **5.35** | **5.07** | **5.15** | **5.16** | **5.17** | **5.26** | **5.27** |
| Salt & Pepper Noise | FID↓ | Baseline | 17.82 | 18.62 | 19.97 | 22.41 | 25.68 | 29.85 | 35.11 |
| | | $+\mathcal{N}(0, 0.04)$ | **16.47** | **16.11** | **16.28** | **16.26** | **16.08** | **16.07** | **16.99** |
| | KID↓ | Baseline | 7.08 | 8.49 | 9.62 | 11.76 | 14.58 | 17.93 | 21.99 |
| | | $+\mathcal{N}(0, 0.04)$ | **5.35** | **4.99** | **5.23** | **5.26** | **5.19** | **5.18** | **5.29** |

Table 2: Quantitative comparison on **Photo→Sketch** image translation tasks. For photo-to-sketch, we evaluate sketch synthesis using 338 test photos from the test set.

| Noise Type | Metric | Method | Noise Intensity | | | | | | |
| --- | --- | --- | --- | --- | --- | --- | --- | --- | --- |
| | | | Clean | S1 | S2 | S3 | S4 | S5 | S6 |
| Gaussian Noise | FID↓ | Baseline | **31.49** | 53.33 | 69.88 | 108.26 | 124.72 | 228.02 | 404.01 |
| | | $+\mathcal{N}(0, 0.04)$ | 64.12 | **34.53** | **31.25** | **31.16** | **32.39** | **40.95** | **64.87** |
| | LPIPS↓ | Baseline | **0.3055** | 0.3991 | 0.4397 | 0.5123 | 0.5385 | 0.5994 | 0.6525 |
| | | $+\mathcal{N}(0, 0.04)$ | 0.3601 | **0.3186** | **0.3129** | **0.3119** | **0.3192** | **0.3411** | **0.4043** |
| Uniform Noise | FID↓ | Baseline | **31.49** | 53.14 | 70.36 | 112.59 | 133.71 | 250.89 | 431.58 |
| | | $+\mathcal{N}(0, 0.04)$ | 64.12 | **34.94** | **31.71** | **31.68** | **32.93** | **44.39** | **79.62** |
| | LPIPS↓ | Baseline | **0.3055** | 0.4003 | 0.4423 | 0.5182 | 0.5445 | 0.6098 | 0.6641 |
| | | $+\mathcal{N}(0, 0.04)$ | 0.3601 | **0.3185** | **0.3134** | **0.3161** | **0.3204** | **0.3488** | **0.4316** |
| Color Noise | FID↓ | Baseline | **31.49** | 64.45 | 97.99 | 176.49 | 226.99 | 400.63 | 448.24 |
| | | $+\mathcal{N}(0, 0.04)$ | 64.12 | **32.52** | **31.71** | **36.71** | **42.19** | **71.65** | **135.09** |
| | LPIPS↓ | Baseline | **0.3055** | 0.4176 | 0.4873 | 0.5726 | 0.5946 | 0.6519 | 0.6931 |
| | | $+\mathcal{N}(0, 0.04)$ | 0.3601 | **0.3194** | **0.3188** | **0.3227** | **0.3454** | **0.4125** | **0.5102** |
| Laplacian Noise | FID↓ | Baseline | **31.49** | 51.48 | 68.01 | 101.18 | 115.63 | 181.81 | 321.31 |
| | | $+\mathcal{N}(0, 0.04)$ | 64.12 | **34.93** | **31.37** | **31.11** | **32.18** | **38.26** | **51.61** |
| | LPIPS↓ | Baseline | **0.3055** | 0.3965 | 0.4337 | 0.5001 | 0.5248 | 0.5835 | 0.6341 |
| | | $+\mathcal{N}(0, 0.04)$ | 0.3601 | **0.3191** | **0.3134** | **0.3101** | **0.3169** | **0.3321** | **0.3702** |
| Salt & Pepper Noise | FID↓ | Baseline | **31.49** | 69.78 | 110.24 | 162.21 | 258.34 | 365.98 | 443.48 |
| | | $+\mathcal{N}(0, 0.04)$ | 64.12 | **31.99** | **32.45** | **37.42** | **46.89** | **58.03** | **74.25** |
| | LPIPS↓ | Baseline | **0.3055** | 0.4351 | 0.5201 | 0.5777 | 0.6199 | 0.6481 | 0.6671 |
| | | $+\mathcal{N}(0, 0.04)$ | 0.3601 | **0.3149** | **0.3187** | **0.3322** | **0.3576** | **0.3905** | **0.4303** |

**Ablation study of training noise intensity**: We conduct an ablation study of $\sigma_t^2$ for the photo-to-sketch translation task. Fig. 4 shows the comparison of visual qualities. Fig. 5 further illustrates the FID metrics for generated images by varying $\sigma_t^2$ with different types of i.i.d. noise at the inference stage. Here, the setting of "Learnable $\sigma_t^2$" treats $\sigma_t^2$ as a tuned hyperparameter. We observe that $\sigma_t^2$ controls the balance between robustness and quality - low values favor cleaner inputs, while high values prioritize noisy cases. This highlights the need to select an appropriate $\sigma_t^2$ balancing performance across expected conditions. With a maximum $\sigma_e^2 = 0.16$ in these simulations,

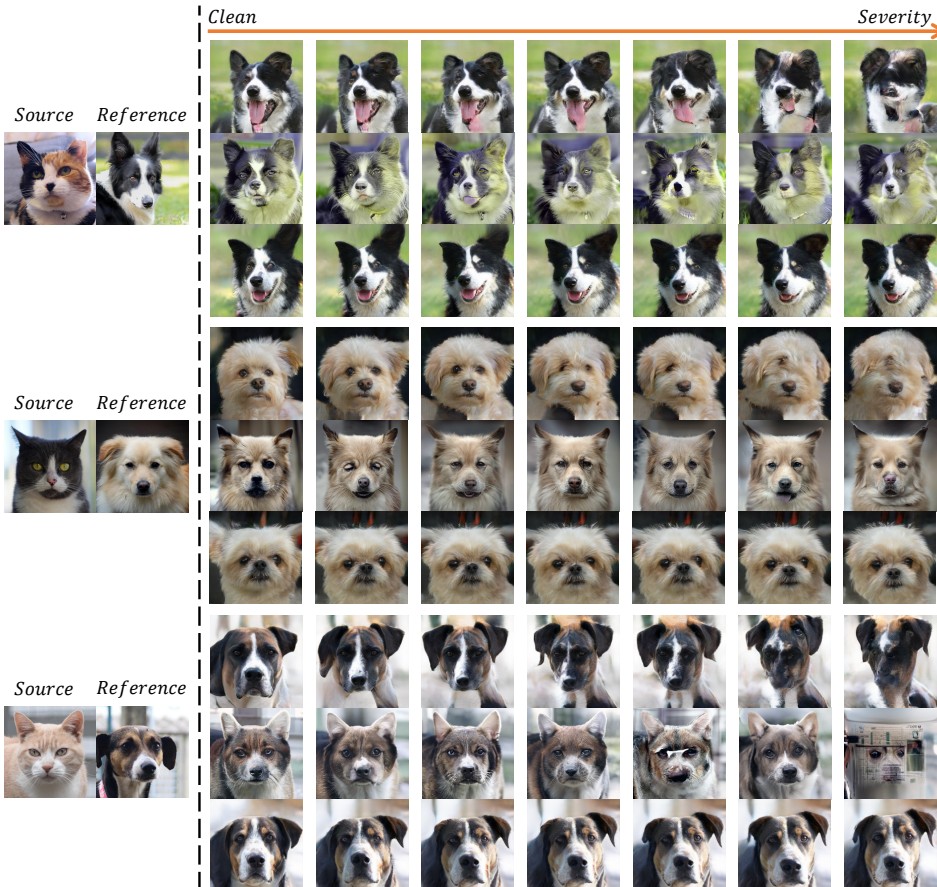

Figure 3: Noise injection method compared to DiffuseIT (Kwon & Ye, 2022) generation on the Cat→Dog task. The baseline model is GP-UNIT, and the noise environment is Color noise. In each pair of source and reference image comparisons, the first row is the result produced by the baseline model, the second row is produced by DiffuseIT, and the third is produced by the noise injection.

Corollary 1 states the optimal $\sigma_t^2$ minimizing average KL-divergence is $0.16/2 = 0.08$. Numerical FID results in Table 7 (Appendix D) confirm that $\sigma_t^2 = 0.08$ yields the smallest average FID.

It is interesting to note that although our analysis uses the KL-divergence, for supervised Photo→Sketch, the FID scores in Fig. 5 exhibit (near-) convexity in $\sigma_e^2$, which follows a very similar pattern as theoretical KL-divergence in Fig. 6(a). Besides, for i.i.d. noise, Part 2 of Theorem 2 indicates noise-trained systems can outperform clean-trained ones on noisy inputs given $\sigma_e^2 > 0.5\sigma_t^2$. FID scores in Fig. 5 agree with this theory. This empirically validates the value of our theoretical analysis for anticipating how models respond to unseen noises.

**Additional Results:** Due to space constraints, we provide more results in Appendix D, which includes theoretical results of Gaussian mixture model (GMM) signal sources, further simulation results contrasting empirical and theoretical findings, results on various types of image corruptions, out-of-domain tests, and comparisons between noise injection training and pre-denoising approaches. We also discuss theoretical and implementation limitations, highlighting certain failure scenarios.

## 5 CONCLUSION

In this paper, we investigated the challenge of noise resilience in GAN-based I2I translation models, focusing on the impact of injecting isotropic Gaussian noise into source domain images during training. By establishing a novel connection between $f$-divergence and score matching, we illuminated how Gaussian noise influences the alignment of probability distributions. We then demonstrated that for

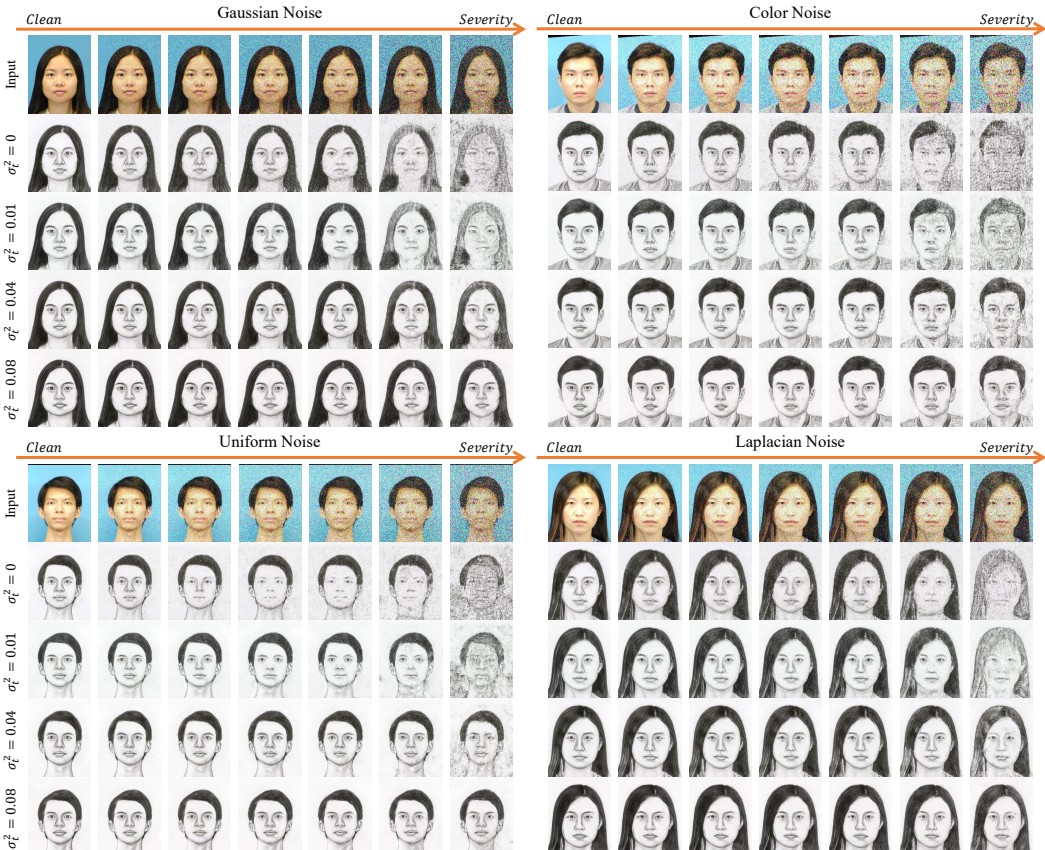

Figure 4: Photo→Sketch: Input photos with 4 noise types, each at 6 levels from clean to high severity, and their corresponding sketch outputs.

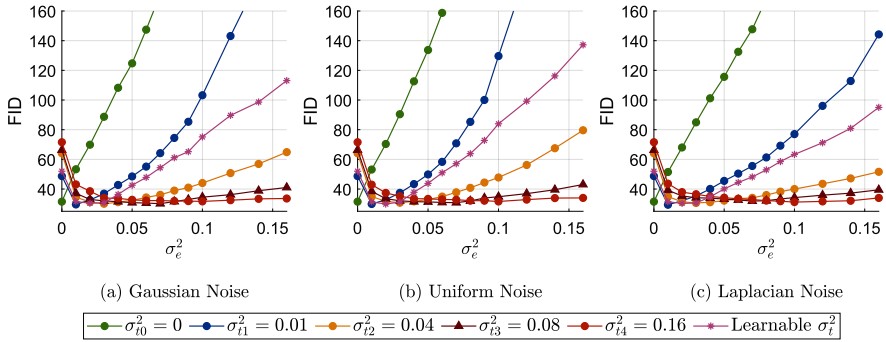

(a) Gaussian Noise    (b) Uniform Noise    (c) Laplacian Noise

$\sigma_{t0}^2 = 0$    $\sigma_{t1}^2 = 0.01$    $\sigma_{t2}^2 = 0.04$    $\sigma_{t3}^2 = 0.08$    $\sigma_{t4}^2 = 0.16$    Learnable $\sigma_t^2$

Figure 5: FID score comparison on noisy input images for models trained with different Gaussian noise levels. Test Noise Types: (a) Gaussian (b) Uniform, (c) Laplacian.

arbitrary signal sources, the robustness of I2I systems to Gaussian noise implies resilience to other noise types with a matched covariance matrix. Additionally, we addressed the selection of an optimal training noise variance. Extensive experimentation has validated our insights, showing a substantial performance improvement across diverse I2I translation tasks in noisy environments. Our results highlight the usefulness and efficiency of Gaussian noise injection for enhancing model robustness. This work offers valuable perspective into leveraging noise for more resilient I2I systems. Overall, it represents an important step towards reliable I2I translation in real-world noisy environments through a rigorous theoretical grounding.

ACKNOWLEDGEMENTS

We thank Dr. Cong Ling from Imperial College London and Dr. Yan Zhang from Chongqing University of Posts and Telecommunications for their invaluable assistance, support, and insightful feedback on this paper. We want to thank the three anonymous reviewers who spent a lot of time and effort and raised constructive questions and suggestions, which immensely helped us improve the theory and experiments of the paper. We would also like to thank the Program Chairs and Area Chairs for their approval and processing of this paper and for providing valuable and comprehensive comments.

The theoretical framework for this work was primarily established during Chaohua Shi and Kexin Huang's undergraduate research within the CQUPT-Brunel University London joint program. Meanwhile, this work was supported in part by the National Natural Science Foundation of China under Grants U22A2096, 62036007 and 62106184; in part by the Fundamental Research Funds for the Central Universities under Grants QTZX23042 and YJSJ24011; in part by the Young Talent Fund of Association for Science and Technology in Shaanxi China under Grant 20230121; in part by the Youth Innovation Team of Shaanxi Universities; in part by the Technology Innovation Leading Program of Shaanxi under Grant 2022QFY01-15 and in part by the Innovation Fund of Xidian University.

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

## A    NOTATIONS AND DEFINITIONS

**Notations**: In this paper, capital letters indicate random variables or vectors, while lowercase letters represent their realisations. For a random vector $\boldsymbol{X}$, $h(\boldsymbol{X})$ and $\boldsymbol{J}(\boldsymbol{X})$ denote its differential entropy and Fisher information matrix, respectively. For two random vectors $\boldsymbol{X}$ and $\boldsymbol{Y}$, $I(\boldsymbol{X};\boldsymbol{Y})$ corresponds to their mutual information. The notation $\mathcal{N}(\boldsymbol{\mu}, \boldsymbol{\Sigma})$ represents a multidimensional normal (Gaussian) distribution with mean $\boldsymbol{\mu}$ and covariance matrix $\boldsymbol{\Sigma}$. For a matrix $\boldsymbol{A}$, $\mathrm{Tr}(\boldsymbol{A})$ and $|\boldsymbol{A}|$ denote its trace and determinant, respectively. For a symmetric positive-definite matrix $\boldsymbol{A}$, $\lambda_i(\boldsymbol{A})$ represents the $i$-th largest eigenvalue of $\boldsymbol{A}$. For two real-valued symmetric matrices $\boldsymbol{A}$ and $\boldsymbol{B}$, the notation $\boldsymbol{A} > \boldsymbol{B}$ (or $\boldsymbol{A} < \boldsymbol{B}$) indicates that $\boldsymbol{A} - \boldsymbol{B}$ (or $\boldsymbol{B} - \boldsymbol{A}$) is positive definite.

**Definition of $f$-divergence**: The $f$-divergence belongs to a class of statistical metrics designed to quantify the discrepancies between two probability distributions. Let $P$ and $Q$ be distributions on a measurable space $\mathcal{X}$ with density functions $p$ and $q$, respectively. If $P \ll Q$, the $f$-divergence between these absolutely continuous distributions is defined as (Polyanskiy, 2019):

$$D_f(P\|Q) = \int_{\mathcal{X}} q(x) f\left(\frac{p(x)}{q(x)}\right) \mathrm{d}x, \tag{13}$$

In this formula, $f$ is a strictly convex, continuous function satisfying $f(1) = 0$, and is referred to as the generator function. Popular $f$-divergences and their corresponding generator functions are outlined in Table 3.

Table 3: List of $f$-divergences $D_f(P\|Q)$ and corresponding generator functions (Nielsen & Nock, 2013).

| Name | $D_f(P\|Q)$ | Generator $f(t)$ |
|---|---|---|
| Kullback-Leibler | $D_{KL} = \int p(x) \log \frac{p(x)}{q(x)} \, \mathrm{d}x$ | $t \log t$ |
| Neyman $\chi^2$ | $D_{\chi^2} = \int \frac{(q(x)-p(x))^2}{q(x)} \, \mathrm{d}x$ | $\frac{(1-t)^2}{t}$ |
| Total Variation | $D_{TV} = \frac{1}{2} \int |p(x) - q(x)| \, \mathrm{d}x$ | $\frac{1}{2}|t - 1|$ |
| Squared Hellinger | $D_{H^2} = \int \left(\sqrt{p(x)} - \sqrt{q(x)}\right)^2 \, \mathrm{d}x$ | $\left(\sqrt{t} - 1\right)^2$ |
| Jensen-Shannon | $D_{JSD} = D_{KL}\left(P \left\| \frac{P+Q}{2}\right.\right) + D_{KL}\left(Q \left\| \frac{P+Q}{2}\right.\right)$ | $-(t+1)\log\frac{1+t}{2} + t\log t$ |

**Definition of small $o$:** Let $\xi(\sigma_e^2)$ be a function of $\sigma_e^2$. We say $\xi(\sigma_e^2)$ is $o(\sigma_e^2)$ as $\sigma_e^2$ approaches 0 if and only if: $\lim_{\sigma_e^2 \to 0} \frac{\xi(\sigma_e^2)}{\sigma_e^2} = 0$. This notation means that $\xi(\sigma_e^2)$ becomes insignificant relative to $\sigma_e^2$ as $\sigma_e^2$ tends towards 0.

## B    THEORETICAL PROOFS

### B.1    PROOF OF THEOREM 1

The theorem's proof leverages the heat equation and Green's identities in vector calculus, which is similar to the proof of entropy power inequality in (Costa & Cover, 1984; Costa, 1985).

*Proof.* Note that Eq. (2) can be expanded as

$$
\begin{aligned}
&\frac{d}{d\sigma^2} D_f(\bar{P}\|\bar{Q}) \\
&= \frac{d}{d\sigma^2} \int_{-\infty}^{+\infty} \int_{-\infty}^{+\infty} \bar{q}(\bar{\boldsymbol{x}}, \boldsymbol{y}) \cdot f\left(\frac{\bar{p}(\bar{\boldsymbol{x}}, \boldsymbol{y})}{\bar{q}(\bar{\boldsymbol{x}}, \boldsymbol{y})}\right) \mathrm{d}\bar{\boldsymbol{x}} \mathrm{d}\boldsymbol{y} \\
&= \int_{-\infty}^{+\infty} \int_{-\infty}^{+\infty} \frac{\partial \bar{q}(\bar{\boldsymbol{x}}, \boldsymbol{y})}{\partial \sigma^2} f\left(\frac{\bar{p}(\bar{x}, \boldsymbol{y})}{\bar{q}(\bar{x}, \boldsymbol{y})}\right) + \bar{q}(\bar{\boldsymbol{x}}, \boldsymbol{y}) \frac{\partial}{\partial \sigma^2} f\left(\frac{\bar{p}(\bar{\boldsymbol{x}}, \boldsymbol{y})}{\bar{q}(\bar{\boldsymbol{x}}, \boldsymbol{y})}\right) \mathrm{d}\bar{\boldsymbol{x}} \mathrm{d}\boldsymbol{y} \\
&= \int_{-\infty}^{+\infty} \int_{-\infty}^{+\infty} \frac{\partial \bar{q}(\bar{\boldsymbol{x}}, \boldsymbol{y})}{\partial \sigma^2} \left[ f\left(\frac{\bar{p}(\bar{\boldsymbol{x}}, \boldsymbol{y})}{\bar{q}(\bar{\boldsymbol{x}}, \boldsymbol{y})}\right) - \frac{\bar{p}(\bar{\boldsymbol{x}}, \boldsymbol{y})}{\bar{q}(\bar{\boldsymbol{x}}, \boldsymbol{y})} \cdot f'\left(\frac{\bar{p}(\bar{\boldsymbol{x}}, \boldsymbol{y})}{\bar{q}(\bar{\boldsymbol{x}}, \boldsymbol{y})}\right) \right] + \frac{\partial \bar{p}(\bar{\boldsymbol{x}}, \boldsymbol{y})}{\partial \sigma^2} \cdot f'\left(\frac{\bar{p}(\bar{\boldsymbol{x}}, \boldsymbol{y})}{\bar{q}(\bar{\boldsymbol{x}}, \boldsymbol{y})}\right) \mathrm{d}\bar{\boldsymbol{x}} \mathrm{d}\boldsymbol{y}.
\end{aligned}
\tag{14}
$$

Using heat equation in diffusion, we know that

$$\frac{\partial \bar{p}(\bar{\boldsymbol{x}}, \boldsymbol{y})}{\partial \sigma^2} = \frac{1}{2} \nabla_{\bar{\boldsymbol{x}}} \cdot \nabla_{\bar{\boldsymbol{x}}} \bar{p}(\bar{\boldsymbol{x}}, \boldsymbol{y}) \quad \text{and} \quad \frac{\partial \bar{q}(\bar{\boldsymbol{x}}, \boldsymbol{y})}{\partial \sigma^2} = \frac{1}{2} \nabla_{\bar{\boldsymbol{x}}} \cdot \nabla_{\bar{\boldsymbol{x}}} \bar{q}(\bar{\boldsymbol{x}}, \boldsymbol{y}),$$

where $\nabla_{\bar{\boldsymbol{x}}}$ is the gradient operator with regard to $\bar{\boldsymbol{x}}$ and $\nabla_{\bar{\boldsymbol{x}}} \cdot \nabla_{\bar{\boldsymbol{x}}} = \sum_{i=1}^{d} \frac{\partial^2}{\partial \bar{\boldsymbol{x}}_i^2}$. This leads to

$$
\begin{aligned}
&\frac{d}{d\sigma^2} D_f(\bar{P} \| \bar{Q}) \\
&= \frac{1}{2} \int_{-\infty}^{+\infty} \int_{-\infty}^{+\infty} \nabla_{\bar{\boldsymbol{x}}} \cdot \nabla_{\bar{\boldsymbol{x}}} \bar{p}(\bar{\boldsymbol{x}}, \boldsymbol{y}) \cdot f'\left(\frac{\bar{p}(\bar{\boldsymbol{x}}, \boldsymbol{y})}{\bar{q}(\bar{\boldsymbol{x}}, \boldsymbol{y})}\right) \mathrm{d}\bar{\boldsymbol{x}}\mathrm{d}\boldsymbol{y} \\
&+ \frac{1}{2} \int_{-\infty}^{+\infty} \int_{-\infty}^{+\infty} \nabla_{\bar{\boldsymbol{x}}} \cdot \nabla_{\bar{\boldsymbol{x}}} \bar{q}(\bar{\boldsymbol{x}}, \boldsymbol{y}) \left[ f\left(\frac{\bar{p}(\bar{\boldsymbol{x}}, \boldsymbol{y})}{\bar{q}(\bar{\boldsymbol{x}}, \boldsymbol{y})}\right) - \frac{\bar{p}(\bar{\boldsymbol{x}}, \boldsymbol{y})}{\bar{q}(\bar{\boldsymbol{x}}, \boldsymbol{y})} \cdot f'\left(\frac{\bar{p}(\bar{\boldsymbol{x}}, \boldsymbol{y})}{\bar{q}(\bar{\boldsymbol{x}}, y)}\right) \right] \mathrm{d}\bar{\boldsymbol{x}}\mathrm{d}\boldsymbol{y}
\end{aligned}
\tag{15}
$$

For simplicity of presentation, we set $\bar{p} = \bar{p}(\bar{\boldsymbol{x}}, \boldsymbol{y}), \bar{q} = \bar{q}(\bar{\boldsymbol{x}}, \boldsymbol{y})$. As $g'h = (gh)' - gh'$, the above can be simplified as (Guo, 2009)

$$
\begin{aligned}
&\nabla_{\bar{\boldsymbol{x}}} \cdot \nabla_{\bar{\boldsymbol{x}}} \bar{p} \cdot \left[ f'\left(\frac{\bar{p}}{\bar{q}}\right) \right] + \nabla_{\bar{\boldsymbol{x}}} \cdot \nabla_{\bar{\boldsymbol{x}}} \bar{q} \cdot \left[ f\left(\frac{\bar{p}}{\bar{q}}\right) - \frac{\bar{p}}{\bar{q}} f'\left(\frac{\bar{p}}{\bar{q}}\right) \right] \\
&= \nabla_{\bar{\boldsymbol{x}}} \cdot \nabla_{\bar{\boldsymbol{x}}} \left[ \bar{q} f\left(\frac{\bar{p}}{\bar{q}}\right) \right] - \nabla_{\bar{\boldsymbol{x}}} \bar{p} \cdot f''\left(\frac{\bar{p}}{\bar{q}}\right) + \frac{\bar{p}}{\bar{q}} \nabla_{\bar{\boldsymbol{x}}} \bar{q} \cdot f''\left(\frac{\bar{p}}{\bar{q}}\right).
\end{aligned}
\tag{16}
$$

For the first term, it can be simplified by Green's formula:

$$
\int_{-\infty}^{+\infty} \int_{-\infty}^{+\infty} \nabla_{\bar{\boldsymbol{x}}} \cdot \nabla_{\bar{\boldsymbol{x}}} \left[ \bar{q} f\left(\frac{\bar{p}}{\bar{q}}\right) \right] \mathrm{d}\bar{\boldsymbol{x}}\mathrm{d}\boldsymbol{y} = \int_S \nabla_{\bar{\boldsymbol{x}}} \cdot \bar{q} f\left(\frac{\bar{p}}{\bar{q}}\right) \cdot \mathrm{d}S = \nabla_{\bar{\boldsymbol{x}}} \cdot \int_S \bar{q} f\left(\frac{\bar{p}}{\bar{q}}\right) \cdot \mathrm{d}S = 0, \tag{17}
$$

where $S$ is a piece-wise smooth, closed, oriented surface in $\mathbb{R}^d$. Since $D_f()$ is finite, the above result can be obtained using a similar argument as Eq. (B.15) in (Costa, 1985).

The last two terms can be reduced to:

$$
-\nabla_{\bar{\boldsymbol{x}}} \bar{p} \cdot f''\left(\frac{\bar{p}}{\bar{q}}\right) + \frac{\bar{p}}{\bar{q}} \nabla_{\bar{\boldsymbol{x}}} \bar{q} \cdot f''\left(\frac{\bar{p}}{\bar{q}}\right) = -\bar{q} \nabla_{\bar{\boldsymbol{x}}} \left(\frac{\bar{p}}{\bar{q}}\right) \cdot f''\left(\frac{\bar{p}}{\bar{q}}\right) = -\bar{q} f''\left(\frac{\bar{p}}{\bar{q}}\right) \cdot \left(\nabla_{\bar{\boldsymbol{x}}} \left(\frac{\bar{p}}{\bar{q}}\right)\right)^2. \tag{18}
$$

Organizing the previous derivations, the final result can be written in the following form:

$$
\begin{aligned}
\frac{d}{d\sigma^2} D_f(\bar{P} \| \bar{Q}) &= -\frac{1}{2} \mathbb{E}_{\bar{Q}} \left\{ f''\left(\frac{\bar{p}(\bar{\boldsymbol{x}}, \boldsymbol{y})}{\bar{q}(\bar{\boldsymbol{x}}, \boldsymbol{y})}\right) \left\| \nabla_{\bar{\boldsymbol{x}}} \frac{\bar{p}(\bar{\boldsymbol{x}}, \boldsymbol{y})}{\bar{q}(\bar{\boldsymbol{x}}, \boldsymbol{y})} \right\|^2 \right\} \\
&= -\frac{1}{2} \mathbb{E}_{\bar{P}} \left\{ \frac{\bar{p}(\bar{\boldsymbol{x}}, \boldsymbol{y})}{\bar{q}(\bar{\boldsymbol{x}}, \boldsymbol{y})} f''\left(\frac{\bar{p}(\bar{\boldsymbol{x}}, \boldsymbol{y})}{\bar{q}(\bar{\boldsymbol{x}}, \boldsymbol{y})}\right) \| \nabla_{\bar{\boldsymbol{x}}} \log \bar{p}(\bar{\boldsymbol{x}}, \boldsymbol{y}) - \nabla_{\bar{\boldsymbol{x}}} \log \bar{q}(\bar{\boldsymbol{x}}, \boldsymbol{y}) \|^2 \right\},
\end{aligned}
\tag{19}
$$

which completes the proof. $\qquad\square$

## B.2 Proof of Lemma 1

*Proof.* As shown in Table KL-divergence is the special case of $f$-divergence with $f(t) = t \log t$ in Eq. (13)

$$
D_{KL}(P \| Q) = \int_{\mathcal{X}} p(x) \ln\left(\frac{p(x)}{q(x)}\right) \mathrm{d}x. \tag{20}
$$

Assume that $t(\hat{\boldsymbol{x}})$ is the probability density function of $P_{\hat{\boldsymbol{X}}}$, the KL-divergence $D_{KL}(P_{\hat{\boldsymbol{X}}} \| P_{\bar{\boldsymbol{X}}}) = D_{KL}(P_{\hat{\boldsymbol{X}}} \| \mathcal{N}(\boldsymbol{\mu}_s, \boldsymbol{\Sigma}_s + \sigma_t^2 \boldsymbol{I}_d))$ can be expanded as

$$
\begin{aligned}
&D_{KL}(P_{\hat{\boldsymbol{X}}} \| \mathcal{N}(\boldsymbol{\mu}_s, \boldsymbol{\Sigma}_s + \sigma_t^2 \boldsymbol{I}_d)) \\
&= -h(\hat{\boldsymbol{X}}) - \int t(\hat{\boldsymbol{x}}) \log\left((2\pi)^{-d/2} \cdot \det(\boldsymbol{\Sigma_2})^{-1/2} \cdot \exp\left[-\frac{1}{2}(\hat{\boldsymbol{x}} - \boldsymbol{\mu}_s)^\top \boldsymbol{\Sigma}_2^{-1}(\hat{\boldsymbol{x}} - \boldsymbol{\mu}_s)\right]\right) \mathrm{d}\hat{\boldsymbol{x}} \\
&= -h(\hat{\boldsymbol{X}}) + \frac{d}{2}\log(2\pi) + \frac{1}{2}\log|\boldsymbol{\Sigma}_2| - \int t(\hat{\boldsymbol{x}})\left(-\frac{1}{2}(\hat{\boldsymbol{x}} - \boldsymbol{\mu}_s)^\top \boldsymbol{\Sigma}_2^{-1}(\hat{\boldsymbol{x}} - \boldsymbol{\mu}_s)\right) \mathrm{d}\hat{\boldsymbol{x}},
\end{aligned}
\tag{21}
$$

in which $h(\hat{\boldsymbol{x}})$ is the differential entropy of $\hat{\boldsymbol{x}}$. The last term can be further simplified as

$$
\begin{aligned}
-\int t(\hat{\boldsymbol{x}}) \left(-\frac{1}{2} (\hat{\boldsymbol{x}} - \boldsymbol{\mu}_s)^\top \boldsymbol{\Sigma}_2^{-1} (\hat{\boldsymbol{x}} - \boldsymbol{\mu}_s)\right) \mathrm{d}\hat{\boldsymbol{x}} &= \frac{1}{2} E\left[\operatorname{Tr}\left((\hat{\boldsymbol{x}} - \boldsymbol{\mu}_s)^\top \boldsymbol{\Sigma}_2^{-1} (\hat{\boldsymbol{x}} - \boldsymbol{\mu}_s)\right)\right] \\
&= \frac{1}{2} \operatorname{Tr}\left[\boldsymbol{\Sigma}_2^{-1} \cdot E\left((\hat{\boldsymbol{x}} - \boldsymbol{\mu}_s)^\top (\hat{\boldsymbol{x}} - \boldsymbol{\mu}_s)\right)\right] \\
&= \frac{1}{2} \operatorname{Tr}\left[\boldsymbol{\Sigma}_2^{-1} \cdot \boldsymbol{\Sigma}_1\right],
\end{aligned}
\tag{22}
$$

where $\boldsymbol{\Sigma}_1 = E\left[(\hat{\boldsymbol{x}} - \boldsymbol{\mu}_s)^\top (\hat{\boldsymbol{x}} - \boldsymbol{\mu}_s)\right]$. Hence, Eq. (6) is proved. □

### B.3 PROOF OF THEOREM 2

*Proof.* Part 1 of the proof is based on the concave property of mutual information derived in (Payaró & Palomar, 2009), while Part 2 is on matrices trace and determinant properties.

- **Part 1**:
  - Proof of $\rho(\sigma_t^2, \sigma_e^2 \boldsymbol{\Sigma}_{\widetilde{e}})$'s convexity in $\sigma_e^2$:
    For $\boldsymbol{E} = \sigma_e \widetilde{\boldsymbol{E}}$, with covariance matrix $\boldsymbol{\Sigma}_e = \sigma_e^2 \boldsymbol{\Sigma}_{\widetilde{e}}$, Eq. (6) gives:

    $$
    \begin{aligned}
    \rho(\sigma_t^2, \sigma_e^2 \boldsymbol{\Sigma}_{\widetilde{e}}) = &-h(\boldsymbol{X} + \sigma_e \widetilde{\boldsymbol{E}}) + \frac{d}{2} \log(2\pi) + \frac{1}{2} \log|\boldsymbol{\Sigma}_2| \\
    &+ \frac{1}{2} \operatorname{Tr}\left(\boldsymbol{\Sigma}_2^{-1} \boldsymbol{\Sigma}_s\right) + \frac{\sigma_e^2}{2} \operatorname{Tr}\left(\boldsymbol{\Sigma}_2^{-1} \boldsymbol{\Sigma}_{\widetilde{e}}\right).
    \end{aligned}
    \tag{23}
    $$

    in which $\boldsymbol{\Sigma}_2 = \boldsymbol{\Sigma}_s + \sigma_t^2 \boldsymbol{I}_d$. The term $\frac{\sigma_e^2}{2} \operatorname{Tr}\left(\boldsymbol{\Sigma}_2^{-1} \boldsymbol{\Sigma}_{\widetilde{e}}\right)$ is evidently convex in $\sigma_e^2$. To demonstrate the convexity of $-h(\boldsymbol{X} + \sigma_e \widetilde{\boldsymbol{E}})$ in $\sigma_e^2$, given $\boldsymbol{X}$ is Gaussian and independent of $\widetilde{\boldsymbol{E}}$, we express it as:

    $$
    -h(\boldsymbol{X} + \sigma_e \widetilde{\boldsymbol{E}}) = -I(\boldsymbol{X} + \sigma_e \widetilde{\boldsymbol{E}}; \widetilde{\boldsymbol{E}}) + \log(2\pi) + \log|\boldsymbol{X}|,
    $$

    where $I(\boldsymbol{X} + \sigma_e \widetilde{\boldsymbol{E}}; \widetilde{\boldsymbol{E}})$ denotes the mutual information between $\boldsymbol{X} + \sigma_e \widetilde{\boldsymbol{E}}$ and $\widetilde{\boldsymbol{E}}$. Given $\boldsymbol{X}$ is Gaussian, it was previously established in Corollary 1 of (Payaró & Palomar, 2009) that the mutual information $I(\boldsymbol{X} + \sigma_e \widetilde{\boldsymbol{E}}; \widetilde{\boldsymbol{E}})$ is concave with respect to $\sigma_e^2$. This implies that $-I(\boldsymbol{X} + \sigma_e \widetilde{\boldsymbol{E}}; \widetilde{\boldsymbol{E}})$ and therefore, $-h(\boldsymbol{X} + \sigma_e \widetilde{\boldsymbol{E}})$ is convex in $\sigma_e^2$.
    **Remark**: The work in (Payaró & Palomar, 2009) assumes Gaussian noise with an arbitrary signal. In contrast, our analysis considers a Gaussian signal with arbitrary noise. As a result, in our context, $\sigma_e^2$ serves the role analogous to the SNR in Corollary 1 of (Payaró & Palomar, 2009).
  - Proof of Eq. (8):
    It can be derived that

    $$
    \begin{aligned}
    \rho(\sigma_t^2, \boldsymbol{\Sigma}_e) - \rho_g(\sigma_t^2, \boldsymbol{\Sigma}_e) &= h(\boldsymbol{X} + \boldsymbol{E}_g) - h(\boldsymbol{X} + \boldsymbol{E}) \\
    &= h(\boldsymbol{X} + \sigma_e \widetilde{\boldsymbol{E}}_g) - h(\boldsymbol{X} + \sigma_e \widetilde{\boldsymbol{E}})
    \end{aligned}
    \tag{24}
    $$

    According to the generalized De Bruijn's Identity (Proposition 7 in (Rioul, 2010))

    $$
    \left. \frac{d\, h(\boldsymbol{X} + \sigma_e \widetilde{\boldsymbol{E}})}{d\sigma_e^2} \right|_{\sigma_e = 0} = \frac{1}{2} \operatorname{Tr}\left(\boldsymbol{J}(\boldsymbol{X}) \boldsymbol{\Sigma}_{\widetilde{e}}\right),
    \tag{25}
    $$

    in which $\boldsymbol{J}(\boldsymbol{X})$ is the Fisher information matrix of $\boldsymbol{X}$. Hence, for small $\sigma_e$, using first-order Taylor series expansion as in Eq. (37) of (Rioul, 2010), one can obtain

    $$
    h(\boldsymbol{X} + \sigma_e \widetilde{\boldsymbol{E}}) = h(\boldsymbol{X}) + \frac{\sigma_e^2}{2} \operatorname{Tr}\left(\boldsymbol{J}(\boldsymbol{X}) \boldsymbol{\Sigma}_{\widetilde{e}}\right) + o(\sigma_e^2).
    \tag{26}
    $$

    and similarly,

    $$
    h(\boldsymbol{X} + \sigma_e \widetilde{\boldsymbol{E}}_g) = h(\boldsymbol{X}) + \frac{\sigma_e^2}{2} \operatorname{Tr}\left(\boldsymbol{J}(\boldsymbol{X}) \boldsymbol{\Sigma}_{\widetilde{e}_g}\right) + o(\sigma_e^2).
    \tag{27}
    $$

    When $\boldsymbol{\Sigma}_{\widetilde{e}} = \boldsymbol{\Sigma}_{\widetilde{e}_g}$, we have $h(\boldsymbol{X} + \sigma_e \widetilde{\boldsymbol{E}}_g) - h(\boldsymbol{X} + \sigma_e \widetilde{\boldsymbol{E}}) = o(\sigma_e^2)$. Combined with Eq. (24), Eq. (8) is proved.

- **Part 2**: Given Eq. (6), the difference $\rho_g(0, \boldsymbol{\Sigma}_e) - \rho(\sigma_t^2, \boldsymbol{\Sigma}_e)$ can be expressed as:

$$
\begin{aligned}
&\rho(0, \boldsymbol{\Sigma}_e) - \rho_g(\sigma_t^2, \boldsymbol{\Sigma}_e) \\
=&\frac{1}{2} \log \left| \boldsymbol{\Sigma}_s + \sigma_t^2 \boldsymbol{I}_d \right| - \frac{1}{2} \log |\boldsymbol{\Sigma}_s| \\
&+ \frac{1}{2} \operatorname{Tr}\big( \boldsymbol{\Sigma}_s^{-1} \left( \boldsymbol{\Sigma}_s + \boldsymbol{\Sigma}_e \right) \big) - \frac{1}{2} \operatorname{Tr}\big( \left( \boldsymbol{\Sigma}_s + \sigma_t^2 \boldsymbol{I}_d \right)^{-1} \left( \boldsymbol{\Sigma}_s + \boldsymbol{\Sigma}_e \right) \big).
\end{aligned}
\tag{28}
$$

The proof of $\rho(0, \boldsymbol{\Sigma}_e) \geq \rho(\sigma_t^2, \boldsymbol{\Sigma}_e)$ when $\boldsymbol{\Sigma}_e \geq \frac{1}{2}\sigma_t^2 \boldsymbol{I}_d$ is equivalent to showing

$$
\operatorname{Tr}(\boldsymbol{\Sigma}_s^{-1}(\boldsymbol{\Sigma}_s + \boldsymbol{\Sigma}_e)) - \operatorname{Tr}((\boldsymbol{\Sigma}_s + \sigma_t^2 \boldsymbol{I}_d)^{-1}(\boldsymbol{\Sigma}_s + \boldsymbol{\Sigma}_e)) > \log \frac{\left| \boldsymbol{\Sigma}_s + \sigma_t^2 \boldsymbol{I}_d \right|}{|\boldsymbol{\Sigma}_s|}.
\tag{29}
$$

Bounding the left-hand side (LHS) of Eq. (29), we have:

$$
\begin{aligned}
&\operatorname{Tr}(\boldsymbol{\Sigma}_s^{-1}(\boldsymbol{\Sigma}_s + \boldsymbol{\Sigma}_e)) - \operatorname{Tr}((\boldsymbol{\Sigma}_s + \sigma_t^2 \boldsymbol{I}_d)^{-1}(\boldsymbol{\Sigma}_s + \sigma_t^2 \boldsymbol{I}_d - \sigma_t^2 \boldsymbol{I}_d + \boldsymbol{\Sigma}_e)) \\
=&\operatorname{Tr}(\boldsymbol{\Sigma}_s^{-1} \boldsymbol{\Sigma}_e) - \operatorname{Tr}((\boldsymbol{\Sigma}_s + \sigma_t^2 \boldsymbol{I}_d)^{-1} \boldsymbol{\Sigma}_e) + \sigma_t^2 \operatorname{Tr}((\boldsymbol{\Sigma}_s + \sigma_t^2 \boldsymbol{I}_d)^{-1}) \\
=&\operatorname{Tr}\Big( \big( \boldsymbol{\Sigma}_s^{-1} - (\boldsymbol{\Sigma}_s + \sigma_t^2 \boldsymbol{I}_d)^{-1} \big) \cdot \boldsymbol{\Sigma}_e \Big) + \sigma_t^2 \operatorname{Tr}\big( (\boldsymbol{\Sigma}_s + \sigma_t^2 \boldsymbol{I}_d)^{-1} \big) \\
\overset{(a)}{>}&\operatorname{Tr}\Big( \big( \boldsymbol{\Sigma}_s^{-1} - (\boldsymbol{\Sigma}_s + \sigma_t^2 \boldsymbol{I}_d)^{-1} \big) \cdot \frac{\sigma_t^2}{2} \Big) + \sigma_t^2 \operatorname{Tr}\big( (\boldsymbol{\Sigma}_s + \sigma_t^2 \boldsymbol{I}_d)^{-1} \big) \\
=&\frac{\sigma_t^2}{2} \operatorname{Tr}(\boldsymbol{\Sigma}_s^{-1}) + \frac{\sigma_t^2}{2} \operatorname{Tr}\big( (\boldsymbol{\Sigma}_s + \sigma_t^2 \boldsymbol{I}_d)^{-1} \big) = \frac{1}{2} \sum_{i=1}^d \frac{\sigma_t^2}{\lambda_i(\boldsymbol{\Sigma}_s)} + \frac{1}{2} \sum_{i=1}^d \frac{\sigma_t^2}{\lambda_i(\boldsymbol{\Sigma}_s) + \sigma_t^2} \\
=&\frac{1}{2} \sum_{i=1}^d \left( \frac{\sigma_t^2}{\lambda_i(\boldsymbol{\Sigma}_s)} + 1 - \frac{1}{1 + \sigma_t^2/\lambda_i(\boldsymbol{\Sigma}_s)} \right),
\end{aligned}
\tag{30}
$$

where the inequality at $(a)$ arises from $\boldsymbol{\Sigma}_e \geq \frac{\sigma_t^2}{2} \boldsymbol{I}_d$ and $\boldsymbol{\Sigma}_s^{-1} > (\boldsymbol{\Sigma}_s + \sigma_t^2 \boldsymbol{I}_d)^{-1}$. The right-hand side of Eq. (29) can be written as

$$
\log \frac{\left| \boldsymbol{\Sigma}_s + \sigma_t^2 \boldsymbol{I}_d \right|}{|\boldsymbol{\Sigma}_s|} = \sum_{i=1}^d \log \left( 1 + \frac{\sigma_t^2}{\lambda_i(\boldsymbol{\Sigma}_s)} \right)
$$

Next, consider the following function

$$
\psi(x) = \frac{1}{2} \left( x + 1 - \frac{1}{1+x} \right) - \log(1+x).
$$

It is obvious that when $x = 0$, $\psi(0) = 0$ and $\frac{\mathrm{d}\,\psi(x)}{\mathrm{d}x} = \frac{x^2}{2(1+x)^2} > 0$ for $x > 0$. Thus, $\psi(x)$ is strictly increasing for positive $x$, implying that

$$
\frac{1}{2} \left( \frac{\sigma_t^2}{\lambda_i(\boldsymbol{\Sigma}_s)} + 1 - \frac{1}{1 + \sigma_t^2/\lambda_i(\boldsymbol{\Sigma}_s)} \right) > \log \left( 1 + \frac{\sigma_t^2}{\lambda_i(\boldsymbol{\Sigma}_s)} \right).
$$

This establishes Eq. (29) and completes the proof.

$\square$

**Example**: We evaluate an AR(1) signal model with $d = 256$ and its signal covariance matrix given by $\boldsymbol{\Sigma}_s(k, l) = \sigma_s^2 \rho^{|k-l|}$ (for $0 \leq k, l \leq d-1$). Parameters $\sigma_s^2 = 0.125$ and $\rho = 0.95$ are derived from the normalized CUFS dataset. Fig. 6 illustrates $\rho_g(\sigma_t^2, \sigma_e^2 \boldsymbol{\Sigma}_{\widetilde{e}})$ across two different $\boldsymbol{\Sigma}_{\widetilde{e}}$. In Fig. 6(a), i.i.d. noise is considered with $\boldsymbol{\Sigma}_{\widetilde{e}} = I_d$. Meanwhile, Fig. 6(b) shows non-i.i.d. noise with $\boldsymbol{\Sigma}_{\widetilde{e}} = \operatorname{diag}(1.6\boldsymbol{I}_{64}, 1.2\boldsymbol{I}_{64}, 0.8\boldsymbol{I}_{64}, 0.4\boldsymbol{I}_{64})$. All curves display convex behavior in $\sigma_e^2$, which agrees with Part 1 of Theorem 2. Besides, as evident in Fig. 6(a), $\rho_g(0, \sigma_e^2 \boldsymbol{I}_d) > \rho_g(\sigma_t^2, \sigma_e^2 \boldsymbol{I}_d)$ whenever $\sigma_e^2 > 0.5\sigma_t^2$, reaffirming Part 2 of Theorem 2.

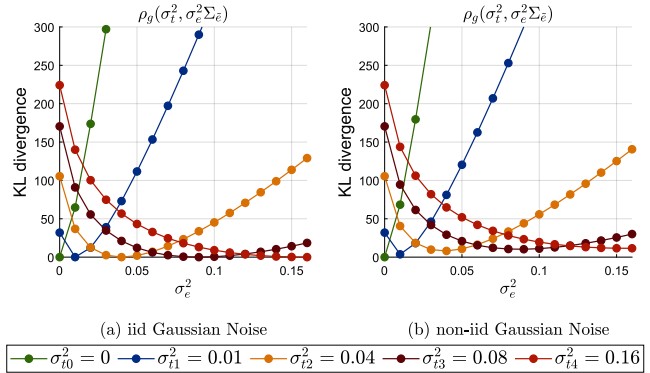

Figure 6: Visualization of $\rho_g(\sigma_t^2, \sigma_e^2\boldsymbol{\Sigma}_{\widetilde{e}})$ for AR(1) signal model with $d = 256$ and covariance matrix $\boldsymbol{\Sigma}_s(k, l) = 0.95\rho^{|k-l|}$ (for $0 \le k, l \le 255$). (a) i.i.d. noise with $\boldsymbol{\Sigma}_{\widetilde{e}} = \boldsymbol{I}_d$. (b) Non-i.i.d. noise with $\boldsymbol{\Sigma}_{\widetilde{e}} = \mathrm{diag}(1.6\boldsymbol{I}_{64}, 1.2\boldsymbol{I}_{64}, 0.8\boldsymbol{I}_{64}, 0.4\boldsymbol{I}_{64})$.

### B.4   PROOF OF THEOREM 3

*Proof.* The previous proof in Theorem 2 is based on the assumption that the source signal is Gaussian distributed. Here we further consider the non-Gaussian signal with arbitrary noise.

- **Part 1**

   Expanding the KL-divergence,

$$\theta(\sigma_t^2, \sigma_e^2\boldsymbol{\Sigma}_{\widetilde{e}}) = D_{KL}\left(\hat{P}_{\boldsymbol{X}+\sigma_e\widetilde{\boldsymbol{E}}}\|\bar{P}_{\boldsymbol{X}+\boldsymbol{\sigma}_t\boldsymbol{N}}\right)$$
$$= -h(\boldsymbol{X} + \sigma_e\widetilde{\boldsymbol{E}}) - \int_{-\infty}^{+\infty} \hat{p}(\boldsymbol{x} + \sigma_e\widetilde{\boldsymbol{E}}) \log \bar{p}(\boldsymbol{x} + \sigma_t\boldsymbol{N}) \, \mathrm{d}\boldsymbol{x}. \tag{31}$$

   According to the Lemma 1 in (Guo, 2009)

$$\frac{d}{d\sigma_e^2} \left.\hat{p}(\boldsymbol{x} + \sigma_e\widetilde{\boldsymbol{E}})\right|_{\sigma_e=0} = \frac{1}{2}\nabla_{\boldsymbol{x}}\nabla_{\boldsymbol{x}}p(\boldsymbol{x}),$$

   where $p(\boldsymbol{x})$ is the probability density function of the clean data, and thus

$$\frac{d}{d\sigma_e^2} \int_{-\infty}^{+\infty} \hat{p}(\boldsymbol{x} + \sigma_e\widetilde{\boldsymbol{E}}) \log \bar{p}(\boldsymbol{x} + \sigma_t\boldsymbol{N}) \, \mathrm{d}\boldsymbol{x}\bigg|_{\sigma_e=0} = \frac{1}{2}\nabla_{\boldsymbol{x}}p(\boldsymbol{x}) \log \bar{p}(\boldsymbol{x} + \sigma_t\boldsymbol{N}). \tag{32}$$

   The first-order Taylor series expansion of the integral part of Eq. (31) can be derived,

$$\int_{-\infty}^{+\infty} \hat{p}(\boldsymbol{x} + \sigma_e\widetilde{\boldsymbol{E}}) \log \bar{p}(\boldsymbol{x} + \sigma_t\boldsymbol{N}) \, \mathrm{d}\boldsymbol{x}$$
$$= \int_{-\infty}^{+\infty} p(\boldsymbol{x}) \log \bar{p}(\boldsymbol{x} + \sigma_t\boldsymbol{N}) \, \mathrm{d}\boldsymbol{x} + \frac{\sigma_e^2}{2}\nabla_{\boldsymbol{x}}p(\boldsymbol{x}) \log \bar{p}(\boldsymbol{x} + \sigma_t\boldsymbol{N}) + o(\sigma_e^2). \tag{33}$$

   By gathering Eq (26) and Eq. (33), the Taylor series expansion of the KL-divergence

$$\theta(\sigma_t^2, \sigma_e^2\boldsymbol{\Sigma}_{\widetilde{e}}) = -h(\boldsymbol{X}) - \frac{\sigma_e^2}{2}\mathrm{Tr}\left(\boldsymbol{J}(\boldsymbol{X})\boldsymbol{\Sigma}_{\widetilde{e}}\right) - \int_{-\infty}^{+\infty} p(\boldsymbol{x}) \log \bar{p}(\boldsymbol{x} + \sigma_t\boldsymbol{N}) \, \mathrm{d}\boldsymbol{x}$$
$$- \frac{\sigma_e^2}{2}\nabla_{\boldsymbol{x}}p(\boldsymbol{x}) \log \bar{p}(\boldsymbol{x} + \sigma_t\boldsymbol{N}) + o(\sigma_e^2). \tag{34}$$

   The KL divergence for Gaussian noise can be expanded according to the Taylor series expansion in Proposition 7 in (Rioul, 2010) as

$$\theta_g(\sigma_t^2, \sigma_e^2\boldsymbol{\Sigma}_{\widetilde{e}}) = D_{KL}\left(\hat{P}_{\boldsymbol{X}+\sigma_e\widetilde{\boldsymbol{E}}_g}\|\bar{P}_{\boldsymbol{X}+\boldsymbol{\sigma}_t\boldsymbol{N}}\right)$$
$$= \frac{\delta^2}{2}\boldsymbol{J}(\boldsymbol{X}) + o(\delta^2), \tag{35}$$

in which $\sigma_e = \sigma_t + \delta$. Obviously the $\theta(\sigma_t^2, \sigma_e^2 \boldsymbol{\Sigma}_{\widetilde{e}})$ only relates to the covariance matrix $\boldsymbol{\Sigma}_e$ of the arbitrary noise instead of the noise distribution. When $\boldsymbol{\Sigma}_{\widetilde{e}} = \boldsymbol{\Sigma}_{\widetilde{e}_g}$, one have $\theta(\sigma_t^2, \sigma_e^2 \boldsymbol{\Sigma}_{\widetilde{e}}) = \theta_g(\sigma_t^2, \sigma_e^2 \boldsymbol{\Sigma}_{\widetilde{e}}) + o(\sigma_e^2)$, and thus Eq. (9) is proved.

- **Part 2**:

Derive $\sigma_e^2$ of $\theta_g(\sigma_t^2, \sigma_e^2 \boldsymbol{I}_d)$,

$$
\begin{aligned}
& \frac{d}{d\sigma_t^2} \theta_g(\sigma_t^2, \sigma_e^2 \boldsymbol{I}_d) \\
& = -\frac{d}{d\sigma_t^2} h(\boldsymbol{X} + \sigma_e \boldsymbol{N}) - \frac{d}{d\sigma_t^2} \int_{-\infty}^{+\infty} \hat{p}(\hat{\boldsymbol{x}}) \log \bar{p}(\bar{\boldsymbol{x}}) \, \mathrm{d}\boldsymbol{x}.
\end{aligned}
\tag{36}
$$

Then we calculate derivatives one by one,

$$
\begin{aligned}
& \frac{d}{d\sigma_t^2} h(\boldsymbol{X} + \sigma_e \boldsymbol{N}) \\
& = -\int_{-\infty}^{+\infty} \frac{d}{d\sigma_t^2} \hat{p}(\hat{\boldsymbol{x}}) \, \mathrm{d}\hat{\boldsymbol{x}} - \int_{-\infty}^{+\infty} \left( \frac{d}{d\sigma_t^2} \hat{p}(\hat{\boldsymbol{x}}) \right) \log \hat{p}(\hat{\boldsymbol{x}}) \, \mathrm{d}\hat{\boldsymbol{x}} \\
& = 0 - \frac{1}{2} \int_{-\infty}^{+\infty} \left( \nabla_{\hat{\boldsymbol{x}}}^2 \hat{p}(\hat{\boldsymbol{x}}) \right) \log \hat{p}(\hat{\boldsymbol{x}}) \, \mathrm{d}\hat{\boldsymbol{x}},
\end{aligned}
\tag{37}
$$

in which $\frac{d}{d\sigma_t^2} \hat{p}(\hat{\boldsymbol{x}}) = \frac{1}{2} \nabla_{\hat{\boldsymbol{x}}}^2 \hat{p}(\hat{\boldsymbol{x}})$. According to Green's identity (Amazigo & Rubenfeld, 1980) if $\phi(x)$ and $\psi(\boldsymbol{x})$ are twice continuously differentiable functions in $\boldsymbol{R}^n$ and if $V$ is any set bounded by a piecewise smooth, closed, and oriented surface $S$ in $\mathbf{R}^n$, then

$$
\int_V \phi \nabla^2 \psi \mathrm{d}V = \int_S \phi \nabla \psi \cdot \mathrm{d}\boldsymbol{s} - \int_V \nabla \phi \cdot \nabla \psi \, \mathrm{d}V
\tag{38}
$$

where $\nabla \psi$ denotes the gradient of $\psi$, $\mathrm{d}\boldsymbol{s}$ denotes the elementary area vector, and $\nabla \psi \cdot \mathrm{d}\boldsymbol{s}$ is the inner product of these two vectors. This identity plays the role of integration by parts in $\mathbf{R}^n$. To apply Green's identity to Eq. (37), we let $V_r$ be the $n$ sphere of radius $r$ centered at the origin and having surface $S_r$. Then we use Green's identity on $V_r$ and $S_r$ with $\phi(\hat{\boldsymbol{x}}) = \log \hat{p}(\hat{\boldsymbol{x}})$ and $\psi(\hat{\boldsymbol{x}}) = \hat{p}(\hat{\boldsymbol{x}})$ and take the limit as $r \to \infty$. Hence we obtain

$$
\begin{aligned}
\frac{d}{d\sigma_t^2} h(\boldsymbol{X} + \sigma_e \boldsymbol{N}) & = -\frac{1}{2} \int_{-\infty}^{+\infty} \nabla_{\hat{\boldsymbol{x}}} \hat{p}(\hat{\boldsymbol{x}}) \nabla_{\hat{\boldsymbol{x}}} \log \hat{p}(\hat{\boldsymbol{x}}) \, \mathrm{d}\hat{\boldsymbol{x}} \\
& = \frac{1}{2} \int_{-\infty}^{+\infty} \frac{\|\nabla_{\hat{\boldsymbol{x}}} \hat{p}(\hat{\boldsymbol{x}})\|^2}{\hat{p}(\hat{\boldsymbol{x}})} \, \mathrm{d}\hat{\boldsymbol{x}}.
\end{aligned}
\tag{39}
$$

The second term in Eq. (36) can be further simplified by using Green's identity in Eq. (38),

$$
\frac{d}{d\sigma_t^2} \int_{-\infty}^{+\infty} \hat{p}(\hat{\boldsymbol{x}}) \log \bar{p}(\bar{\boldsymbol{x}}) \, \mathrm{d}\boldsymbol{x} = -\frac{1}{2} \int_{-\infty}^{+\infty} \nabla_{\hat{\boldsymbol{x}}} \hat{p}(\hat{\boldsymbol{x}}) \nabla_{\bar{\boldsymbol{x}}} \log \bar{p}(\bar{\boldsymbol{x}}) \, \mathrm{d}\boldsymbol{x}.
\tag{40}
$$

Therefore, substituting Eq. (39) and Eq. (40) into Eq. (36),

$$
\begin{aligned}
& \frac{d}{d\sigma_t^2} \theta_g(\sigma_t^2, \sigma_e^2 \boldsymbol{I}_d) \\
& = -\frac{1}{2} \int_{-\infty}^{+\infty} \frac{\|\nabla_{\hat{\boldsymbol{x}}} \hat{p}(\hat{\boldsymbol{x}})\|^2}{\hat{p}(\hat{\boldsymbol{x}})} \, \mathrm{d}\hat{\boldsymbol{x}} + \frac{1}{2} \int_{-\infty}^{+\infty} \nabla_{\hat{\boldsymbol{x}}} \hat{p}(\hat{\boldsymbol{x}}) \nabla_{\bar{\boldsymbol{x}}} \log \bar{p}(\bar{\boldsymbol{x}}) \, \mathrm{d}\boldsymbol{x} \\
& = \mathbb{E}_{\hat{p}(\hat{\boldsymbol{x}})} \left\{ -\frac{1}{2} \|\nabla_{\hat{\boldsymbol{x}}} \log \hat{p}(\hat{\boldsymbol{x}})\|^2 + \frac{1}{2} \nabla_{\hat{\boldsymbol{x}}} \log \hat{p}(\hat{\boldsymbol{x}}) \cdot \nabla_{\bar{\boldsymbol{x}}} \log \bar{p}(\bar{\boldsymbol{x}}) \right\}
\end{aligned}
\tag{41}
$$

$\square$

### B.5 Proof of Corollary 1

*Proof.* We first prove that $\sigma_{t,o}^2$ satisfying $\rho(\sigma_{t,o}^2, \mathbf{0}_d) = \rho(\sigma_{t,o}^2, \lambda_{\max} \boldsymbol{I}_d)$ is the optimal solution of Eq. (11). For i.i.d. inference noise with $\boldsymbol{\Sigma}_e = \sigma_e^2 \boldsymbol{I}_d$, it can be easily derived from Eq. (6) that

$$
\frac{\partial \rho(\sigma_t^2, \boldsymbol{\Sigma}_e)}{\partial \sigma_t^2} = \frac{1}{2} \frac{\partial \log |\boldsymbol{\Sigma}_s + \sigma_t^2 \boldsymbol{I}_d|}{\partial \sigma_t^2} + \frac{1}{2} \frac{\partial \operatorname{Tr}\left((\boldsymbol{\Sigma}_s + \sigma_t^2 \boldsymbol{I}_d)^{-1}(\boldsymbol{\Sigma}_s + \sigma_e^2 \boldsymbol{I}_d)\right)}{\partial \sigma_t^2}
$$

$$
= \sum_{i=1}^d \frac{\sigma_t^2 - \sigma_e^2}{2(\sigma_t^2 + \lambda_i(\boldsymbol{\Sigma}_s))^2}.
$$

Hence, we know that for a fixed $\sigma_e^2$, the function $\rho(\sigma_t^2, \sigma_e^2 \boldsymbol{I}_d)$ is decreasing in $\sigma_t^2$ when $\sigma_t^2 < \sigma_e^2$ and increasing for $\sigma_t^2 > \sigma_e^2$.

Therefore, for $\sigma_t^2$ in the range of $\sigma_t^2 > \sigma_{t,o}^2 > 0$, at $\sigma_e^2 = 0$, we have $\rho(\sigma_t^2, \mathbf{0}_d) > \rho(\sigma_{t,o}^2, \mathbf{0}_d)$. Likewise, for $\sigma_t^2 < \sigma_{t,o}^2 < \lambda_{\max}$, at $\sigma_e^2 = \lambda_{\max}$, we have $\rho(\sigma_t^2, \lambda_{\max} \boldsymbol{I}_d) > \rho(\sigma_{t,o}^2, \lambda_{\max} \boldsymbol{I}_d)$. As a result, $\sigma_{t,o}^2$ is the optimal solution of Eq. (11).

Next, we derive the expression of $\bar{\sigma}_{t,o}$ that minimizes the average KL-divergence when $\sigma_e^2$ is uniformly distributed between 0 to $\lambda_{\max}$. Let $\phi(\sigma_t^2) = \mathbb{E}_{\sigma_e^2 \sim \mathcal{U}(0, \lambda_{\max})}\left\{\rho(\sigma_t^2, \sigma_e^2 \boldsymbol{I}_d)\right\}$. From Eq. (6), we can obtain

$$
\phi(\sigma_t^2)
$$
$$
= \frac{1}{\lambda_{\max}} \int_0^{\lambda_{\max}} \rho(\sigma_t^2, \sigma_e^2 \boldsymbol{I}_d) \, \mathrm{d}\sigma_e^2
$$
$$
= \frac{1}{2\lambda_{\max}} \int_0^{\lambda_{\max}} \operatorname{Tr}\left[(\boldsymbol{\Sigma}_s + \sigma_t^2 \boldsymbol{I}_d)^{-1}(\boldsymbol{\Sigma}_s + \sigma_e^2 \boldsymbol{I}_d)\right] + \log\left|\boldsymbol{\Sigma}_s + \sigma_t^2 \boldsymbol{I}_d\right| - \log\left|\boldsymbol{\Sigma}_s + \sigma_e^2 \boldsymbol{I}_d\right| - d \, \mathrm{d}\sigma_e^2
$$
$$
= \frac{1}{4} \operatorname{Tr}\left[(2\boldsymbol{\Sigma}_s + \lambda_{\max} \boldsymbol{I}_d)(\sigma_t^2 \boldsymbol{I}_d + \boldsymbol{\Sigma}_s)^{-1}\right] + \frac{1}{2} \log\left|\boldsymbol{\Sigma}_s + \sigma_t^2 \boldsymbol{I}_d\right| + \alpha,
$$

where $\alpha = -\frac{1}{2} \int_0^{\lambda_{\max}} \log\left|\boldsymbol{\Sigma}_s + \sigma_e^2 \boldsymbol{I}_d\right| - d \, \mathrm{d}\sigma_e^2$ is a constant. Taking the derivative of $\phi(\sigma_t^2)$,

$$
\frac{\mathrm{d}}{\mathrm{d}\sigma_t^2} \phi(\sigma_t^2)
$$
$$
= \frac{\mathrm{d}}{\mathrm{d}\sigma_t^2} \frac{1}{4} \operatorname{Tr}\left[(2\boldsymbol{\Sigma}_s + \lambda_{\max} \boldsymbol{I}_d)(\sigma_t^2 \boldsymbol{I}_d + \boldsymbol{\Sigma}_s)^{-1}\right] + \frac{\mathrm{d}}{\mathrm{d}\sigma_t^2} \frac{1}{2} \log\left|\boldsymbol{\Sigma}_s + \sigma_t^2 \boldsymbol{I}_d\right|
$$
$$
= -\frac{1}{4} \operatorname{Tr}\left[(2\boldsymbol{\Sigma}_s + 2\lambda_{\max} \boldsymbol{I}_d)(\sigma_t^2 \boldsymbol{I}_d + \boldsymbol{\Sigma}_s)^{-2}\right] + \frac{1}{2} \operatorname{Tr}\left[(\boldsymbol{\Sigma}_s + \sigma_t^2 \boldsymbol{I}_d)^{-1}\right]
$$
$$
= \frac{1}{4} \operatorname{Tr}\left[(2\sigma_t^2 - \lambda_{\max})(\boldsymbol{\Sigma}_s + \sigma_t^2 \boldsymbol{I}_d)^{-2}\right].
$$

As $(\boldsymbol{\Sigma}_s + \sigma_t^2 \boldsymbol{I}_d)^2 > 0$, the above result implies that $\phi(\sigma_t^2)$ is a decreasing function when $\sigma_t^2 < \frac{1}{2}\lambda_{\max}$ and an increasing one if $\sigma_t^2 > \frac{1}{2}\lambda_{\max}$. Therefore, the minimal value of $\phi(\sigma_t^2)$ is achieved when $\sigma_t^2 = \frac{1}{2}\lambda_{\max}$. $\qquad \square$

### B.6 Extending to Gaussian Mixture Models (GMM) Sources

Gaussian Mixture Models (GMM) are often extended to fit a vector of unknown parameters. Given that numerous natural image signals can be effectively represented by GMM, we therefore posit the fitting of the source signal through a GMM assumption.

Consider a source signal $\boldsymbol{x}$ represented by the Gaussian mixture model that $p_{\boldsymbol{X}}(\boldsymbol{x}) = \sum_{k=1}^N \pi_k \cdot \mathcal{N}(\boldsymbol{x}|\boldsymbol{\mu}_k, \boldsymbol{\Sigma}_k)$, $\sum_{k=1}^N \pi_k = 1$ and $\pi_k > 0$. Although we cannot get a closed-form expression of the KL divergence for the GMM signal source, we can extend our results to its upper-bound, as listed below:

- **Extention of Lemma 1:** By using the convexity property of $KL$-divergence, we have
$D_{KL}(\hat{P}_{\boldsymbol{X}+\boldsymbol{E}} \| \bar{P}_{\boldsymbol{X}+\sigma_t \boldsymbol{N}}) \leq \zeta(\sigma_t^2, \boldsymbol{\Sigma}_e)$.

$$\zeta(\sigma_t^2, \mathbf{\Sigma}_e) = \sum_{k=1}^{N} \pi_k \cdot D_{KL}(\hat{P}_{\mathbf{X}_k + \mathbf{E}} \| \bar{P}_{\mathbf{X}_k + \sigma_t \mathbf{N}})$$

$$= \sum_{k=1}^{N} \pi_k \cdot \rho_k(\sigma_t^2, \mathbf{\Sigma}_e), \tag{42}$$

where $\mathbf{X}$ is d-dimensional random variable of the GMM represented source signal $\mathbf{x}$, $\mathbf{X}_k$ is the $k$-th Gaussian distribution with its weight $\pi_k$ for GMM and $\rho_k(\sigma_t^2, \mathbf{\Sigma}_e)$ is the KL-divergence of $k$-th Gaussian distribution for arbitrary noise. The above equation implies that for GMM fitted source signal, its KL-divergence concerning arbitrary noise is upper-bounded by the weighted sum of the KL-divergence of $N$ Gaussian source signals with respect to arbitrary noise.

- **Extension of Theorem 2:** For $\zeta(\sigma_t^2, \mathbf{\Sigma}_e)$ given above, when $\mathbf{\Sigma}_e = \sigma_e^2 \mathbf{\Sigma}_{\tilde{e}}$, $\zeta(\sigma_t^2, \mathbf{\Sigma}_e)$ is also convex in $\sigma_e^2$ as each $\rho_k(\sigma_t^2, \mathbf{\Sigma}_e)$ is convex in $\sigma_e^2$. Likewise, $\zeta(\sigma_t^2, \mathbf{\Sigma}_e) < \zeta(0, \mathbf{\Sigma}_e)$ when $\mathbf{\Sigma}_e > \sigma_t^2/2$.

- **Extension of Corollary 1:** Under the assumptions of Corollary 1, the optimal solution

$$\bar{\sigma}_{t,o}^2 = \arg\min_{\sigma_t^2} \mathbb{E}_{\sigma_e^2 \sim \mathcal{U}(0, \lambda_{\max})} \left\{ \zeta(\sigma_t^2, \sigma_e^2 \mathbf{I}_d) \right\} \tag{43}$$

is still $\bar{\sigma}_{t,o}^2 = \frac{1}{2}\lambda_{\max}$, which remains the same as that of the Gaussian source.

## C  EXPERIMENT SETTINGS

### C.1  BASELINES

We have employed the Gaussian noise-injected training methodology in various image-to-image translation models and contrasted them with their original baselines. Specifically,

- HiFaceGAN[1] (Yang et al., 2020), a GAN-based I2I model primarily utilized for Face Super-Resolution task on real-life facial photographs, was tested with the FFHQ dataset, specifically on a $16\times$ $(32 \rightarrow 512)$ environment.

- GP-UNIT[2] (Yang et al., 2023b), a generative prior-based image translation model for converting images between unpaired data domains, was subjected to the Cat$\rightarrow$Dog image translation task on the AFHQ dataset.

- Sketch Transformer[3] (Zhu et al., 2021), a Transformer-based image translation model, was evaluated on its ability to convert photo-sketch paired data from the CUFS dataset in the Photo$\rightarrow$Sketch image translation task.

### C.2  DATASETS

We validate the GNI method on various datasets in the baseline models presented below: (1) FFHQ natural face dataset (Karras et al., 2019). It comprises 70000 high-quality facial images. We selected the 10000 images with the lowest serial number for training, while the final 1000 images were used for testing. We perform a $16\times$ face super-resolution task on this dataset, where the HR resolution is $512\times$, and the LR resolution is $32\times$. (2) AFHQ animal dataset (Choi et al., 2020). It contains high-resolution images of animal faces, including cats, dogs, and wild animals, from three domains with substantial variations. Each domain comprises 500 test images. We perform the Cat$\rightarrow$Dog image translation task on this dataset. (3) CUFS face sketch dataset (Wang et al., 2018). It contains 188 identities from the CUHK student database (Tang & Wang, 2003), 123 from the AR database (Martinez & Benavente, 1998), and 295 from the XM2VTS database (Messer et al., 1999). We perform the Photo$\rightarrow$Sketch image translation task on this dataset.

---

[1]https://github.com/Lotayou/Face-Renovation

[2]https://github.com/williamyang1991/GP-UNIT

[3]shared by the authors

Table 4: Parameter configuration for noise intensity in the test set for testing I2I translation models.

| Noise Type | | Noise | | | | | | | Blur | | Digital | |
|---|---|---|---|---|---|---|---|---|---|---|---|---|
| | | Gaussian $\sigma_e^2$ | Uniform $\sigma_e^2$ | Color $\sigma_e^2$ | Laplacian $\sigma_e^2$ | Salt&Pepper Density | Shot Density | Speckle Density | Glass Density | Defocus Radius | Pixelate Density | JPEG Quality |
| Intensity | 1 | 0.01 | 0.01 | 0.01 | 0.01 | 0.05 | 60 | 0.15 | 1 | 1 | 0.5 | 30 |
| | 2 | 0.02 | 0.02 | 0.02 | 0.02 | 0.10 | 40 | 0.25 | 2 | 2 | 1.0 | 25 |
| | 3 | 0.04 | 0.04 | 0.04 | 0.04 | 0.15 | 25 | 0.35 | 3 | 4 | 1.5 | 20 |
| | 4 | 0.05 | 0.05 | 0.05 | 0.05 | 0.20 | 12 | 0.45 | 4 | 5 | 2.0 | 15 |
| | 5 | 0.09 | 0.09 | 0.09 | 0.09 | 0.25 | 5 | 0.55 | 6 | 6 | 2.5 | 10 |
| | 6 | 0.16 | 0.16 | 0.16 | 0.16 | 0.30 | 3 | 0.65 | 7 | 7 | 3.0 | 6 |

## C.3 DEGRADATION

We considered multiple types of image degradation, each with 6 different intensities from weak to strong, as detailed in Table 4. The parameter configurations for noise, blur and digital degradation are based on those used in (Hendrycks & Dietterich, 2018) for ImageNetC. For colored noise, we adopt a strategy similar to (Kaneko & Harada, 2020) by applying a 2D Gaussian filter to i.i.d. Gaussian noise with a standard deviation of $0.5$ and a $7 \times 7$ window size. To generate noisy images, we start by normalizing the pixel intensities to fit the $[0, 1]$ first. Then, after introducing the noise, we ensure the pixel values remain bounded within the same range by a clipping operation. After that, we adjust the pixel values to the $[0, 255)$ range to produce an image file.

## C.4 SETTING OF $\sigma_t^2$

For i.i.d. noise, the peak inference noise level in our simulation is $\lambda_{\max} = 0.16$, as indicated in Table 4. According to Corollary 1, the optimal value to minimize the average KL-divergence is $\bar{\sigma}_{t,o}^2 = \lambda_{\max}/2 = 0.08$. This value is noise type-independent and is straightforward to compute. In our simulations, we set $\sigma_t^2 = 0.04$ by default. Several reasons motivate this choice over $0.08$:

1. **Model integrity for clean Images**: Eq. (4) suggests a smaller $\sigma_t^2$ ensures the model's efficacy on noise-free images. A larger value risks training noise bias, undermining its performance on clean data.

2. **Near optimal Min-Max Value for Gaussian Noise**: As depicted in Fig. 6(a), the optimal $\sigma_t^2$ from Eq. (11) to minimize worst-case KL-divergence falls below $\lambda_{\max}/2$. Specifically, $\sigma_t^2 = 0.04$ stands as a near-optimal min-max solution.

3. **Visual quality vs. KL-divergence**: While KL-divergence offers statistical insight, it might not reflect the visual quality of translated images. Simulations suggest $\sigma_t^2 = 0.04$ strikes a balanced robustness-quality trade-off.

## C.5 TRAINING

During our training process, we follow the default settings of each baseline model, only substituting clean images in the source domain with their noisy variants. Consequently, both training duration and memory requirements remain unchanged. Drawing insights from prior work on unconditional GANs, it's suggested that incorporating noise into training images can improve convergence and enhance training stability. Indeed, in our experiments, all three GAN-based I2I models achieved convergence without any instances of model collapse—a frequent challenge in GAN training. Importantly, with a noise intensity of $\sigma_t^2 = 0.04$, the Gaussian noise-injected model does not bias the noisy inputs, and they demonstrate good performance on clean inputs as well, as we will show in the next Section.

## D ADDITIONAL RESULTS

### D.1 QUANTITATIVE RESULTS AND COMPARISON WITH THEORETICAL ANALYSIS

In the Cat→Dog translation task, the model, trained using Gaussian noise injection, shows superior noise robustness over the baseline across five different noise scenarios, as highlighted in Table 5 (latent-guided). Even with significant noise interference, the GNI approach consistently delivers

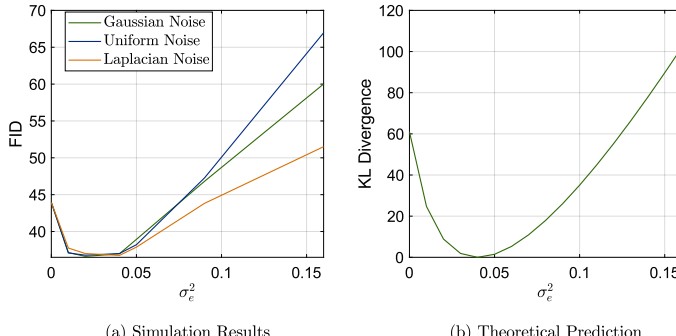

Figure 7: Comparison results of face super-resolution (a) FID results. (B) Theoretical results of KL-divergence with Gaussian noise.

stable results. Besides, Table 6 demonstrates that the noise-injected model outperforms the baseline in popular evaluation metrics like PSNR and FID for face super-resolution tasks. These results echo our theoretical analysis on the following aspects:

1. **Clean inputs**: Despite training only with noisy source images, the Gaussian noise-injected model effectively handle clean inputs, manifesting only slight objective result reductions. This is consistent with insights from Eq. 4, as expounded in Theorem 1.

2. **Resilience to various types of Noises**: Part 1 of Theorem 2 indicates that for Gaussian signal sources, resilience to Gaussian noise implies robustness to other noise types with a matched covariance structure. Despite natural images not being strictly Gaussian, the model trained solely on isotropic Gaussian noise demonstrates efficiency against varied non-Gaussian distortions like Laplacian, uniform, colored Gaussian, and even erasure noises like Salt & Pepper. Experiments empirically exhibit this generalized capability gained from Gaussian noise injection.

3. **Comparison with Baselines**: Training with a noise intensity of $\sigma_t^2 = 0.04$ (specifically at intensity level S3), Part 2 of Theorem 2 suggests the Gaussian noise-injected model should surpass clean-trained counterparts for noisy inputs when the inference noise intensity satisfies $\sigma_e^2 > 0.5\sigma_t^2 = 0.02$. As anticipated, the Gaussian noise-injected model excels for inputs with i.i.d. noises (e.g., Gaussian, Uniform, Laplacian) at noise intensity level S2 (corresponding to $\sigma_e^2 = 0.02$) and above, validating the theoretical expectations.

In addition, for the face super-resolution task, Fig. 7 further compares the FID results vs. $\sigma_e^2$ against the theoretical KL-divergence $\rho_g(0.04, \sigma_e^2 \boldsymbol{I}_d)$ predictions for Gaussian noise. To compute $\rho_g(0.04, \sigma_e^2 \boldsymbol{I}_d)$, we employ an AR(1) model for the FFHQ data with $d = 256$ and a covariance matrix defined as $\boldsymbol{\Sigma}_s(k, l) = \sigma_s^2 \cdot \rho^{|k-l|}$ for $0 \leq k, l \leq d - 1$. The parameters $\sigma_s^2 = 0.16$ and $\rho = 0.9$ are informed by the normalized FFHQ dataset. Remarkably, the FID trends closely resemble the derived KL-divergence, $\rho_g(0.04, \sigma_e^2 \boldsymbol{I}_d)$, presenting a convex nature in terms of $\sigma_e^2$. With small $\sigma_e^2$ values, the performance remains consistent acros s various i.i.d. noises. This parallels the insights from Part 1 of Theorem 2 regarding KL-divergence.

In Table 7, we present average FID scores for Photo→Sketch task with different training noises variances $\sigma_t^2$. The simulation results in this table demonstrate that $\sigma_t^2 = 0.08$ yields the lowest average FID scores. This empirical finding closely aligns with the prediction in Theorem 2, thus providing strong empirical support for our theoretical analysis.

While our simulation results align in many respects with theoretical predictions, there are notable discrepancies between simulation outcomes and theoretical anticipations:

1. **Gaussian Source Assumption**: Lemma 1 asserts that for a Gaussian source trained via Gaussian noise injection, Gaussian noise during inference would result in the smallest KL-divergence. Contrary to this, Fig. 7 reveals that Laplacian noise corruption often yields superior objective results. This might stem from the inherently non-Gaussian nature of image signals. Furthermore, we clip all test image pixel values to the range $[0, 1)$ to produce noisy

Table 5: Quantitative comparison on the **Cat→Dog** image translation task. In the latent-guided approach, we randomly generate 10 latent codes as additional guides, and synthesize a total of 5000 images. **Note:** the random seed remains fixed at 777.

| Noise Type | Metric | Method | Noise Intensity | | | | | | |
|---|---|---|---|---|---|---|---|---|---|
| | | | Clean | S1 | S2 | S3 | S4 | S5 | S6 |
| Gaussian Noise | FID ↓ | Baseline | **22.82** | **22.35** | 24.29 | 26.12 | 26.98 | 31.33 | 40.34 |
| | | $+\mathcal{N}(0, 0.04)$ | 25.04 | 23.81 | **22.89** | **22.29** | **22.21** | **21.88** | **21.76** |
| | KID ↓ | Baseline | **9.67** | 10.61 | 11.78 | 13.92 | 14.78 | 18.46 | 27.11 |
| | | $+\mathcal{N}(0, 0.04)$ | 9.89 | **9.14** | **8.52** | **8.09** | **7.99** | **7.82** | **7.88** |
| Uniform Noise | FID ↓ | Baseline | **22.82** | 23.61 | 24.28 | 26.42 | 27.74 | 32.42 | 45.38 |
| | | $+\mathcal{N}(0, 0.04)$ | 25.04 | 23.72 | **22.87** | **22.32** | **22.21** | **21.97** | **21.71** |
| | KID ↓ | Baseline | **9.67** | 10.81 | 11.78 | 13.94 | 15.66 | 20.19 | 30.56 |
| | | $+\mathcal{N}(0, 0.04)$ | 9.89 | **9.09** | **8.55** | **8.21** | **8.12** | **7.89** | **7.91** |
| Color Noise | FID ↓ | Baseline | **22.82** | 23.47 | 24.85 | 27.49 | 29.17 | 35.31 | 48.61 |
| | | $+\mathcal{N}(0, 0.04)$ | 25.04 | 23.57 | **22.86** | **22.45** | **22.19** | **22.13** | **22.11** |
| | KID ↓ | Baseline | **9.67** | 11.03 | 12.51 | 15.19 | 16.89 | 22.08 | 33.98 |
| | | $+\mathcal{N}(0, 0.04)$ | 9.89 | **9.01** | **8.46** | **8.23** | **7.99** | **7.96** | **8.02** |
| Laplacian Noise | FID ↓ | Baseline | **22.82** | 23.51 | 24.18 | 25.66 | 26.42 | 29.65 | 35.11 |
| | | $+\mathcal{N}(0, 0.04)$ | 25.04 | 23.79 | **22.97** | **22.47** | **22.36** | **21.92** | **21.76** |
| | KID ↓ | Baseline | **9.67** | 10.83 | 11.71 | 13.33 | 14.02 | 17.18 | 22.12 |
| | | $+\mathcal{N}(0, 0.04)$ | 9.89 | **9.18** | **8.58** | **8.25** | **8.09** | **7.79** | **7.74** |
| Salt & Pepper Noise | FID ↓ | Baseline | **22.82** | 23.86 | 26.42 | 27.53 | 31.67 | 35.91 | 41.11 |
| | | $+\mathcal{N}(0, 0.04)$ | 25.04 | 23.92 | **22.51** | **22.19** | **22.02** | **21.91** | **22.01** |
| | KID ↓ | Baseline | **9.67** | 10.61 | 11.78 | 13.92 | 14.78 | 18.46 | 27.11 |
| | | $+\mathcal{N}(0, 0.04)$ | 9.89 | **9.14** | **8.52** | **8.09** | **7.99** | **7.82** | **7.88** |

Table 6: Quantitative comparison on the **Face Super-Resolution** image translation task. For face super-resolution, we adopt the configuration provided by HiFaceGAN (Yang et al., 2020), explicitly selecting the final 1000 face images from the FFHQ dataset as our designated testset.

| Noise Type | Metric | Method | Noise Intensity | | | | | | |
|---|---|---|---|---|---|---|---|---|---|
| | | | Clean | S1 | S2 | S3 | S4 | S5 | S6 |
| Gaussian Noise | FID ↓ | Baseline | **34.83** | 96.21 | 122.48 | 166.94 | 191.21 | 263.96 | 320.41 |
| | | $+\mathcal{N}(0, 0.04)$ | 43.94 | **37.15** | **36.62** | **36.98** | **38.04** | **46.81** | **60.01** |
| | PSNR ↑ | Baseline | **22.09** | 20.09 | 19.76 | 19.01 | 17.35 | 16.24 | 15.11 |
| | | $+\mathcal{N}(0, 0.04)$ | 20.43 | **21.23** | **21.65** | **21.59** | **21.14** | **20.51** | **19.77** |
| Uniform Noise | FID ↓ | Baseline | **34.83** | 95.74 | 124.22 | 173.31 | 197.33 | 279.81 | 329.63 |
| | | $+\mathcal{N}(0, 0.04)$ | 43.94 | **37.09** | **36.79** | **37.01** | **38.21** | **42.27** | **66.96** |
| | PSNR ↑ | Baseline | **22.09** | 20.11 | 18.99 | 17.69 | 17.25 | 16.06 | 24.76 |
| | | $+\mathcal{N}(0, 0.04)$ | 20.43 | **20.98** | **21.41** | **21.84** | **21.89** | **21.35** | **19.75** |
| Color Noise | FID ↓ | Baseline | **34.83** | 99.91 | 130.08 | 182.57 | 208.83 | 282.82 | 328.49 |
| | | $+\mathcal{N}(0, 0.04)$ | 43.94 | **37.91** | **37.39** | **36.81** | **40.11** | **48.99** | **65.11** |
| | PSNR ↑ | Baseline | **22.09** | 19.98 | 18.84 | 17.56 | 17.25 | 16.06 | 14.95 |
| | | $+\mathcal{N}(0, 0.04)$ | 20.43 | **20.95** | **21.31** | **21.74** | **21.83** | **21.71** | **20.94** |
| Laplacian Noise | FID ↓ | Baseline | **34.83** | 93.73 | 117.37 | 155.33 | 172.15 | 235.65 | 292.31 |
| | | $+\mathcal{N}(0, 0.04)$ | 43.94 | **37.76** | **37.01** | **36.76** | **37.88** | **43.82** | **51.51** |
| | PSNR ↑ | Baseline | **22.09** | 20.21 | 19.21 | 18.04 | 17.67 | 16.69 | 15.72 |
| | | $+\mathcal{N}(0, 0.04)$ | 20.43 | **20.95** | **21.31** | **21.74** | **21.83** | **21.71** | **20.94** |
| Salt & Pepper Noise | FID ↓ | Baseline | **34.83** | 112.66 | 153.43 | 201.41 | 250.88 | 291.62 | 317.24 |
| | | $+\mathcal{N}(0, 0.04)$ | 43.94 | **39.33** | **41.25** | **44.58** | **50.54** | **57.21** | **64.78** |
| | PSNR ↑ | Baseline | **22.09** | 19.38 | 17.98 | 17.01 | 16.23 | 15.54 | 14.94 |
| | | $+\mathcal{N}(0, 0.04)$ | 20.43 | **21.23** | **21.65** | **21.59** | **21.14** | **20.51** | **19.77** |

Table 7: For **Photo→Sketch** image translation task, average FID scores comparison on noisy input images of different intensities for models trained with different Gaussian noise levels.

| Noise Type | Noise Injection $\sigma_t^2$ | | | | | |
|---|---|---|---|---|---|---|
| | 0 | 0.01 | 0.04 | 0.08 | 0.16 | Learnable |
| Gaussian Noise | 182.55 | 78.89 | 42.17 | **36.18** | 36.64 | 59.19 |
| Uniform Noise | 199.17 | 101.68 | 45.37 | **36.79** | 36.89 | 64.81 |
| Laplacian Noise | 152.09 | 64.12 | 39.17 | **36.99** | 37.01 | 52.41 |

Table 8: More image degradation test comparisons on **Photo→Sketch** task.

| Type | | Metric | Method | Intensity | | | | | | |
|---|---|---|---|---|---|---|---|---|---|---|
| | | | | Clean | S1 | S2 | S3 | S4 | S5 | S6 |
| Noise | Shot Noise | FID ↓ | Baseline | **31.49** | 47.33 | 54.74 | 64.79 | 91.46 | 219.11 | 385.72 |
| | | | $+\mathcal{N}(0, 0.04)$ | 64.12 | **38.93** | **35.07** | **32.49** | **30.85** | **39.41** | **58.06** |
| | | LPIPS ↓ | Baseline | **0.3055** | 0.3851 | 0.4036 | 0.4286 | 0.4848 | 0.5885 | 0.6434 |
| | | | $+\mathcal{N}(0, 0.04)$ | 0.3601 | **0.3329** | **0.3267** | **0.3212** | **0.3196** | **0.3459** | **0.3954** |
| | Speckle Noise | FID ↓ | Baseline | **31.49** | **41.14** | 58.76 | 84.59 | 125.21 | 194.61 | 270.52 |
| | | | $+\mathcal{N}(0, 0.04)$ | 64.12 | 42.91 | **34.44** | **32.22** | **34.08** | **41.16** | **51.73** |
| | | LPIPS ↓ | Baseline | **0.3055** | 0.3684 | 0.4186 | 0.4721 | 0.5213 | 0.5637 | 0.5955 |
| | | | $+\mathcal{N}(0, 0.04)$ | 0.3601 | **0.3453** | **0.3338** | **0.3216** | **0.3321** | **0.3634** | **0.3858** |
| Blur | Glass Blur | FID ↓ | Baseline | **31.49** | **31.72** | 48.45 | 52.53 | 64.82 | 83.08 | 91.04 |
| | | | $+\mathcal{N}(0, 0.04)$ | 64.12 | 63.39 | **51.59** | **52.01** | **53.47** | **57.18** | **58.54** |
| | | LPIPS ↓ | Baseline | **0.3055** | **0.3061** | **0.3512** | 0.3621 | 0.3869 | 0.4111 | 0.4231 |
| | | | $+\mathcal{N}(0, 0.04)$ | 0.3601 | 0.3589 | 0.3521 | **0.3511** | **0.3559** | **0.3645** | **0.3704** |
| | Defocus Blur | FID ↓ | Baseline | **31.49** | **32.17** | **36.38** | **45.86** | **52.59** | **64.22** | 92.82 |
| | | | $+\mathcal{N}(0, 0.04)$ | 64.12 | 64.51 | 68.65 | 70.91 | 77.95 | 84.21 | **92.11** |
| | | LPIPS ↓ | Baseline | **0.3055** | **0.3098** | **0.3232** | **0.3516** | **0.3678** | **0.3901** | 0.4256 |
| | | | $+\mathcal{N}(0, 0.04)$ | 0.3601 | 0.3641 | 0.3701 | 0.3823 | 0.3931 | 0.4122 | **0.4213** |
| Digital | Pixelate | FID ↓ | Baseline | **31.49** | **39.77** | **57.61** | 77.33 | 89.87 | 108.11 | 145.32 |
| | | | $+\mathcal{N}(0, 0.04)$ | 64.12 | 62.59 | 64.04 | **67.14** | **74.08** | **85.41** | **100.01** |
| | | LPIPS ↓ | Baseline | **0.3055** | **0.3282** | **0.3671** | 0.3987 | 0.4341 | 0.4611 | 0.4901 |
| | | | $+\mathcal{N}(0, 0.04)$ | 0.3601 | 0.3641 | 0.3684 | **0.3752** | **0.3738** | **0.3869** | **0.4011** |
| | JPEG | FID ↓ | Baseline | **31.49** | **34.79** | **36.02** | **36.89** | **41.16** | 47.39 | 71.35 |
| | | | $+\mathcal{N}(0, 0.04)$ | 64.12 | 62.51 | 61.22 | 60.63 | 59.21 | **55.52** | **53.72** |
| | | LPIPS ↓ | Baseline | **0.3055** | **0.3192** | **0.3236** | **0.3301** | **0.3571** | 0.3699 | 0.4291 |
| | | | $+\mathcal{N}(0, 0.04)$ | 0.3601 | 0.3611 | 0.3602 | 0.3631 | 0.3653 | **0.3665** | **0.3786** |

    test images. Given the extended tails of Laplacian noise, this clipping could inadvertently reduce noise levels.

2. **FID vs. KL-Divergence in I2I Tasks**: For supervised I2I tasks such as photo-to-sketch translation and face super-resolution, FID scores largely mirror KL-divergence trends. However, in the unsupervised Cat→Dog translation via the GP-UNIT model, FID results diverge from theoretical KL calculations. This discrepancy might arise because, apart from the source image, GP-UNIT integrates an additional latent code or reference image for I2I translation. In contrast, our theoretical framework solely contemplates the source images as inputs. As noise intensities escalate, FID and KID metrics for GP-UNIT models improve.

A comprehensive analysis of these results, especially the intriguing behavior of GP-UNIT models under increased noise, will be the subject of our future investigations.

## D.2  QUANTITATIVE RESULTS OF OTHER IMAGE DEGRADATION MODELS

We also assessed the effectiveness of Gaussian noise injection in handling other image degradation models outlined in Imagenet-C. These models include a range of degradations, including signal-dependent noises such as shot and speckle noises, as well as various blurring operators and digital operations like JPEG compression and pixelation (as detailed in Table 4).

Table 9: More image degradation test comparisons on **Cat→Dog** task (latent-guided).

| | Type | Metric | Method | Clean | S1 | S2 | S3 | S4 | S5 | S6 |
|---|---|---|---|---|---|---|---|---|---|---|
| Noise | Shot Noise | FID↓ | Baseline | **22.82** | 24.42 | 23.74 | 24.31 | 26.09 | 30.91 | 37.51 |
| | | | +$\mathcal{N}(0, 0.04)$ | 25.04 | **24.03** | **23.44** | **22.85** | **22.39** | **21.76** | **21.88** |
| | | KID↓ | Baseline | **9.67** | 10.86 | 11.09 | 11.59 | 13.49 | 18.27 | 24.51 |
| | | | +$\mathcal{N}(0, 0.04)$ | 9.89 | **9.31** | **8.98** | **8.46** | **8.12** | **7.79** | **7.91** |
| | Speckle Noise | FID↓ | Baseline | **22.82** | **23.36** | 24.25 | 25.41 | 26.98 | 29.35 | 31.53 |
| | | | +$\mathcal{N}(0, 0.04)$ | 25.04 | 24.25 | **23.02** | **22.37** | **22.08** | **21.99** | **21.91** |
| | | KID↓ | Baseline | **9.67** | 10.69 | 11.64 | 12.97 | 14.53 | 16.97 | 19.15 |
| | | | +$\mathcal{N}(0, 0.04)$ | 9.89 | **9.46** | **8.61** | **8.37** | **7.96** | **8.07** | **7.92** |
| Blur | Glass Blur | FID↓ | Baseline | **22.82** | **22.85** | **23.91** | **23.04** | **23.12** | **23.51** | **23.98** |
| | | | +$\mathcal{N}(0, 0.04)$ | 25.04 | 25.06 | 25.09 | 25.18 | 24.89 | 24.61 | 24.51 |
| | | KID↓ | Baseline | **9.67** | **9.69** | **9.82** | **9.86** | **10.06** | **10.44** | **10.56** |
| | | | +$\mathcal{N}(0, 0.04)$ | 9.89 | 9.88 | 9.92 | 10.03 | **9.98** | **9.91** | **9.77** |
| | Defocus Blur | FID↓ | Baseline | **22.82** | **23.08** | **23.41** | **24.71** | **25.71** | 26.52 | 27.13 |
| | | | +$\mathcal{N}(0, 0.04)$ | 25.04 | 25.92 | 26.01 | 26.02 | 25.91 | **25.69** | **25.45** |
| | | KID↓ | Baseline | **9.67** | **9.81** | **9.98** | 10.71 | 11.31 | 11.97 | 12.45 |
| | | | +$\mathcal{N}(0, 0.04)$ | 9.89 | 10.44 | 10.61 | **10.67** | **10.59** | **10.48** | **10.29** |
| Digital | Pixelate | FID↓ | Baseline | **22.82** | **22.89** | **23.31** | **23.42** | **23.85** | **23.91** | **24.01** |
| | | | +$\mathcal{N}(0, 0.04)$ | 25.04 | 25.29 | 25.84 | 26.01 | 25.81 | 25.71 | 25.69 |
| | | KID↓ | Baseline | **9.67** | **9.72** | **9.82** | **10.06** | **10.21** | **10.44** | 10.91 |
| | | | +$\mathcal{N}(0, 0.04)$ | 9.89 | 10.03 | 10.35 | 10.51 | 10.54 | 10.45 | **10.42** |
| | JPEG | FID↓ | Baseline | **22.82** | **22.91** | **22.95** | **23.01** | **23.26** | **23.41** | **24.36** |
| | | | +$\mathcal{N}(0, 0.04)$ | 25.04 | 25.19 | 24.94 | 25.21 | 25.41 | 25.49 | 25.53 |
| | | KID↓ | Baseline | **9.67** | **9.79** | 10.01 | 10.07 | 10.35 | 10.53 | 12.24 |
| | | | +$\mathcal{N}(0, 0.04)$ | 9.89 | 10.03 | **9.77** | **9.96** | **10.11** | **10.25** | **10.31** |

Table 10: More image degradation test comparisons on **Cat→Dog** task (reference-guided).

| | Type | Metric | Method | Clean | S1 | S2 | S3 | S4 | S5 | S6 |
|---|---|---|---|---|---|---|---|---|---|---|
| Noise | Shot Noise | FID↓ | Baseline | 17.82 | 18.36 | 18.39 | 19.08 | 20.71 | 26.51 | 34.07 |
| | | | +$\mathcal{N}(0, 0.04)$ | **16.47** | **16.29** | **16.21** | **16.06** | **16.19** | **15.97** | **16.01** |
| | | KID↓ | Baseline | 7.08 | 8.04 | 8.06 | 8.72 | 10.03 | 14.61 | 21.08 |
| | | | +$\mathcal{N}(0, 0.04)$ | **5.35** | **5.23** | **5.13** | **5.01** | **5.21** | **5.09** | **5.16** |
| | Speckle Noise | FID↓ | Baseline | 17.82 | 18.34 | 19.09 | 20.41 | 22.41 | 24.55 | 27.45 |
| | | | +$\mathcal{N}(0, 0.04)$ | **16.47** | **16.39** | **16.11** | **16.03** | **16.04** | **16.07** | **16.15** |
| | | KID↓ | Baseline | 7.08 | 7.95 | 8.61 | 9.82 | 11.46 | 13.31 | 15.51 |
| | | | +$\mathcal{N}(0, 0.04)$ | **5.35** | **5.25** | **5.11** | **5.08** | **5.12** | **5.26** | **5.27** |
| Blur | Glass Blur | FID↓ | Baseline | 17.82 | 17.89 | 17.91 | 17.96 | 17.99 | 18.06 | 18.11 |
| | | | +$\mathcal{N}(0, 0.04)$ | **16.47** | **16.45** | **16.44** | **16.46** | **16.55** | **16.48** | **16.47** |
| | | KID↓ | Baseline | 7.08 | 7.09 | 7.23 | 7.31 | 7.33 | 7.46 | 7.48 |
| | | | +$\mathcal{N}(0, 0.04)$ | **5.35** | **5.34** | **5.32** | **5.33** | **5.44** | **5.38** | **5.39** |
| | Defocus Blur | FID↓ | Baseline | 17.82 | 17.84 | 17.91 | 18.46 | 18.92 | 19.01 | 19.36 |
| | | | +$\mathcal{N}(0, 0.04)$ | **16.47** | **16.74** | **16.81** | **16.79** | **16.69** | **16.68** | **16.55** |
| | | KID↓ | Baseline | 7.08 | 7.09 | 7.11 | 7.15 | 7.53 | 7.86 | 8.31 |
| | | | +$\mathcal{N}(0, 0.04)$ | **5.35** | **5.54** | **5.61** | **5.57** | **5.53** | **5.56** | **5.45** |
| Digital | Pixelate | FID↓ | Baseline | 17.82 | 17.86 | 17.93 | 17.94 | 17.96 | 17.99 | 18.16 |
| | | | +$\mathcal{N}(0, 0.04)$ | **16.47** | **16.46** | **16.66** | **16.79** | **16.67** | **16.64** | **16.69** |
| | | KID↓ | Baseline | 7.08 | 7.09 | 7.12 | 7.17 | 7.26 | 7.43 | 7.89 |
| | | | +$\mathcal{N}(0, 0.04)$ | **5.35** | **5.32** | **5.48** | **5.62** | **5.54** | **5.56** | **5.58** |
| | JPEG | FID↓ | Baseline | 17.82 | 17.85 | 17.85 | 17.96 | 18.06 | 18.49 | 19.91 |
| | | | +$\mathcal{N}(0, 0.04)$ | **16.47** | **16.49** | **16.43** | **16.53** | **16.54** | **16.69** | **16.74** |
| | | KID↓ | Baseline | 7.08 | 7.10 | 7.25 | 7.53 | 7.56 | 7.91 | 9.56 |
| | | | +$\mathcal{N}(0, 0.04)$ | **5.35** | **5.37** | **5.31** | **5.32** | **5.34** | **5.38** | **5.51** |

Table 11: More image degradation test comparisons on **Face Super-Resolution** task.

| Type | | Metric | Method | Intensity | | | | | | |
|---|---|---|---|---|---|---|---|---|---|---|
| | | | | Clean | S1 | S2 | S3 | S4 | S5 | S6 |
| Noise | Shot Noise | FID↓ | Baseline | **34.83** | 86.23 | 98.06 | 114.55 | 162.75 | 261.81 | 308.57 |
| | | | $+\mathcal{N}(0, 0.04)$ | 43.94 | **37.61** | **37.21** | **36.97** | **39.11** | **47.25** | **59.41** |
| | | PSNR↑ | Baseline | **22.09** | 20.56 | 20.04 | 19.32 | 18.07 | 16.62 | 15.76 |
| | | | $+\mathcal{N}(0, 0.04)$ | 20.43 | **21.02** | **21.16** | **21.29** | **21.32** | **20.43** | **19.45** |
| | Speckle Noise | FID↓ | Baseline | **34.83** | 78.13 | 109.01 | 147.72 | 191.09 | 225.86 | 256.43 |
| | | | $+\mathcal{N}(0, 0.04)$ | 43.94 | **38.25** | **38.15** | **39.33** | **43.71** | **49.43** | **56.81** |
| | | PSNR↑ | Baseline | **22.09** | 20.85 | 19.47 | 18.23 | 17.28 | 16.59 | 16.06 |
| | | | $+\mathcal{N}(0, 0.04)$ | 20.43 | **20.99** | **21.19** | **21.21** | **20.63** | **20.18** | **19.75** |
| Blur | Glass Blur | FID↓ | Baseline | **34.83** | **34.86** | **35.89** | **36.12** | **37.45** | **40.55** | **42.41** |
| | | | $+\mathcal{N}(0, 0.04)$ | 43.94 | 43.91 | 43.85 | 43.84 | 43.81 | 43.78 | 43.56 |
| | | PSNR↑ | Baseline | **22.09** | **22.07** | **22.05** | **22.03** | **21.98** | **21.89** | **21.87** |
| | | | $+\mathcal{N}(0, 0.04)$ | 20.43 | 20.43 | 20.45 | 20.46 | 20.52 | 20.59 | 20.62 |
| | Defocus Blur | FID↓ | Baseline | **34.83** | **34.84** | **34.88** | **34.91** | **34.96** | **35.01** | **35.06** |
| | | | $+\mathcal{N}(0, 0.04)$ | 43.94 | 43.95 | 43.88 | 43.75 | 43.71 | 43.68 | 43.59 |
| | | PSNR↑ | Baseline | **22.09** | **22.08** | **22.05** | **22.01** | **21.95** | **21.94** | **21.91** |
| | | | $+\mathcal{N}(0, 0.04)$ | 20.43 | 20.42 | 20.46 | 20.56 | 20.63 | 20.73 | 20.81 |
| Digital | Pixelate | FID↓ | Baseline | **34.83** | **34.93** | **35.68** | **35.86** | **38.31** | **38.89** | **40.78** |
| | | | $+\mathcal{N}(0, 0.04)$ | 43.94 | 44.01 | 43.64 | 44.02 | 43.95 | 43.97 | 43.96 |
| | | PSNR↑ | Baseline | **22.09** | **22.05** | **22.01** | **21.97** | **21.94** | **21.91** | **21.88** |
| | | | $+\mathcal{N}(0, 0.04)$ | 20.43 | 20.43 | 20.52 | 20.53 | 20.62 | 20.67 | 20.76 |
| | JPEG | FID↓ | Baseline | **34.83** | **38.56** | **39.56** | **42.01** | 46.51 | 56.16 | 81.39 |
| | | | $+\mathcal{N}(0, 0.04)$ | 43.94 | 44.19 | 44.42 | 44.71 | **45.11** | **47.81** | **57.32** |
| | | PSNR↑ | Baseline | **22.09** | **21.96** | **21.92** | **21.84** | 21.33 | 21.35 | 20.04 |
| | | | $+\mathcal{N}(0, 0.04)$ | 20.43 | 21.98 | 21.72 | 21.51 | **21.41** | **21.47** | **21.11** |

Table 12: Robustness evaluation to multiple image degradations on the **Cat→Dog** image translation task (latent-guided). **Note:** $+Noise^1$ and $+Noise^2$ represent using Gaussian and Shot noise to perturb the degraded image.

| Setting | Blur | | Digital | | Mixing | |
|---|---|---|---|---|---|---|
| | Glass Blur | Defocus Blur | Pixelate | JPEG | Glass Blur + Pixelate | Defocus Blur + JPEG |
| Baseline | 23.12 | 25.71 | 23.85 | 23.26 | 26.35 | 25.93 |
| Noise Injection | 24.89 | 25.91 | 25.81 | 25.41 | 25.86 | 25.99 |
| Baseline+$Noise^1$ | 26.98 | 28.99 | 27.81 | 26.75 | 28.51 | 29.51 |
| Noise Injection+$Noise^1$ | **22.21** | **22.75** | **22.53** | **22.41** | **22.44** | **22.61** |
| Baseline+$Noise^2$ | 26.24 | 27.31 | 26.56 | 26.01 | 27.11 | 27.44 |
| Noise Injection+$Noise^2$ | 22.67 | 22.84 | 22.81 | 22.49 | 23.05 | 22.75 |

The objective results for all three tasks are summarized in Tables 8 to 11. Table 12 further provides results for combinations of these degradation models. Our key findings from these tables are as follows:

- For corruptions involving signal-dependent noises alone, models trained with GNI consistently demonstrated substantial improvements over baseline models trained solely on clean images, mirroring those observed with signal-independent noises.

- In cases where degradations included blurring, pixelation, or JPEG compression without additional noise, models trained with GNI were occasionally outperformed by the baselines, particularly at low to medium degradation intensities.

- In combined degradation models, whenever the noise component was present (e.g., JPEG+noise or Blur+noise), models trained with GNI consistently exhibited significant enhancements over the baseline models, as shown in Table 12. This observation is particularly relevant to real-world scenarios where noises are often present, emphasizing the effectiveness of GNI in I2I tasks.

Notably, as demonstrated in Table 12, in cases where the corruption model excluded the noise component, the addition of a small amount of noise to the input images proved effective in producing

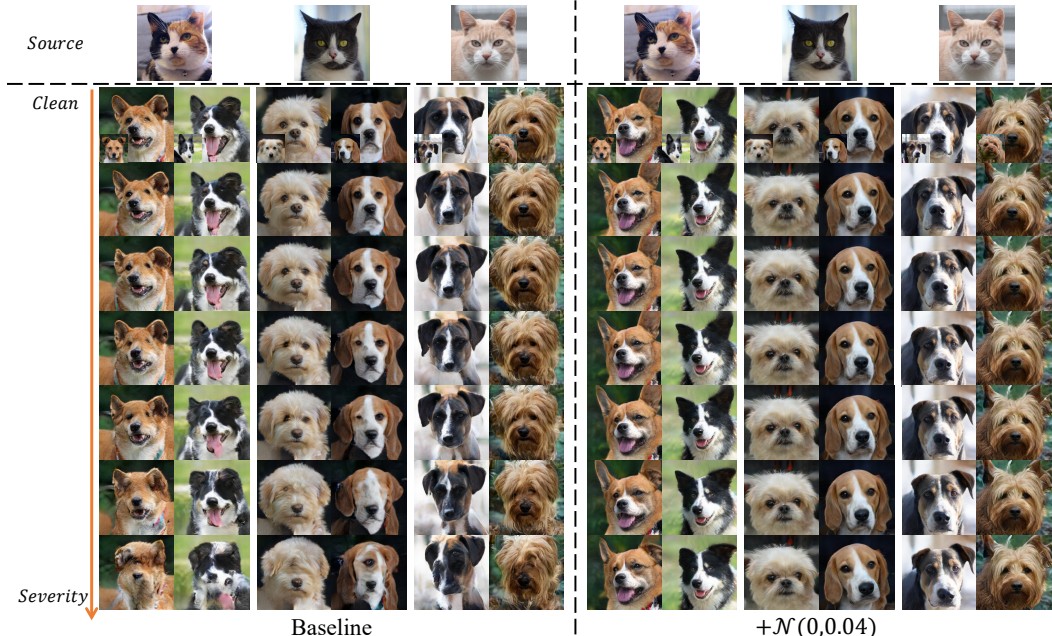

Figure 8: Comparison of reference-guided generation for Cat→Dog under the interference of Gaussian noise. Each source image is guided using two reference images. On the left is the result obtained by the baseline (GP-UNIT), and on the right is the result obtained by applying noise injection to the baseline model.

outputs with stable quality. These findings provide further insights into the robustness and potential of Gaussian noise injection in I2I for various degradations.

### D.3 ADDITIONAL QUALITATIVE RESULTS

In this subsection, we present additional comparative findings. Precisely, Figs. 8 through 12 depict the results of the Cat→Dog image translation task when subjected to Gaussian noise, Uniform noise, Color noise, Laplacian noise, and Salt & Pepper noise. Meanwhile, Figs. 13 to 17 reveal results from the same noisy environment, guided by latent factors. Furthermore, Figs. 18 through 22 display the translation outcomes of the Face Super-Resolution image translation task when exposed to Gaussian noise, Uniform noise, Color noise, Laplacian noise, and Salt & Pepper noise. These visual representations confirm the significant enhancement in noise robustness of various image-to-image translation models achieved by the GNI training method, all without incurring additional resource overhead.

Fig. 23 and 24 show the image conversion results under different image degradation and their combinations on tasks Cat→Dog and Photo→Sketch respectively. As can be seen here, systems trained with Gaussian noise injection produce pictures with more stable visual qualities compared with those baseline models.

### D.4 COMPARISON WITH DENOISING-BASED APPROACHES

An alternative to noise injection is to denoise the noisy input and feed these denoised images into an I2I model trained solely on clean images. In the context of photo-to-sketch translation, Table 13 and Fig. 25 compare the results of denoising-based approaches against the noise-injection method. Specifically, we employed CBM3D (Mäkinen et al., 2020) for images tainted by Gaussian noise and median filtering (using a $5 \times 5$ window) for those affected by Salt & Pepper noise. Visually, the quality of outputs from the noise-injection approach is comparable to those achieved through denoising. However, noise injection offers several advantages over the denoising-based pre-processing strategy:

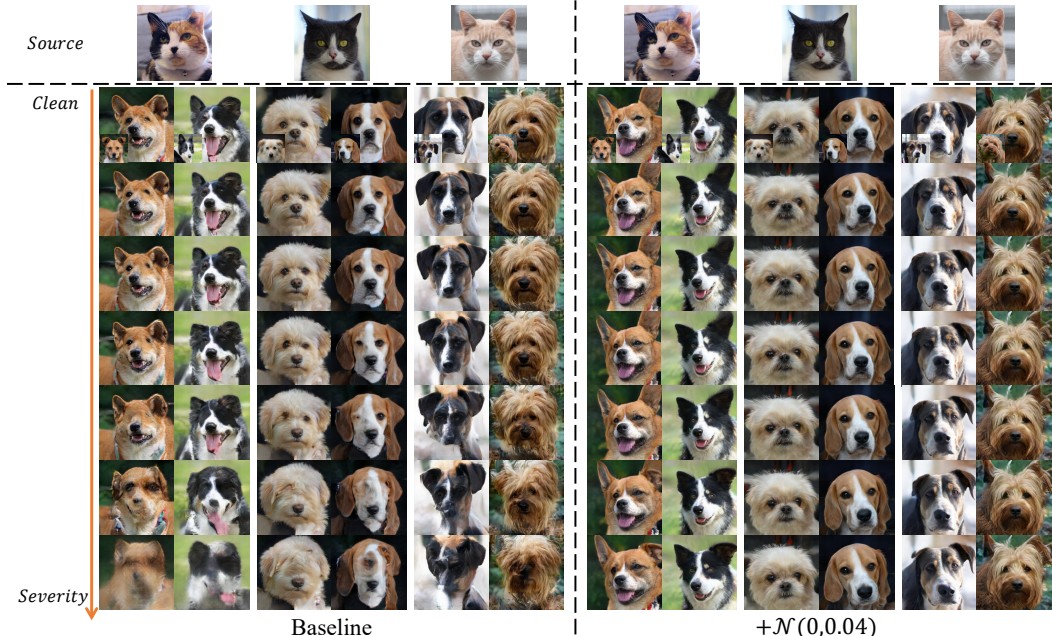

Figure 9: Comparison of reference-guided generation for Cat→Dog under the interference of Uniform noise. Each source image is guided using two reference images. On the left is the result obtained by the baseline (GP-UNIT), and on the right is the result obtained by applying noise injection to the baseline model.

1. **Generality**: Unlike denoising, which often requires precise noise details, the GNI mothod is broadly applicable without specific noise characterizations.

2. **Adaptive Robustness**: the GNI approach naturally conditions the I2I model to diverse perturbations, enhancing its resilience.

3. **Efficiency**: Bypassing the denoising step reduces computational overhead, offering faster translations, especially for high-resolution inputs.

4. **Scalability**: the GNI mothod's adaptability ensures relevance against evolving noise challenges without extensive re-engineering.

In essence, while denoising-based pre-processing attempts to "clean" the input, noise injection empowers the model to "understand" and "adapt" to the noise.

### D.5 OUT-OF-DOMAIN RESULTS

The Sketch Transformer was trained on a highly specific dataset. In this subsection, we demonstrate its capability in handling out-of-domain (OOD) face photos when the model is trained using noise injection. As depicted in Fig. 26, the model reliably translates OOD face photos into sketches, even in noise perturbations. However, compared to its performance on the in-domain CUFS test dataset, the model's resistance to noise slightly drops. This decrease in resilience becomes especially evident at higher noise levels, revealing visible distortions and suggesting an opportunity for future enhancements.

Regarding the two remaining image translation tasks, the employed training datasets boast even greater expanses. As a result, their capacity to handle data beyond their designated domains is a topic afforded less discourse. Instead, the focus leans towards assessing whether the data input model necessitates supplementary data processing and ancillary procedures.

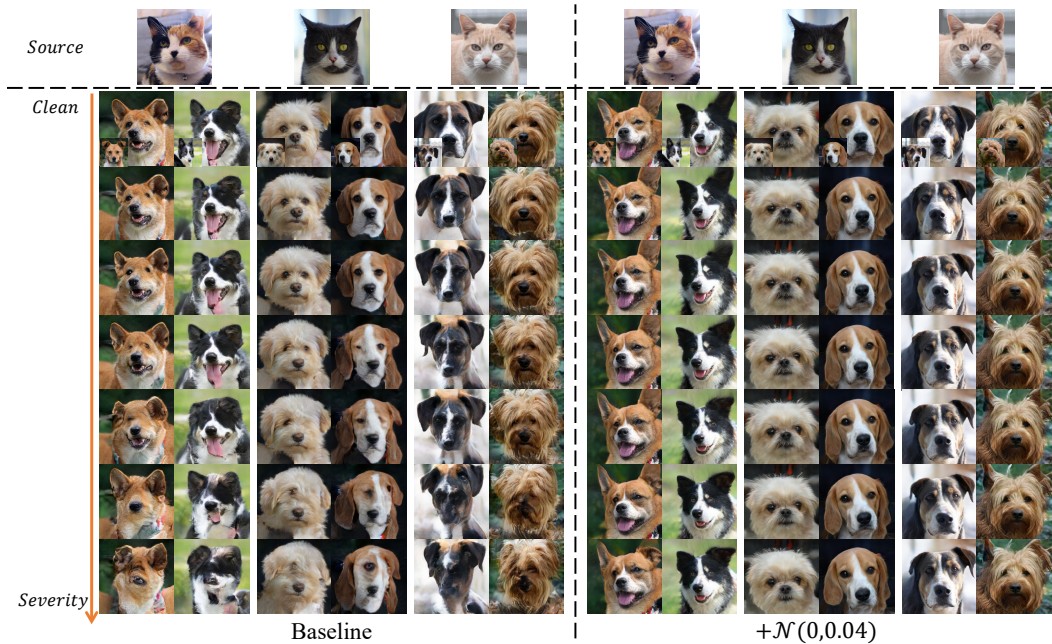

Figure 10: Comparison of reference-guided generation for Cat→Dog under the interference of Color noise. Each source image is guided using two reference images. On the left is the result obtained by the baseline (GP-UNIT), and on the right is the result obtained by applying noise injection to the baseline model.

## D.6 LIMITATIONS

In this subsection, we examine the limitations of the noise injection method. Our prior simulations predominantly employed uni-directional I2I models. Both visual and metric evaluations indicate that generated image quality may suffer, particularly in scenarios with clean or low-intensity noise.

Besides, the method's efficacy appears less consistent for bidirectional I2I translations. This is demonstrated in Table 14, which presents FID scores, and Fig. 27, which depicts qualitative results when noise injection is applied to CycleGAN (Zhu et al., 2017) for Horse→Zebra translation task. Notably, the Gaussian noise-injected model struggles with clean inputs, and despite better FID scores with noisy inputs, visual distortions, especially around the zebra's head and legs, are evident. It is, therefore, crucial to acknowledge that when applied to bi-directional I2I translation models, the GNI mothod cannot be considered a straightforward "plug-and-play" solution. Adaptations in network architectures and/or loss functions might be requisite to achieve desired results.

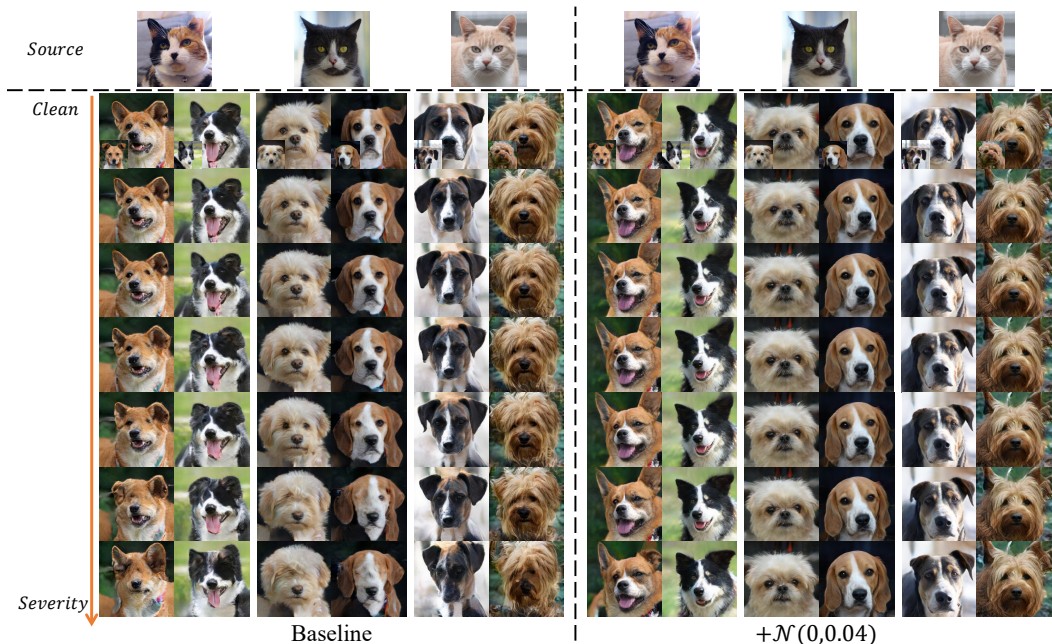

Figure 11: Comparison of reference-guided generation for Cat→Dog under the interference of Laplacian noise. Each source image is guided using two reference images. On the left is the result obtained by the baseline (GP-UNIT), and on the right is the result obtained by applying noise injection to the baseline model.

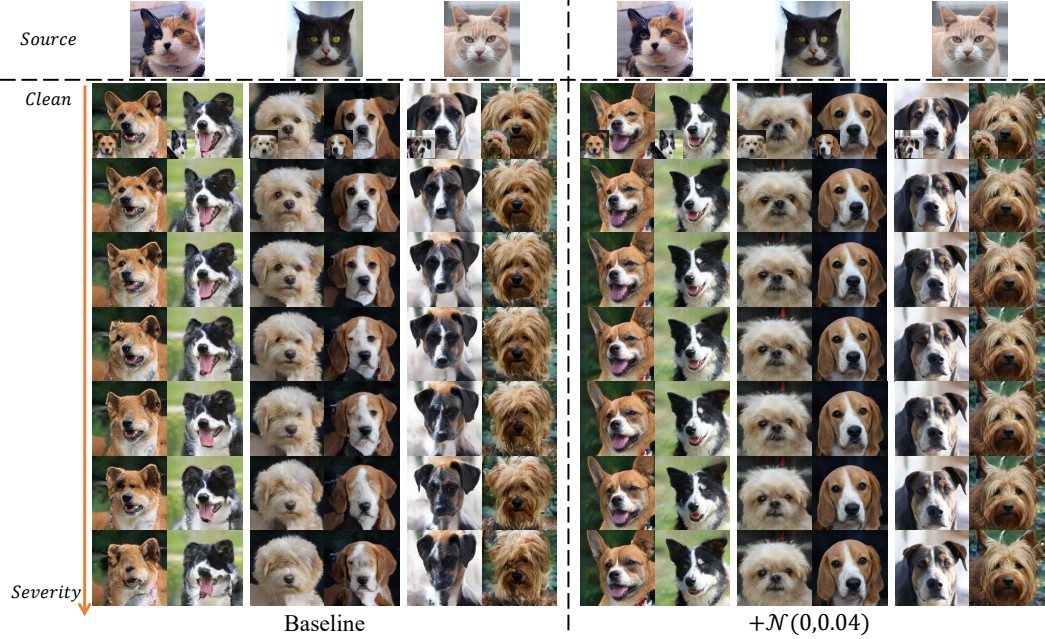

Figure 12: Comparison of reference-guided generation for Cat→Dog under the interference of Salt & Pepper noise. Each source image is guided using two reference images. On the left is the result obtained by the baseline (GP-UNIT), and on the right is the result obtained by applying noise injection to the baseline model.

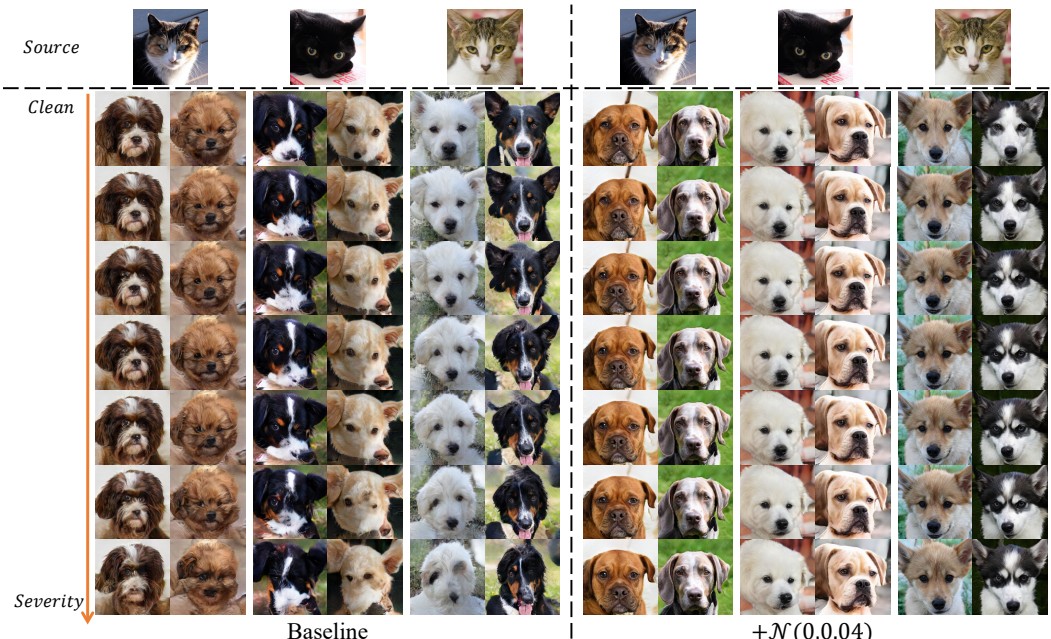

Figure 13: Comparison of latent-guided generation for Cat→Dog under the interference of Gaussian noise. Each source image is guided using two style latents randomly sampled from $\mathcal{N}(0, 1)$. On the left is the result obtained by the baseline (GP-UNIT), and on the right is the result obtained by applying noise injection to the baseline model.

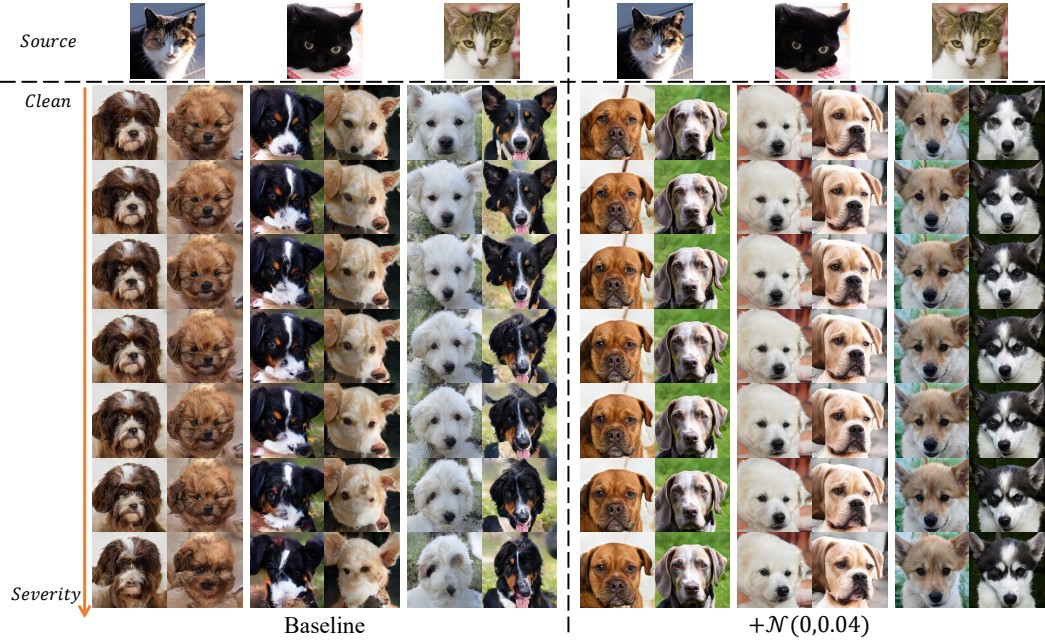

Figure 14: Comparison of latent-guided generation for Cat→Dog under the interference of Uniform noise. Each source image is guided using two style latents randomly sampled from $\mathcal{N}(0, 1)$. On the left is the result obtained by the baseline (GP-UNIT), and on the right is the result obtained by applying noise injection to the baseline model.

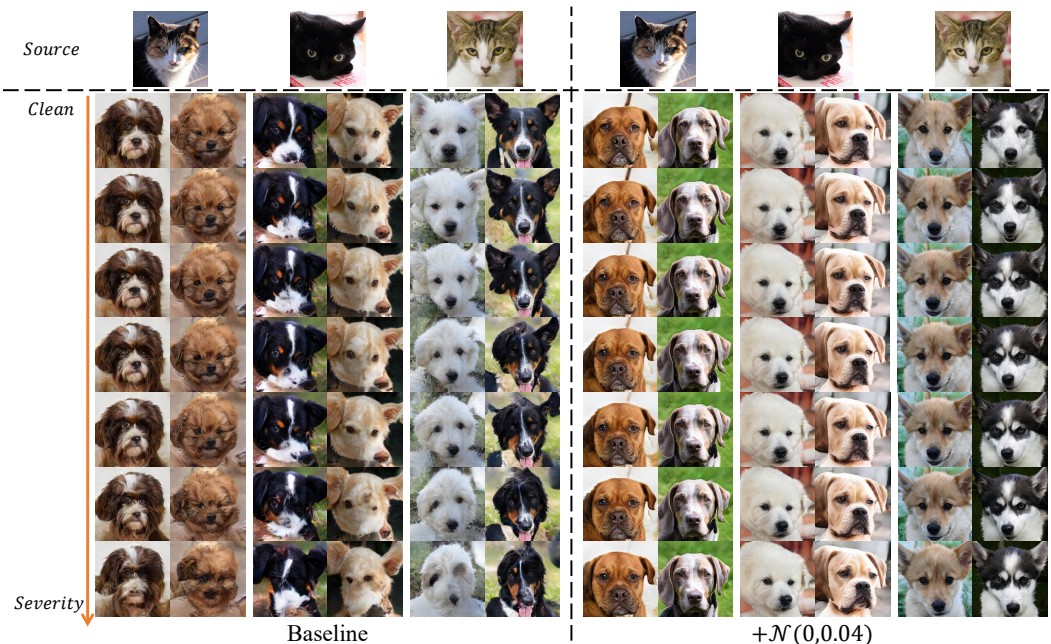

Figure 15: Comparison of latent-guided generation for Cat→Dog under the interference of Color noise. Each source image is guided using two style latents randomly sampled from $\mathcal{N}(0,1)$. On the left is the result obtained by the baseline (GP-UNIT), and on the right is the result obtained by applying noise injection to the baseline model.

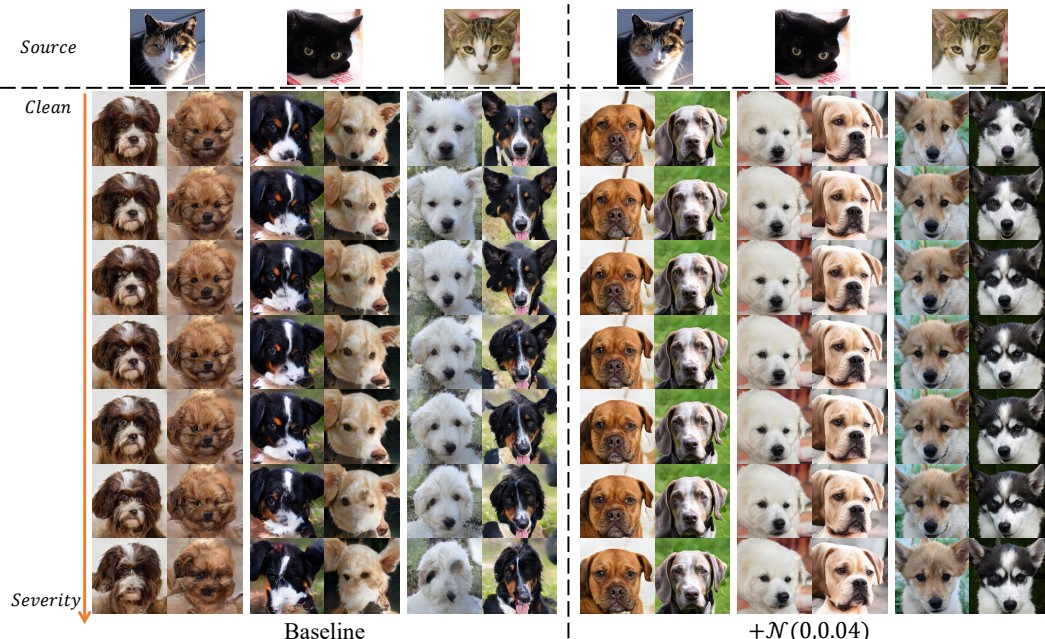

Figure 16: Comparison of latent-guided generation for Cat→Dog under the interference of Laplacian noise. Each source image is guided using two style latents randomly sampled from $\mathcal{N}(0,1)$. On the left is the result obtained by the baseline (GP-UNIT), and on the right is the result obtained by applying noise injection to the baseline model.

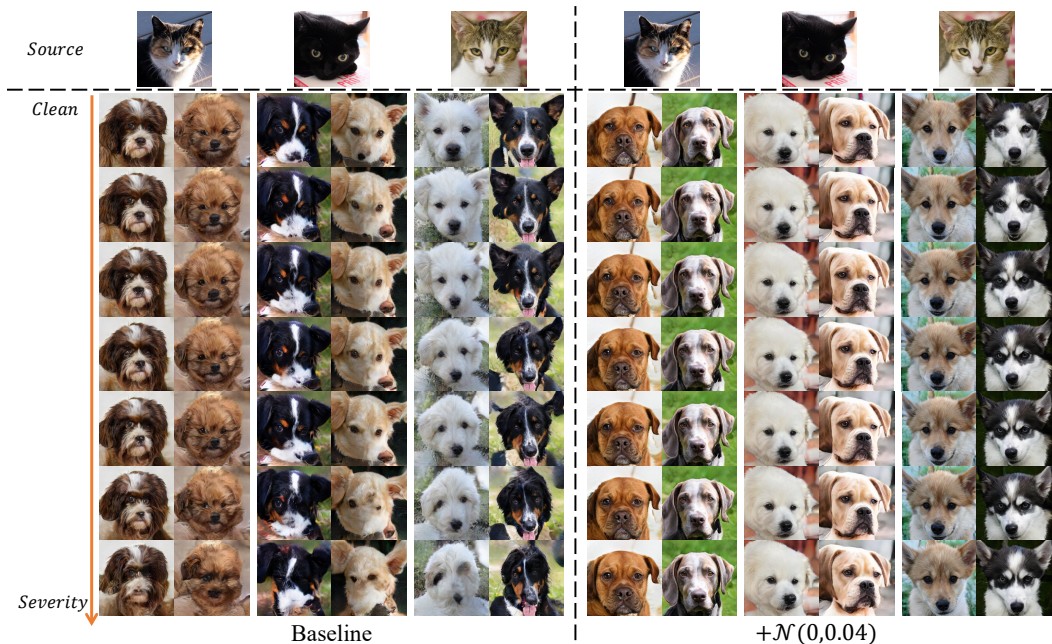

Figure 17: Comparison of latent-guided generation for Cat→Dog under the interference of Salt & Pepper noise. Each source image is guided using two style latents randomly sampled from $\mathcal{N}(0,1)$. On the left is the result obtained by the baseline (GP-UNIT), and on the right is the result obtained by applying noise injection to the baseline model.

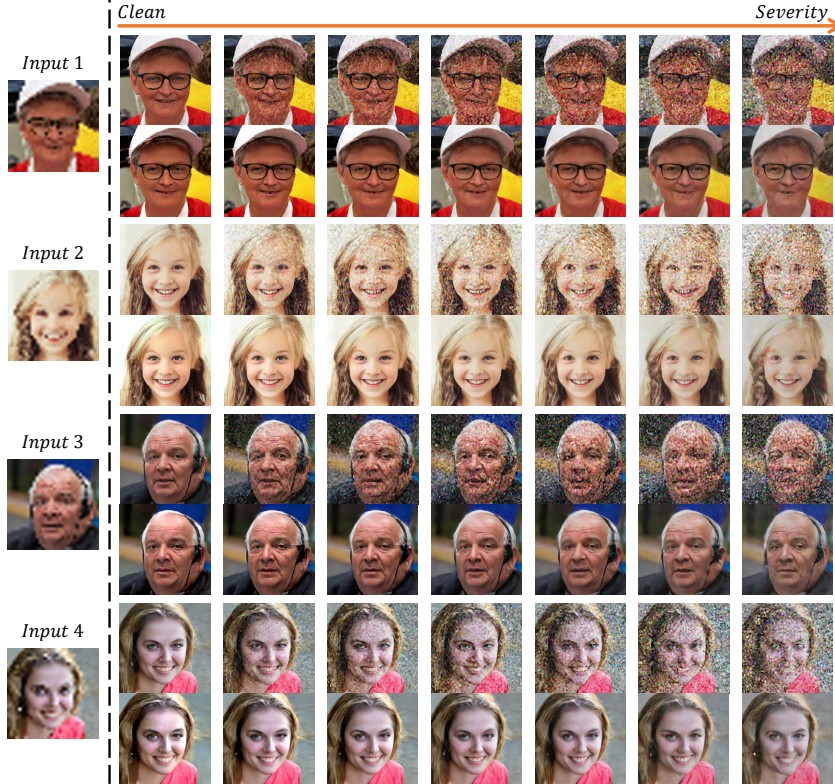

Figure 18: Comparison of Face super-resolution task under Gaussian noise corruption. Each example has the baseline in the 1st row and the $+\mathcal{N}(0, 0.04)$ in the 2nd. The same format is followed for results under other noise types in the subsequent figures.

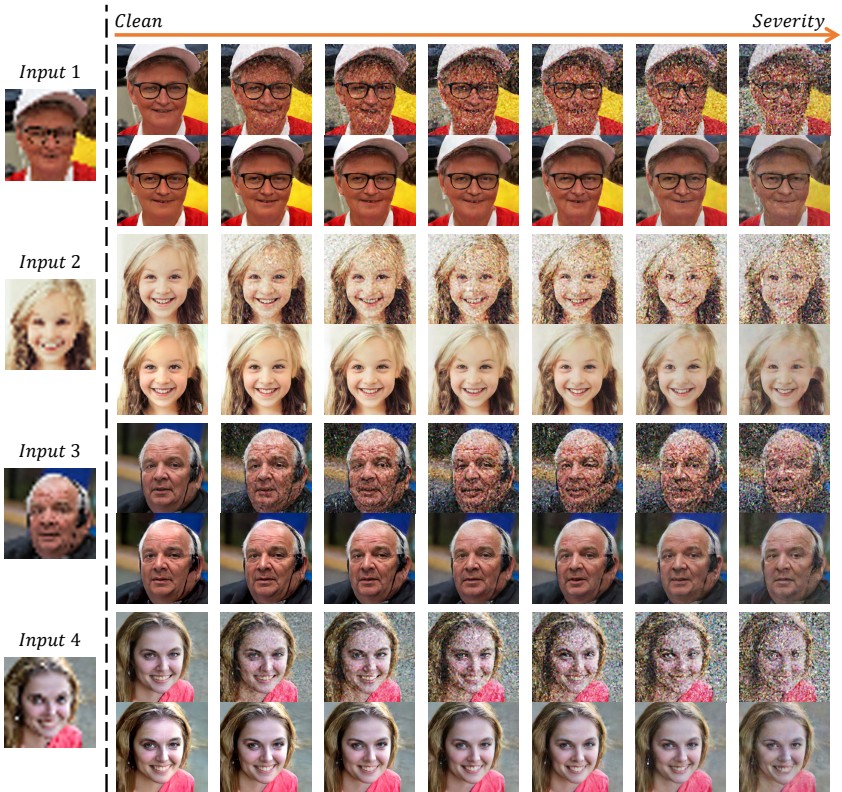

Figure 19: Face Super-Resolution comparison under Uniform noise interference.

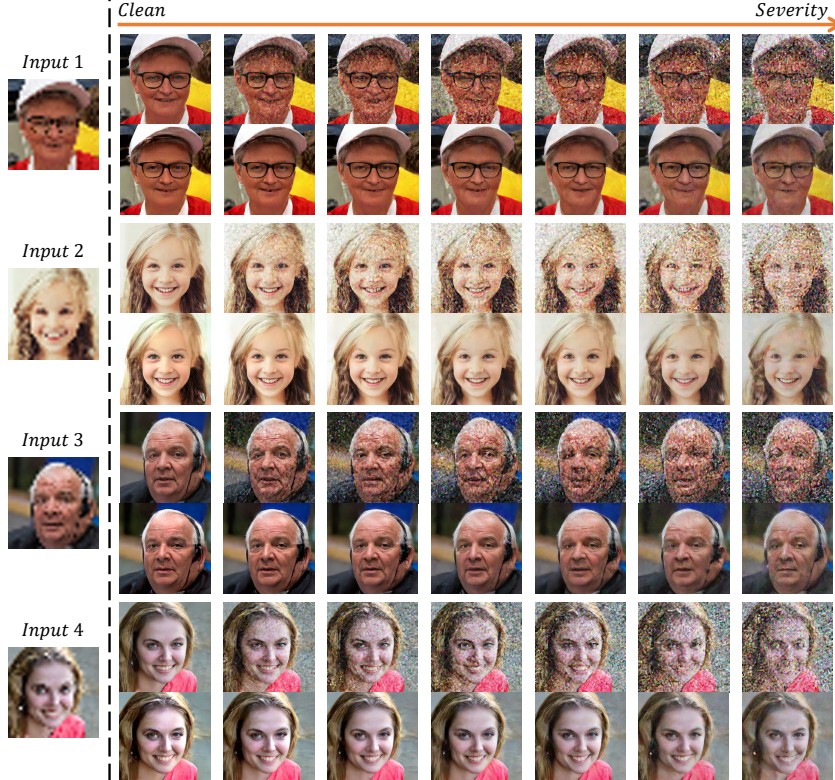

Figure 20: Face Super-Resolution comparison under Color noise interference.

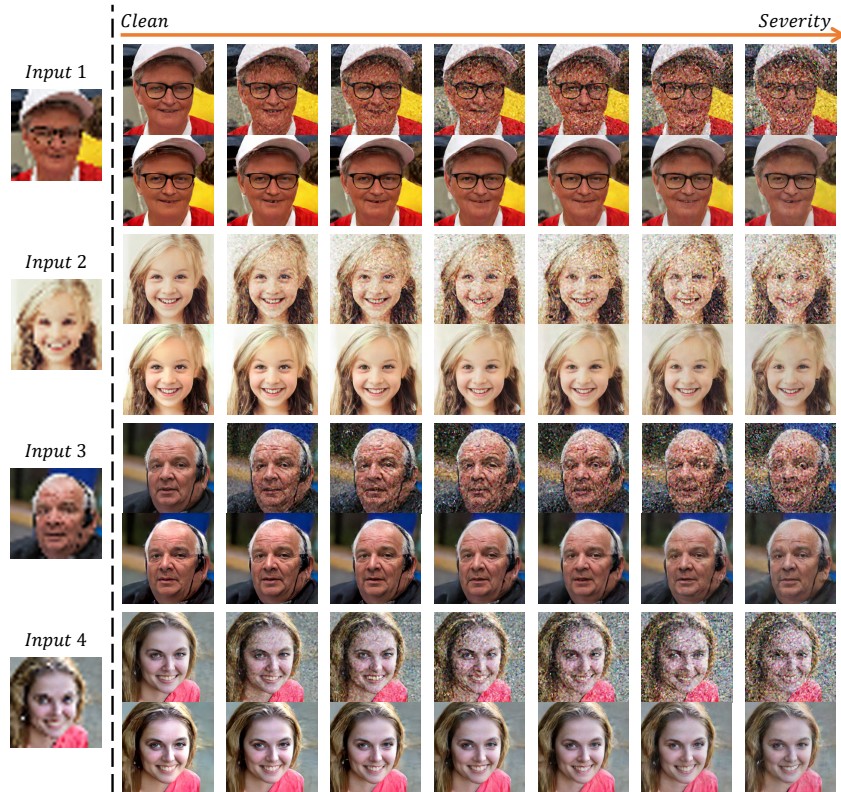

Figure 21: Face Super-Resolution comparison under Laplacian noise interference.

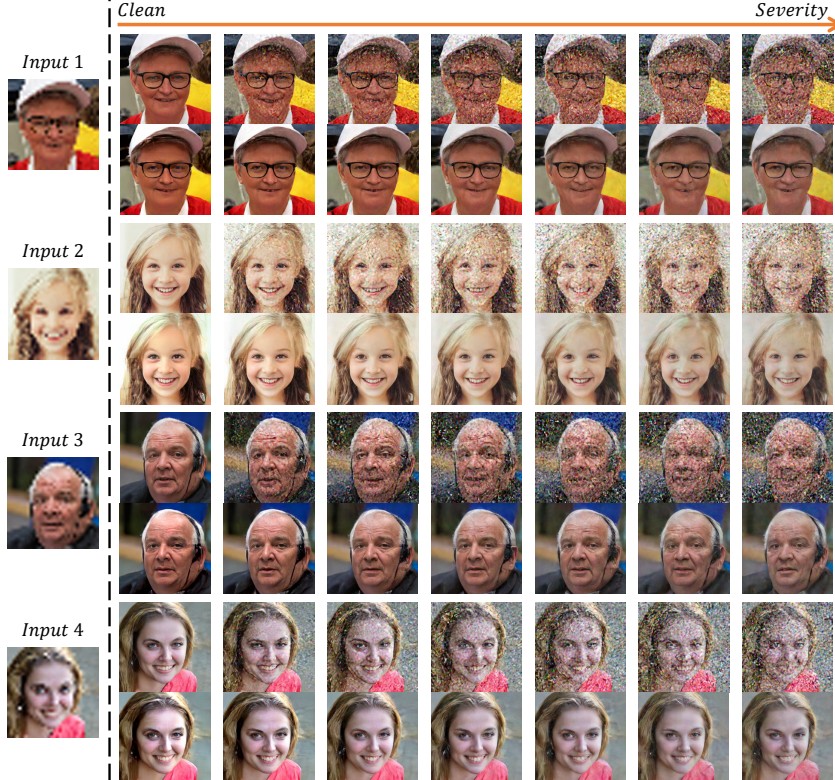

Figure 22: Face Super-Resolution comparison under Salt & Pepper noise interference.

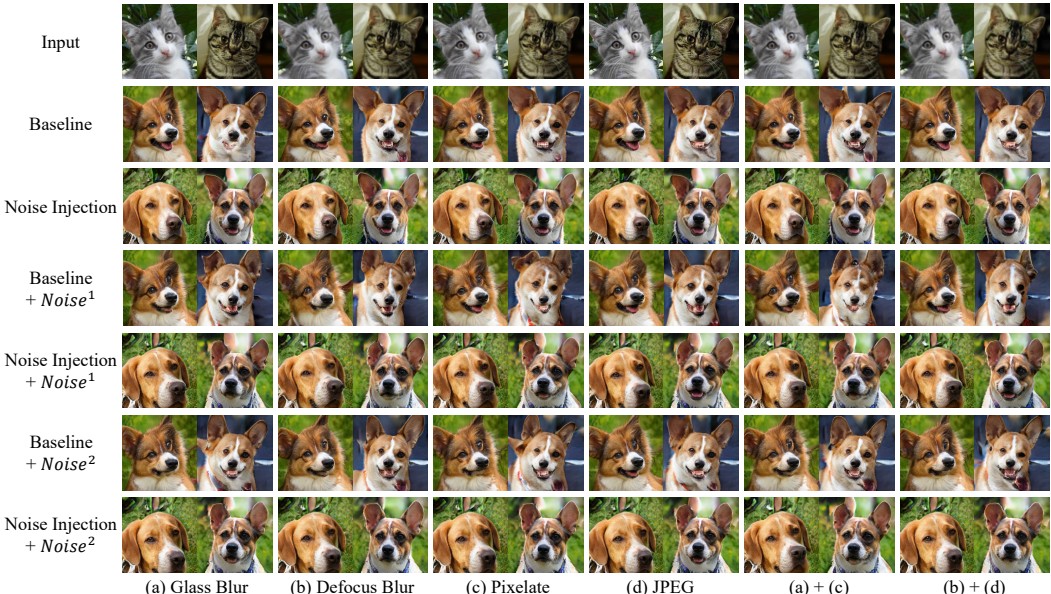

|  | (a) Glass Blur | (b) Defocus Blur | (c) Pixelate | (d) JPEG | (a) + (c) | (b) + (d) |

Rows: Input, Baseline, Noise Injection, Baseline + $Noise^1$, Noise Injection + $Noise^1$, Baseline + $Noise^2$, Noise Injection + $Noise^2$

Figure 23: Image translation results to multiple image degradation on the Cat→Dog image translation task (latent-guided). **Note:** Each source image is guided using two reference images. The selected noise intensity is S3 and other degradation intensity is S4.

Table 13: Comparison with Denoising-based Approach on FID scores for Photo→Sketch translation.

| Methods | Clean | Salt & Pepper | | | Gaussian Noise | | |
|---|---|---|---|---|---|---|---|
|  |  | S2 | S4 | S6 | S3 | S5 | S6 |
| Denoising-based | **31.47** | 61.11 | 114.32 | 241.08 | 36.46 | 41.02 | **45.35** |
| $+\mathcal{N}(0, 0.04)$ | 64.12 | **32.45** | **46.89** | **74.25** | **31.16** | **40.95** | 64.87 |

| | *Clean* | *Salt & Pepper Noise* | | | *Gaussian Noise* | | |

Rows: Input, Denoising, $+\mathcal{N}(0, 0.04)$

| | (a) | (b) | (c) | (d) | (e) | (f) | (g) |

Figure 25: Comparison of denoising-based pre-processing. First row: Input images. Column (a): Clean. Columns (b)-(d): Images corrupted with Salt & Pepper noise densities of 0.1, 0.2, and 0.3. Columns (e)-(g): Images corrupted with i.i.d. Gaussian noises of $\sigma_e^2$=0.04, 0.09, and 0.16. **Second row**: Results using denoising and baseline sketch transformer. Column (a): Unprocessed. Columns (b)-(d): Median filters applied. Columns(e)-(g): CBM3D (Mäkinen et al., 2020) denoising applied. **Third row**: Outputs from the models trained using Gaussian noise injection.

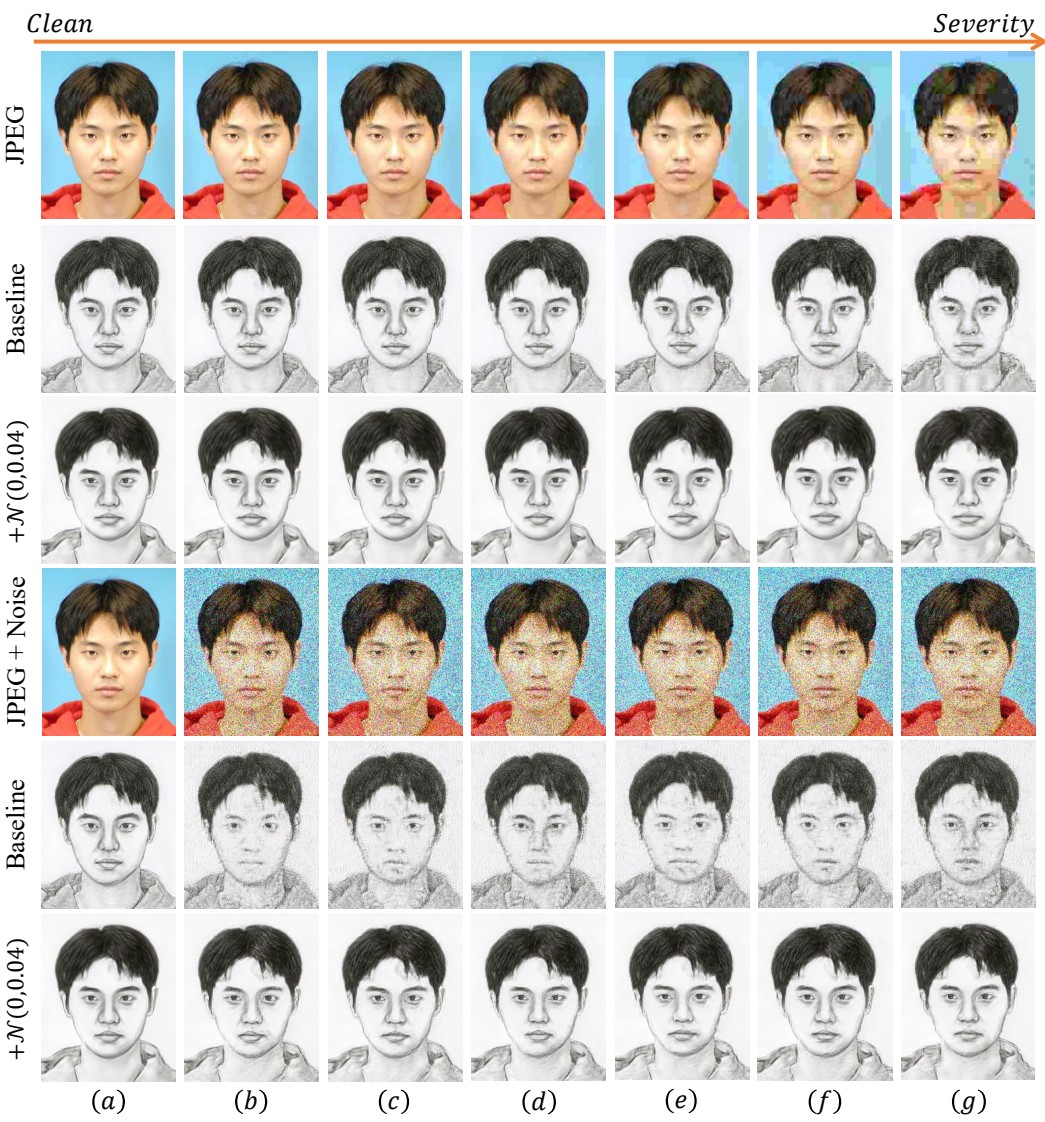

Figure 24: Image translation results to multiple image degradation on the Photo→Sketch image translation task.

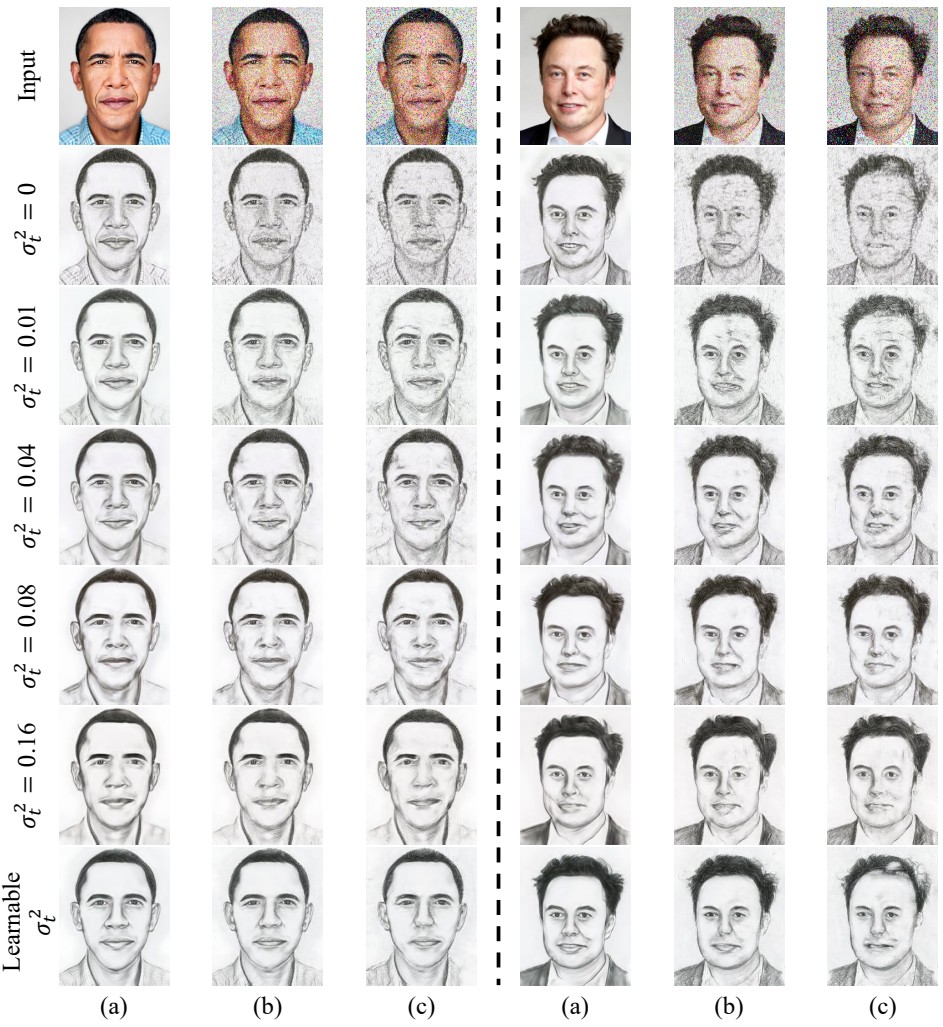

Figure 26: Out-of-domain results of **Photo→Sketch** task under the interference of Salt & Pepper noise and Uniform noise. (a) Input photos not disturbed by noise; (b) Input photo disturbed by Uniform noise of intensity S4; (c) Input photo disturbed by Salt & Pepper noise of intensity S4.

Table 14: FID comparison on the Horse→Zebra image translation task using Cycle-GAN model. We use the same FID calculation method as in (Zhu et al., 2017).

| $\sigma_e^2$ | 0 | 0.01 | 0.04 | 0.05 | 0.09 | 0.16 |
|---|---|---|---|---|---|---|
| Baseline | **76.92** | **92.15** | 118.29 | 135.64 | 147.34 | 180.82 |
| $+\mathcal{N}(0, 0.04)$ | 283.97 | 114.91 | **58.32** | **54.11** | **54.54** | **50.45** |

Figure 27: Failure results for Horse→Zebra translation in Cycle GAN model. Noise injection techniques might require customization for diverse models; otherwise, one may attain inferior results.

