# OpenReview forum: "On the Analysis of GAN-based Image-to-Image Translation with Gaussian Noise Injection"
_ICLR.cc/2024/Conference — ICLR 2024 poster_

### Official Review · Reviewer_YTga · 2023-10-24

**Soundness:** 3 good
**Presentation:** 3 good
**Contribution:** 2 fair
**Rating:** 6
**Confidence:** 3

**Summary:**

## Summary
* This paper studies the problem of noise-resistent I2I generation. Particularly, it studies the noise injection approach both theoretically and practically. The paper shows that the joint f-divergence does not change too abruptly when the source is polluted by Gaussian noise. Further, the paper finds optimal noise level for training when the source is Gaussian, and verify their results on Gaussian source and real images.

**Strengths:**

## Strength
* The result of optimal training level in __Corollary 1__ is useful to practical training of noise resistent generative model, if it is not limited to Gaussian source. It provides theoretical justification to a  intuitative practice.
* The empirical result show that their approach improves the noise-robustness in various cases.

**Weaknesses:**

## Weakness
* The discussion after __Lemma 1__ only convers non-Gaussian noise, but the majority of experiment is about non-Gaussian source. The gap of source distribution should at least be explicitly discussed in the main text not appendix, if not well addressed. Currently the proof of __Corollary 1__ seems to rely on Eq. 6 of __Lemma 1__, which holds only when the source is Gaussian. If I understand correctly, in that case, it becomes unreasonable to derive the optimal noise level $0.08$ for actual image generation, which is obviously non-Gaussian. If the theory is not connected to the experiment, then this paper becomes a little bit unconvincing. As no new empirical approach is proposed and the empirical results alone is not enough for accepting this paper.

**Questions:**

## Questions
* Is it possible to extend current theory (__Lemma 1__, __Theorem 2__, __Corollary 1__) to non-Gaussian source with known $\mu,\Sigma$? As Gaussian is the max-entropy distribution with known 1st and 2nd moment, can similar results be obtained?
* Is it possible to extend current theory (__Lemma 1__, __Theorem 2__, __Corollary 1__) to mixture of Gaussian source? As we can approximate any distribution, including natural image distribution with GMM, it would be much better if we can extend the theory to GMM, even for finite mixture. In this way, the distance between the current theory and empirical results on natural image can be reduced.
* Current analysis on joint distribution looks quite general, can it be extended into any conditional GAN, beyond I2I?
* What about more real-life noise, e.g. JPEG compression?

---

> ### Author Response · Authors · 2023-11-20
> **Reply to Reviewer YTga-1**
>
> Thanks for your insightful feedback and comments. Below are point-to-point reply of your comments.
> ### Q3-1:  Extention of  Lemma 1, Theorem 2, Corollary 1 to non-Gaussian source:
> ### Reply 3-1
> We appreciate the reviewer raising this point.
>
> For **Lemma** 1 and **Corollary** 1, as the derivations require closed-form expressions of KL divergences, extension to general non-Gaussian sources cannot be easily achieved. For **Theorem** 2, we generalize some results to non-Gaussian sources in **Theorem** 3.
> **Theorem 3:** Let $\\mathbf{X}$ be a $d$-dimensional random vector. Denote $\\theta(\\sigma^2\_t,\\sigma^2\_e\\boldsymbol{\\Sigma}\_{\\widetilde{e}})$ as the KL-divergence of non-Gaussian source signal for arbitrary noise, $\\theta\_g(\\sigma^2\_t,\\sigma^2\_e\\boldsymbol{\\Sigma}\_{\\widetilde{e}})$ denotes the special case of $\\theta(\\sigma^2_t,\\sigma^2_e\\boldsymbol{\\Sigma}_{\\widetilde{e}})$ when $\\boldsymbol{E}$ is Gaussian noise. Then for small $\\sigma^2_e$ with $\\sigma^2_e\\ll 1$, we have
>
> $\\theta(\\sigma^2\_t,\\sigma^2\_e\\boldsymbol{\\Sigma}\_{\\widetilde{e}})=\\theta\_g(\\sigma^2\_t,\\sigma^2\_e\\boldsymbol{\\Sigma}\_{\\widetilde{e}})+o(\\sigma^2\_e)$
>
> Thus, **Theorem 3** generalizes the result of **Theorem 2** to non-Gaussian sources, demonstrating that an I2I system's resilience to Gaussian noise ensures robustness against noises with the same covariance matrices, regardless of whether the input signals are Gaussian or non-Gaussian.
>
> We want to emphasize that adopting Gaussian signal models is a common theoretical approach even when practical signals are non-Gaussian. This allows deriving foundational insights that can be obscured in more complex settings. For example, in I2I, the widely used Fréchet Inception Distance (FID) assumes Gaussian distributions, despite natural images not being strictly Gaussian. Our work follows this principled methodology - establishing core theoretical results with Gaussian signals first.
>
> We agree extending the analysis to non-Gaussian sources is an important future direction. In fact, we take a first step in deriving Theorem 3 of the revised paper, showing the covariance-matching result holds for non-Gaussian signals when noise levels are small. However, a full generalization of all results is non-trivial and requires care to handle the increased complexity.
>
> By starting from Gaussian signal analysis, we lay a solid foundation upon which future research can build. This approach provides a rigorous basis for expanding into non-Gaussian settings, offering fundamental insights that can be further enriched to encompass more realistic signal models.

---

> ### Author Response · Authors · 2023-11-20
> **Reply to Reviewer YTga-2**
>
> ### Q 3-2: Extension of Lemma 1, Theorem 2 and Corollary 1 to Gaussian mixture model (GMM)
>
> Thanks for your suggestions of an extension to GMM signals. We include the discussions in Appendix of our revised paper.
>
> Consider a source signal $\\boldsymbol{x}$ represented by the Gaussian mixture model that $p\_{\\boldsymbol{X}}(\\boldsymbol{x})=\\sum\_{k=1}^{N}\\pi_k\\cdot\\mathcal{N}(\\boldsymbol{x}|\\boldsymbol{\\mu}\_k,\\boldsymbol{\\Sigma}\_k)$, $\\sum\_{k=1}^{N}\\pi_k=1$ and $\\pi_k>0$.  Although we cannot get a closed-form expression of the KL divergence for the GMM signal source, we can extend our results to its upper-bound, as listed below:
>
> - **Extention of Lemma 1**:  By using the convexity property of KL divergence, we have $D\_{KL}(\\hat{P}\_{\\boldsymbol{X}+\\boldsymbol{E}}\\|\\bar{P}\_{\\boldsymbol{X}+\\sigma\_t\\boldsymbol{N}})\\leq\\zeta(\\sigma^2\_t,\\boldsymbol{\\Sigma}\_e)$.
> $$
> \begin{aligned}
> \\zeta(\\sigma^2\_t,\\boldsymbol{\\Sigma}\_e)&= \\sum_{k=1}^{N}\\pi\_k\\cdot D\_{KL}(\\hat{P}\_{\\boldsymbol{X}\_k+\\boldsymbol{E}}\\|\\bar{P}\_{\\boldsymbol{X}\_k+\\sigma\_t\\boldsymbol{N}})\=\\sum_{k=1}^{N}\\pi_k\\cdot\\rho\_{k}(\\sigma\_t^2,\\boldsymbol{\\Sigma}\_e),
> \end{aligned}
> $$
> where $\\boldsymbol{X}$ is d-dimensional random variable of the GMM represented source signal $\\boldsymbol{x}$, $\\boldsymbol{X}\_k$ is the $k$-th Gaussian distribution with its weight $\pi_k$ for GMM and $\rho_{k}(\sigma_t^2,\boldsymbol{\Sigma}_e)$ is the KL-divergence of $k$-th Gaussian distribution for arbitrary noise.
> The above equation implies that for GMM fitted source signal, its KL-divergence concerning arbitrary noise is upper-bounded by the weighted sum of the KL-divergence of $N$ Gaussian source signals with respect to arbitrary noise.
>
> - **Extension of Theorem 2**:  For $\\zeta(\\sigma^2\_t,\\boldsymbol{\\Sigma}\_e)$ given above, when $\\boldsymbol{\\Sigma}\_e=\\sigma^2\_e\\boldsymbol{\\Sigma}\_{\\tilde{e}}$,   $\\zeta(\\sigma^2\_t,\\boldsymbol{\\Sigma}\_e)$ is also convex in $\sigma^2_e$ as each  $\rho_{k}(\sigma_t^2,\boldsymbol{\Sigma}_e)$ is convex in $\sigma^2_e$. Likewise, $\zeta(\sigma^2_t,\boldsymbol{\Sigma}_e)<\zeta(0,\boldsymbol{\Sigma}_e)$ when $\boldsymbol{\Sigma}_e>\sigma^2_t/2$.
>
> - **Extension of Corollary 1**: Under the assumptions of Corollary 1, the optimal solution
> $$
> \\bar{\sigma}\_{t,o}^2 = \\arg \\min\_{\\sigma\_t^2} \\mathbb{E}\_{\\sigma\_e^2\\sim\\mathcal{U}(0,\\lambda\_{\\max})}\\left\\{\\zeta(\\sigma^2\_t,\\sigma\_e^2\\boldsymbol{I}\_d)\\right\\}
> $$
> is still $\\bar{\\sigma}\_{t,o}^2=\\frac{1}{2}\\lambda\_{\\max}$, which remains the same as that of the Gaussian source.
>
>
> ### Q3-3: Comments on optimal solution of  $\sigma^2_t$ for non-Gaussian signal source.
> ### Reply 3-3:
>
> We appreciate your comments on this issue. As you have pointed out, many image signals follow GMM. In the above reply **Q3-2**,  we have theoretically extended the result to Gaussian Mixture Model (GMM) signals, which better represent images. For the upper bound of the KL divergence with a GMM source, we show the optimal solution of $\\sigma^2\_t$ is still $\\frac{\\lambda\_{\\max}}{2}$.
>
> Besides, for face images to sketch task, the ablation study on $\\sigma^2\_t$ also showed that the average FID is minimized when $\\sigma^2\_t=0.08$ given a specific $\\lambda\_{\\max}=0.16$.  Thus, the empirical results agree well with our theoretical results.
>
> |    Noise Type   |   $\sigma^2_{t}=0$   |  $\sigma^2_{t}=0.01$  |   $\sigma^2_{t}=0.04$|     $\sigma^2_{t}=0.08$   |   $\sigma^2_{t}=0.16$ | Learnable |
> |:---------------:|:------:|:------:|:-----:|:---------:|:-----:|:---------:|
> |  Gaussian Noise | 182.55 |  78.89 | 42.17 | **36.18** | 36.64 |   59.19   |
> |  Uniform Noise  | 199.17 | 101.68 | 45.37 | **36.79** | 36.89 |   64.81   |
> | Laplacian Noise | 152.09 |  64.12 | 39.17 | **36.99** | 37.01 |   52.41   |
> ------
>
> In summary, while **Corollary** 1 is based on Gaussian signal source assumption, we have extended it to GMM signals and empirically verified the optimal noise results on real image data. This highlights the relevance of the analysis beyond Gaussian sources.
>
> ### Q3-4: Extention to other applications of conditional GANs
> ### Reply 3-4:
> We thank the reviewers for this suggestion. This would be considered in our future work for other applications of conditional GAN.

---

> ### Author Response · Authors · 2023-11-20
> **Reply to Reviewer YTga-3**
>
> ###  Q3-5: Performance on real-life noise
> ### Reply 3-5:
> Following the reviewers' advice,  we expanded our simulations to include signal-dependent noise and corruptions using the ImageNet-C corruption models. Specifically, we adopted the widely-used image degradation model $\\boldsymbol{\\hat{x}}=\\boldsymbol{A}(\\boldsymbol{x})+\\boldsymbol{n}$, where $\\boldsymbol{x}$ and $\\boldsymbol{\\hat{x}}$ denote clean and corrupted images, respectively, $\\boldsymbol{A}$ represents an operator (e.g., JPEG compression, blurring, pixelation, etc.), and $\\boldsymbol{n}$ denotes the noise component.
>
> Our expanded simulations covered shot noise, speckle noise, JPEG compression, pixelation, blurring, and combinations of these distortions. Key findings include:
>
>  - When signal-dependent noise $\\boldsymbol{n}$ is present alone, models trained with Gaussian noise injection exhibited substantial improvements over baseline models trained solely on clean images, akin to the results for signal-independent noises.
>  - For blurring, pixelation and JPEG compression only (with $\\boldsymbol{n}=0$),  models trained with Gaussian noise injection are inferior to the baselines in some cases.
>  - When both $\\boldsymbol{A}$ and $\\boldsymbol{n}$ were present, models trained with Gaussian noise injections consistently demonstrated significant enhancements over the baseline models. This observation aligns with practical scenarios where both $\\boldsymbol{A}$ and $\\boldsymbol{n}$ coexist in degradation models, underscoring the efficacy of Gaussian noise injection.
>
> Notably, in cases where the corruption model excluded the noise component, the addition of a small amount of noise to the input images proved effective in producing outputs with stable quality. These findings provide further insights into the robustness and potential of Gaussian noise injection in I2I for various degradations.

---

> ### Author Response · Authors · 2023-11-20
> **Reply to Reviewer YTga-4**
>
> | Task | Cat->Dog(Latent) |
> |      Type     | Metric |  Method  |   Clean   |     S1    |     S2    |     S3    |     S4    |     S5    |     S6    |
> |:-------------:|:------:|:--------:|:---------:|:---------:|:---------:|:---------:|:---------:|:---------:|:---------:|
> |   Shot Noise  |   FID  | Baseline | **22.82** | **23.42** |   23.74   |   24.31   |   26.09   |   30.91   |   37.51   |
> |               |        | $+\mathcal{N}(0, 0.04)$ |   25.04   |   24.03   | **23.44** | **22.85** | **22.39** | **21.76** | **21.88** |
> |               |   KID  | Baseline |  **9.67** |   10.86   |   11.09   |   11.59   |   13.49   |   18.27   |   24.51   |
> |               |        | $+\mathcal{N}(0, 0.04)$ |    9.89   |  **9.31** |  **8.98** |  **8.46** |  **8.12** |  **7.79** |  **7.91** |
> | Speckle Noise |   FID  | Baseline | **22.82** | **23.36** |   24.25   |   25.41   |   26.98   |   29.35   |   31.53   |
> |               |        | $+\mathcal{N}(0, 0.04)$ |   25.04   |   24.25   | **23.02** | **22.37** | **22.08** | **21.99** | **21.91** |
> |               |   KID  | Baseline |  **9.67** |   10.69   |   11.64   |   12.97   |   14.53   |   16.97   |   19.15   |
> |               |        | $+\mathcal{N}(0, 0.04)$ |    9.89   |  **9.46** |  **8.61** |  **8.37** |  **7.96** |  **8.07** |  **7.92** |
> |   Glass Blur  |   FID  | Baseline | **22.82** | **22.85** | **23.91** | **23.04** | **23.12** | **23.51** | **23.98** |
> |               |        | $+\mathcal{N}(0, 0.04)$ |   25.04   |   25.06   |   25.09   |   25.18   |   24.89   |   24.61   |   24.51   |
> |               |   KID  | Baseline |  **9.67** |  **9.69** |  **9.82** |  **9.86** |   10.06   |   10.44   |   10.56   |
> |               |        | $+\mathcal{N}(0, 0.04)$ |    9.89   |    9.88   |    9.92   |   10.03   |  **9.98** |  **9.91** |  **9.77** |
> |  Defocus Blur |   FID  | Baseline | **22.82** | **23.08** | **23.41** | **24.71** | **25.71** |   26.52   |   27.13   |
> |               |        | $+\mathcal{N}(0, 0.04)$ |   25.04   |   25.92   |   26.01   |   26.02   |   25.91   | **25.69** | **25.45** |
> |               |   KID  | Baseline |  **9.67** |  **9.81** |  **9.98** |   10.71   |   11.31   |   11.97   |   12.45   |
> |               |        | $+\mathcal{N}(0, 0.04)$ |    9.89   |   10.44   |   10.61   | **10.67** | **10.59** | **10.48** | **10.29** |
> |    Pixelate   |   FID  | Baseline | **22.82** | **22.89** | **23.31** | **23.42** | **23.85** | **23.91** | **24.01** |
> |               |        | $+\mathcal{N}(0, 0.04)$ |   25.04   |   25.29   |   25.84   |   26.01   |   25.81   |   25.71   |   25.69   |
> |               |   KID  | Baseline |  **9.67** |  **9.72** |  **9.82** | **10.06** | **10.21** | **10.44** |   10.91   |
> |               |        | $+\mathcal{N}(0, 0.04)$ |    9.89   |   10.03   |   10.35   |   10.51   |   10.54   |   10.45   | **10.42** |
> |      JPEG     |   FID  | Baseline | **22.82** | **22.91** | **22.95** | **23.01** | **23.56** | **24.21** |   25.46   |
> |               |        | $+\mathcal{N}(0, 0.04)$ |   25.04   |   25.19   |   24.94   |   25.21   |   25.41   |   25.43   | **25.45** |
> |               |   KID  | Baseline |  **9.67** |  **9.79** |   10.01   |   10.07   |   10.35   |   10.53   |   12.24   |
> |               |        | $+\mathcal{N}(0, 0.04)$ |    9.89   |   10.03   |  **9.77** |  **9.96** | **10.11** | **10.25** | **10.31** |
>
> ---
> | Cat->Dog(Latent) | FID | Robustness evaluation to multiple image degradations  |
> |     Setting     | Glass Blur | Defocus Blur |  Pixelate |    JPEG   | Glass Blur + Pixelate | Defocus Blur + JPEG |
> |:---------------:|:----------:|:------------:|:---------:|:---------:|:---------------------:|:-------------------:|
> |     Baseline    |    23.12   |     25.71    |   23.85   |   23.26   |         26.35         |        25.93        |
> | Noise Injection |    24.89   |     25.91    |   25.81   |   25.41   |         25.86         |        25.99        |
> |   Baseline $+Noise^1$   |    26.98   |     28.99    |   27.81   | 26.75     |         28.51         |        29.51        |
> |   Baseline $+Noise^2$   |    26.24   |     27.31    |   26.56   | 26.01     |         27.11         |        27.44        |
> |  Noise Injection $+Noise^1$  |  **22.21** |   **22.75**  | **22.53** | **22.41** |       **22.44**       |      **22.61**      |
> |  Noise Injection $+Noise^2$  |    22.67   |     22.84    |   22.81   |   22.49   |         23.05         |        22.75        |

---

> > ### Comment · Reviewer_YTga · 2023-11-21
> >
> > Thanks for the rebuttal, I think the GMM extension is quite meanful for general noise. I raise my rating to accept.

---

> > > ### Author Response · Authors · 2023-11-22
> > > **Appreciation for Changed Rating from Reviewer YTga**
> > >
> > > Thank you Reviewer YTga for changing your rating of our paper from 5 to 6 - we greatly appreciate you taking the time to reassess our work.

---

### Official Review · Reviewer_Ceu7 · 2023-11-04

**Soundness:** 3 good
**Presentation:** 3 good
**Contribution:** 3 good
**Rating:** 6
**Confidence:** 3

**Summary:**

This work provides a robust theoretical framework elucidating the role of Gaussian noise injection in I2I translation models. They address critical questions on the influence of noise variance on distribution divergence, resilience to unseen noise types, and optimal noise intensity selection.

**Strengths:**

- This paper thoroughly investigates the Gaussian noise in I2I area from both the theory and experiment.
- Extensive experiments and analysis make the effects of the Gaussian noise more clear to us.

**Weaknesses:**

- Analysis of the Gaussian noise has been widely investigated, the authors should give some comparison or analysis about them in related works.

**Questions:**

- How do you quantize the real and predictive distribution? Furthermore, how do you measure the diff between them?

---

> ### Author Response · Authors · 2023-11-20
> **Reply to Reviewer Ceu7**
>
> Thanks for your insightful feedback and comments. Below are point-to-point reply of your comments.
>
> ### Q2-1: Measurement of the real and predictive distributions
> ### Reply to 2-1:
> In our theoretical analysis, we use $f$-divergence to quantify the disparity between two distributions, as illustrated in Figure 6 in the appendix. However, we acknowledge that $f$-divergence can be computationally challenging to calculate. Therefore, in our practical experiments, we employ popular I2I quality metrics like FID, KID, LPIPS, among others, to effectively measure the dissimilarity between the generated (predictive) distribution and the actual distribution. These metrics assess the quality of generated samples by evaluating the separation between the characteristic distribution of the generated models and that of authentic samples. Specifically, in Figure 5 of the main paper, we presented the FID results for face image-to-sketch translation tasks. The use of visual quality metrics in our experiments aligns well with our theoretical analysis of KL divergence, providing a comprehensive evaluation of the model's performance in terms of distribution dissimilarity.
>
> We hope this clarifies our approach to quantifying the distinction between the real and predictive distributions.
>
> ### Q2-2: Comments on Related-work on Gaussian noise analysis
> ### Reply 2-2:
> We appreciate your comment and would like to clarify that we have indeed addressed the context of Gaussian noise injection in related work, specifically in Section 2. In our related work section, we emphasize that existing theoretical analyses of Gaussian noise injection primarily focus on robustness against adversarial attacks in image classification and recognition tasks. These tasks involve discrete output variables, making their derivations less directly applicable to image-to-image (I2I) translation tasks, which operate in the domain of continuous image generation.
>
> The key difference lies in the discrete nature of classification outputs compared to the more intricate image generation performed by I2I models. While some prior studies have explored noise injection in GAN-based I2I approaches, they often treat it as an empirical technique without in-depth theoretical analysis. To the best of our knowledge, our paper represents a pioneering effort in providing a comprehensive theoretical framework for understanding the role of noise injection in robust I2I tasks. For a more detailed discussion of these issues, we kindly refer the reviewer to Section 2 (related work) in the paper. We hope this clarifies the context and positioning of our research within the existing literature.

---

> > ### Comment · Reviewer_Ceu7 · 2023-11-23
> > **Response to the rebuttal**
> >
> > Hi, authors:
> >   Thanks for your rebutal.

---

### Official Review · Reviewer_SHt1 · 2023-11-08

**Soundness:** 2 fair
**Presentation:** 3 good
**Contribution:** 1 poor
**Rating:** 3
**Confidence:** 5

**Summary:**

This paper provides a robust theoretical framework for understanding the role of Gaussian noise injection in image-to-image (I2I) translation models. The key contributions of this work include:

(i) Analyzing the influence of noise variance on distribution divergence and resilience to unseen noise types.

(ii) Proposing a method to choose an optimal training noise level for consistent performance in noisy environments.

(iii) Connecting f-divergence and score matching to explain the impact of Gaussian noise on aligning probability distributions.

**Strengths:**

(i) The writing is clear and easy to follow. The presentation is well-dressed.

(ii) This paper conducts a detailed theoretical analysis in Section 3 to understand the role of Gaussian noise injection in I2I models from the perspective of alignment distribution.

(iii) To validate the conjecture proposed in this paper, the authors conduct sufficient experiments including three types of I2I models, i.e., cat to dog, photo to sketch, and human face super-resolution, covering five types of noise: Gaussian, Uniform, Color, Laplacian, and Salt & Pepper.

**Weaknesses:**

(i) This paper has a fatal theoretical flaw, i.e., the authors think too simply and naively about real-world degradations. They set up additive Gaussian noise in the training phase and five synthetic degradations (Gaussian, Uniform, Color, Laplacian, and Salt & Pepper.) in the testing phase. However, the real-world degradations are much more complex and fundamentally different from these degradations. To be specific, as for the noise term, the real-camera raw noise produced by photon sensing comes from multiple sources (e.g., short noise, thermal, noise, dark current noise, etc.) [1, 2, 3] and is further affected by the in-camera signal processing pipeline to become spatio-chromatically correlated. The real-camera noise contains both signal-dependent and -independent terms. However, the authors only consider very naive and simple signal-independent noise types, which is far from their motivation because the degradation patterns they studied simply do not exist in real-world images. Similar to the face super-resolution problem, the blur kernel cannot be explicitly estimated.

[1] CycleISP: Real Image Restoration via Improved Data Synthesis. In CVPR 2020

[2] Variational denoising network: Toward blind noise modeling and removal. In CVPR 2019

[3] Dual adversarial network: Toward real-world noise removal and noise generation. In ECCV 2020

If you want to continue studying this topic, which I think is valuable too, I suggest you to conduct experiments in real degraded images. Here are some suggested datasets:

[4]  A high-quality denoising dataset for smartphone cameras. CVPR 2018

[5] Deep retinex decomposition for low-light enhancement. BMVC 2018

[6] Ntire 2020 challenge on real-world image super-resolution: Methods and results. In CVPRW 2020

(ii) This work does not have any technical contributions. Training with additive Gaussian noise has been studied in the image denoising I2I task for a long time. It is a very common setting. The robustness of noise has also been validated by many prior works. This paper does not propose any new method.

(iii) The idea of adding Gaussian noise and aligning distribution is highly similar to a prior work [7] that also studies real-camera degradations in I2I tasks. There is no discussion or comparison.

[7] Learning to Generate Realistic Noisy Images via Pixel-level Noise-aware Adversarial Training. In NeurIPS 2021.

(iv) Code and pre-trained models are not submitted. The reproducibility cannot be checked.

**Questions:**

(i) For the theoretical noise analysis part, if considering signal-independent and -dependent noise terms, what will happen to the analysis? How does it change?

(ii) To measure the domain discrepancy, why not using the metric PSNR Gap proposed by the prior work DANet [3]?

[3] Dual adversarial network: Toward real-world noise removal and noise generation. In ECCV 2020

---

> ### Author Response · Authors · 2023-11-20
> **Reply to Reviewer SHt1-1**
>
> Dear Reviewers,
> Thank you for your time and valuable feedback. Key updates in our revised manuscript include:
>
> - 1. **Theoretetical analysis:** Introduction of Theorem 3 into the main text, extending the analysis to non-Gaussian sources. Additionally, the results and discussions on Gaussian Mixture Models (GMM) sources are presented in Appendix B.6.
>
> - 2. **Simulation results:** Expanded simulations with variaous image corruptions of signal-dependant noises, blur, JPEG compression etc. have been incorporated for a comprehensive evaluation of the models' performances.
>
> - 3. **Reference list:** Addition of the comparison with denoising-based approaches in Appendix D.4, featuring the inclusion of references recommended by the reviewers.
>
> In summary, we have enhanced the theoretical analysis to handle non-Gaussian signals and significantly expanded the experiments using diverse noise types and image distortions. We believe these augmentations have led to a stronger, more complete manuscript. Thank you again for your insightful suggestions. Below are our point-to-point replies to your comments.
>
> ### Q1-1:  Comment on "fatal theoretical flaw" and real-world degradations
> ### Reply1-1:
> We acknowledge the significance of modeling real-world image distortions and the development of efficient denoising algorithms. However, while crucial, these areas fall outside our work's primary focus. Our objective is to establish a theoretical framework that guarantees robustness in image-to-image (I2I) translation models when Gaussian noise injection is used during training. We address the concerns regarding the simplicity of our noise models about real-world degradations with our theoretical and empirical findings.
>  1. **Theoretical Framework and Applicability:**  In Theorems 2 and 3 of our revised paper, we base our theoretical analysis on the assumption of signal-independent noise with a finite covariance matrix. These foundational assumptions are critical yet do not limit the applicability to potential real-world noise distributions. Common signal-independent noises, such as Gaussian, erasure, and impulsive noises in image transmission, and uniform noise in low-resolution quantization, are prevalent in various applications. The core insight from Theorems 2 and 3 is that certifying robustness to a broader class of signal-independent noise is feasible if an I2I system is designed to withstand Gaussian noise with the same covariance matrix.
>  2. **Comparative Analysis with Related Works:** Our research aligns with significant findings in image classification, notably those presented by Cohen, Rosenfeld, and Kolter (2019) in "Certified adversarial robustness via randomized smoothing." Our study provides theoretical and empirical evidence of robustness in I2I translation models under noisy conditions, akin to the certified robustness against common corruptions and adversarial attacks discussed in their work.
>
>  3.  **Extended Empirical Validation:** The experiments in our original paper were designed to substantiate our theoretical claims. Following the reviewers' suggestions, we expanded our simulations to include signal-dependent noise and corruptions using the ImageNet-C corruption models. This expansion included shot noise, speckle noise, JPEG compression, pixelation, blurring, and combinations of these distortions. The results revealed an improved performance with Gaussian noise injection during training, with exceptions in some cases of pure blurring, JPEG compression, and pixelation. Notably, even in these scenarios, a minor addition of noise was found to stabilize the performance. It's important to mention that *ImageNet-C (Hendrycks et al. 2019)*, published in ICLR, is a widely accepted benchmark in related studies.
>
> Hendrycks, D., & Dietterich, T. (2019). Benchmarking neural network robustness to common corruptions and perturbations. ICLR 2019.
>
> In summary, our paper primarily contributes to the theory and guarantees of noise resilience rather than precise noise modeling and the development of denoising algorithms. We recognize the complexity of real-world image degradation and emphasize that our theoretical foundation is a critical step towards understanding the impact of Gaussian noise on I2I models, paving the way for future integration of more complicated, real-world noise models.

---

> > ### Comment · Reviewer_SHt1 · 2023-11-22
> >
> > You said, "Our objective is to establish a theoretical framework that guarantees robustness in image-to-image (I2I) translation models when Gaussian noise injection is used during training." However, the Gaussian noise is not presented in real-camera images. As you mentioned in the introduction, "The degradation of input image quality is a common occurrence in real-world scenarios, spanning from low-light conditions to data transmission through noisy channels". Your research plan is contradictory to your overall research motivation and is unable to address the problem you wish to study.
> >
> > So if you want to enhance the robustness, you should study the real-camera degradations, especially the noise. However, the noise types used in this paper are all synthetic. There are no methods used in this paper to force the evaluated noise to approach the distribution of real-camera noise. As I explained, the real-camera noise is much more complex than the synthetic noise. Please refer to [1], [3], and [7] for the detailed analysis about real-world noise.
> >
> > Current rebuttal and experiments do not address my concerns.

---

> > > ### Author Response · Authors · 2023-11-23
> > > **Additional Response to Reviewer Sht1**
> > >
> > > Reviewer Sht1 expressed the concerns that our theoretical approach contradicts our goal of exploring I2I model robustness to real-world noises. We respectfully disagree and wish to clarify that our method **aligns with** standard practices in theoretical research.
> > >
> > >  1. By starting with simplified noise and signal models, we aim to gain insights foundational for understanding I2I behavior in real-world scenarios. Making simplifying assumptions is a common approach when formally analyzing system's robustness. This approach is not contradictory but **an important step toward** applicability in more complex, real-world conditions.
> > >  2. To illustrate, in the realm of image denoising, early research often focused on images corrupted by iid Gaussian noise with known variance. This approach, while simplified, was crucial in laying the foundation for advanced, practical denoising methods. Our research emulates this trajectory, **providing vital theoretical insights** for future development of sophisticated noise/corruption models.
> > >  3. Importantly, in the absence of large-scale real noisy images, we have already evaluated on synthetic corruptions from Hendrycks et al. ICLR 2019 paper. With 2600+ citations, their benchmarks are **widely adopted for assessing model robustness**, see examples of references [R1]-[R3] listed below: Our additional experiments further align with standard community practice.
> > > - [R1] Fang, X., Ye, M., & Yang, X. (2023). Robust heterogeneous federated learning under data corruption. In Proceedings of the ICCV (pp. 5020-5030).
> > > - [R2] Gao, J., Zhang, J., Liu, X., Darrell, T., Shelhamer, E., & Wang, D. (2023) Back to the source: Diffusion-driven adaptation to test-time corruption. In Proceedings of the ICCV, 2023 (pp. 11786-11796).
> > > - [R3] Zhou, D., Yu, Z., Xie, E., Xiao, C., Anandkumar, A., Feng, J., & Alvarez, J. M. (2022, June). Understanding the robustness in vision transformers. In ICML (pp. 27378-27394).
> > >
> > > In conclusion, our study is an important step towards understanding noise robustness in I2I models, laying the theoretical groundwork for future exploration into more complex noise models. We believe this approach is well-aligned with our research goals and provides substantial insights to the field.

---

> ### Author Response · Authors · 2023-11-20
> **Reply to Reviewer SHt1-2**
>
> ### Q1-2: Comment on the lack of Technical contributions
>
> ### Reply 1-2:
> -   While noise injection during training has been explored in prior works, our key contribution is providing the first **rigorous theoretical analysis** of this phenomenon for image-to-image translation tasks.
> -  We have derived new analytical results that formally characterize the connections between distribution divergence, score matching, and the impact of Gaussian noise on alignment. To our knowledge, this level of ***mathematical insight*** has not been shown before.
> -   Our theoretical findings lead to new proofs and guarantees on the generalized robustness implications of Gaussian noise resilience. We formally show this can imply robustness to broader noise classes beyond the Gaussian distributions.
> -   These theoretical contributions go significantly beyond simply validating empirically known heuristics. We provide an in-depth analytical basis to fundamentally understand the mechanisms behind improved noise resilience.
> -   We derived the selection of optimal training noise levels for maximally enhancing real-world robustness. This constitutes a new theoretical contribution.
>
> In summary, while noise injection has been explored before, we provide significant technical contributions through new formal analysis unlocking fundamental insights into this phenomenon for I2I translation. Our rigorous analytical treatment and extensive theoretical results constitute the main value of this work.
>
>
> ### Q1-3: Comment on no discussion and comparison to the prior work that also studies real-camera degradations in I2I tasks.
> ### Reply 1-3:
>
> The primary objective of reference [7] is the generation of pairs of real-world noisy images from clean images through adversarial training. It is crucial to note that this approach is different from our focus, which is to train the GAN with images corrupted by known noise, enabling the translation from noisy images. Our primary goal is the translation of images from the source domain, with a specific emphasis on assessing the I2I system's robustness in the presence of noise, rather than precise noise modeling or denoising algorithm formulation.
>
> The alignment highlighted in 7 includes both image and noise alignment. While these alignments bear similarities to the divergence measure between distributions analyzed in our work, it's noteworthy that [7] does not consider these alignments to theoretical analyses. Instead, their approach relies solely on adversarial training of GANs for distribution alignment. It is widely acknowledged that the interpretability of adversarial learning in GANs often poses challenges. In contrast, our contribution involves the development of a comprehensive theoretical framework for *'alignment'* in this context. We delve into an in-depth analysis of the change in f-divergence with regard to training noise intensity, shedding light on the understanding of Gaussian noise injection in I2I systems.
>
>
> ### Q1-4: Comment on Code and pre-trained models are not submitted. The reproducibility cannot be checked.
> ### Reply 1-4:
> As all baseline models utilized in our study are open source, interested readers can readily reproduce our results. Due to the substantial size of the model weights, it is not feasible to include them as supplementary materials. However, we commit to providing complete access to all codes and pre-trained weights upon the acceptance of our work.
>
> ### Q1-5 Comment on the theoretical analysis of both signal-independent and signal dependant noises
> ### Reply 1-5:
> Thank you for your insightful comment regarding the inclusion of signal-dependent noise in our theoretical analysis. In our revised manuscript, we present empirical results that demonstrate improved performance under both signal-independent and -dependent noise conditions.
>
> Our current research focuses on establishing a solid theoretical foundation with signal-independent noise, a critical precursor to addressing the more complicated noises in practice.  Extending our theoretical framework to include signal-dependent noise is a complex challenge. It demands a reevaluation of the existing analysis to account for the statistical dependencies introduced by signal-dependent factors. We recognize the importance and practical relevance of such an extension for real-world applications and consider it a significant direction for future research. However, due to the complexity and the need for a more advanced analytical framework, it remains beyond the scope of our current work.

---

> ### Author Response · Authors · 2023-11-20
> **Reply to Reviewer SHt1-3**
>
> ### Q1-6 Comment on the use of the metric PSNR Gap to measure the domain discrepancy.
>
> ### Reply 1-6:
>
> We appreciate this suggestion provided by you. To make fair comparisons with baseline methods, we have chosen to employ the same objective metrics that are widely recognized and utilized in I2I evaluations. While PSNR-Gap is a valid metric in certain contexts, it is not commonly employed to measure the difference between two probability distributions in the study of I2I tasks.
>
> ### Q1-7 Comments on Recommended Dataset.
>
> ### Reply 1-7:
>
> We appreciate your suggestion to consider image denoising datasets like SIDD for our work. However, our paper primarily focuses on the domain of noise-robust I2I translation tasks,  and we use the widely recognized datasets AFHQ, FFHQ, and CUHK for cat-to-dog translation, face super-resolution, and face-to-sketch translation in our simultions.  It's worth noting that for evaluating performance under various noise conditions, we employ the Imagenet-C degradation models and corresponding source codes to create synthetic images, ensuring a comprehensive assessment in line with our research objectives. While image denoising datasets are valuable in their context, they are not directly applicable to our specific tasks. We hope this clarifies our choice of datasets and evaluation methodology for our research.

---

> ### Author Response · Authors · 2023-11-22
> **Request for Reviewer SHt1's Consideration and Timely Response**
>
> Dear Reviewer SHt1, we kindly request you to consider the perspectives and ratings of 6 from the other 2 Reviewers. Your insights are highly valuable to us, and we hope you can re-evaluate our work in light of the clarifications provided and the discussions thus far. We are eager to hear your thoughts and would greatly appreciate your response within the remaining discussion timeframe. Thank you for your time and consideration.

---

### Public Comment · ~Zhiwei_Jia1 · 2023-11-23
**Related Work**

Hi authors. I like your work and think it would be great to add the following work [1], which injects Gaussian noises to improve the semantic robustness of image-to-image translation, to the related work. Thanks.

[1] Semantically Robust Unpaired Image Translation for Data with Unmatched Semantics Statistics

---

> ### Author Response · Authors · 2023-11-23
> **Added related work**
>
> Dear Dr. Jia,
>  Thanks for pointing out this important related work. As it is very close to deadline, we only manage to add the reference. If this paper is accepted, we will add more descriptions about your work. Again, we really appreciate your suggestions.

---

### Meta-Review · Area_Chair_cKU4 · 2023-12-06

**Metareview:**

2x BA and 1x R. This paper proposes a robust theoretical framework to inject Gaussian and GMM noise into image-to-image translation models. The reviewers consistently appreciate their (1) detailed theoretical analysis and (2) extensive experiments. Although some concern appeared about the gap between theoretical Gaussian noise and real-camera degradation, the rebuttal with GMM extension closes the gap between theoretical derivation and practical usage. After all, this work is in line with standard methodologies in theoretical research, and the AC appreciates the analytical advancements proposed in their learning theory, which can be a useful milestone for future explorations in more realistic scenarios with real-world noise or distortions. The AC therefore leans to accept this submission.

**Justification For Why Not Higher Score:**

N/A

**Justification For Why Not Lower Score:**

N/A

---

### Decision · Program_Chairs · 2024-01-16

Accept (poster)